# Biodiversity resilience in a tropical rainforest

Timo Metz[1,2 ✉], Nina Farwig[3], Carsten F. Dormann[4], H. Martin Schaefer[5], Juan E. Guevara-Andino[6], Gunnar Brehm[7], Santiago Burneo[8], Anne Chao[9], Robin L. Chazdon[10,11], Robert K. Colwell[11,12,13,14], Ugo M. Diniz[15], David A. Donoso[6], María-José Endara[6], Santiago Erazo[8,16], Sebastián Escobar[6], Ana Falconí-López[6,17], Heike Feldhaar[18], Mishell Garcia Villamarin[19], Nina Grella[18], Katrin Heer[20], Michael Heethoff[21], Alexander Keller[19], Anna R. Landim[22,23], Sara D. Leonhardt[15], Eva Tamargo Lopez[3], Diego Marín-Armijos[24], Jörg Müller[17,25], Karla Neira-Salamea[6,26,27], Eike Lena Neuschulz[23], Karen M. Pedersen[1], Mark-Oliver Rödel[27], Matthias Schleuning[23,28], Thomas Schmitt[29], Michael Staab[1,30], Arianna Tartara[21], Boris A. Tinoco[31], Constance J. Tremlett[1], Marco Tschapka[16,32], Sybille Unsicker[33], Edith Villa-Galaviz[1] & Nico Blüthgen[1 ✉]

The UN Decade on Ecosystem Restoration aims to stop biodiversity losses[1]. Approximately 60% of tropical forests have already been lost or severely degraded[2], making restoration essential to achieve conservation goals. Recovery trajectories of trees have been studied intensively[3,4], but a comprehensive understanding of biodiversity recovery is lacking. Here we analyse recovery trajectories across trophic levels including 16 taxonomic groups from three kingdoms in a lowland tropical forest by investigating resistance to perturbation, recovery times and return rates to old-growth forest conditions. Abundance and diversity regained more than 90% and composition approximately 75% similarity to old-growth forests within 30 years, but full recovery takes several decades. Mobile animal communities acting as seed dispersers or pollinators had high resistance levels and recovered faster than trees or tree seedlings. Return rates contributed 1–2.5 times more than resistance to the recovery times of species composition. Taxon-specific recovery times could not be explained by simple mechanisms (life-history strategies, trophic level or mobility). We show the enormous potential of protecting naturally recovering secondary forests to stop and reverse biodiversity losses.

Tropical forests, harbouring at least 77% of tree species[5] and 62% of vertebrate species[6] known on Earth, are increasingly under pressure by a combination of anthropogenic stressors, including habitat conversion and degradation, land-use intensification and climate change[7,8]. Unrestrained deforestation, mainly for conversion to agricultural land[9], drives losses in forest area, structure, biodiversity, climate regulation and ecosystem services[10,11]. Reversing this trend presents an urgent global challenge, mirrored in the United Nations (UN) Decade on Ecosystem Restoration (http://www.decadeonrestoration.org)[1].

Old-growth forests are irreplaceable[11] and need to be conserved. Yet, more than half the tropical forests of the world have already been lost or degraded[2]. Because 70% of tropical forests are secondary[12] (regrowing after deforestation[13]), their conservation can contribute substantially to achieving global biodiversity conservation goals[14–16]. Tropical forests are dynamic ecosystems in which small-scale disturbance–recovery cycles are an inherent feature of the system[13]. However, the recovery potential of tropical forest biodiversity in secondary forests is unclear given the large spatial extent and accelerated rate of

[1]Ecological Networks Lab, Department of Biology, Technical University of Darmstadt, Darmstadt, Germany. [2]Institute for Condensed Matter Physics, Technical University of Darmstadt, Darmstadt, Germany. [3]Conservation Ecology, Department of Biology, University of Marburg, Marburg, Germany. [4]Department of Biometry and Environmental System Analysis, University of Freiburg, Freiburg, Germany. [5]Fundación Jocotoco, Quito, Ecuador. [6]Grupo de Investigación en Ecología y Evolución en los Trópicos-EETrop, Universidad de Las Américas, Quito, Ecuador. [7]Phyletisches Museum, Institute for Zoology and Evolutionary Research, Friedrich-Schiller-University Jena, Jena, Germany. [8]Museo de Zoología, Pontificia Universidad Católica del Ecuador, Quito, Ecuador. [9]Institute of Statistics, National Tsing Hua University, Hsinchu, Taiwan. [10]Forest Research Institute, University of the Sunshine Coast, Sippy Downs, Queensland, Australia. [11]Department of Ecology and Evolutionary Biology, University of Connecticut, Storrs, CT, USA. [12]University of Colorado Museum of Natural History, Boulder, CO, USA. [13]Center for Macroecology, Evolution and Climate, Natural History Museum of Denmark, University of Copenhagen, Copenhagen, Denmark. [14]Departamento de Ecologia, Universidade Federal de Goiás, Goiânia, Brasil. [15]Plant-Insect-Interactions Group, Technical University of Munich, Freising, Germany. [16]Institute of Evolutionary Ecology and Conservation Genomics, University of Ulm, Ulm, Germany. [17]Chair of Conservation Biology and Forest Ecology, Biocenter, University of Würzburg, Rauhenebrach, Germany. [18]Animal Population Ecology, Bayreuth Center for Ecology and Environmental Research (BayCEER), University of Bayreuth, Bayreuth, Germany. [19]Cellular and Organismic Interactions, Biocenter, Faculty of Biology, Ludwig-Maximilians-University München, Planegg-Martinsried, Germany. [20]Faculty of Environment and Natural Resources, Eva Mayr-Stihl Professorship for Forest Genetics, University of Freiburg, Freiburg, Germany. [21]Animal Evolutionary Ecology, Technical University of Darmstadt, Darmstadt, Germany. [22]Faculty of Biological Sciences, Goethe University Frankfurt, Frankfurt, Germany. [23]Senckenberg Biodiversity and Climate Research Centre (SBiK-F), Frankfurt, Germany. [24]Colección de Invertebrados Sur del Ecuador, Museo de Zoología CISEC-MUTPL, Departamento de Ciencias Biológicas y Agropecuarias, Universidad Técnica Particular de Loja, Loja, Ecuador. [25]Bavarian Forest National Park, Grafenau, Germany. [26]Humboldt-Universitat zu Berlin, Faculty of Life Sciences, Berlin, Germany. [27]Department of Evolutionary Diversity Dynamics, Museum für Naturkunde—Leibniz Institute for Evolution and Biodiversity Science, Berlin, Germany. [28]Department of Biology, Philipps-Universität Marburg, Marburg, Germany. [29]Department of Animal Ecology and Tropical Biology, Biocenter, University of Würzburg, Würzburg, Germany. [30]Institute of Ecology, Leuphana University Lüneburg, Lüneburg, Germany. [31]Escuela de Biología, Universidad del Azuay, Cuenca, Ecuador. [32]Smithsonian Tropical Research Institute, Ancón, Republic of Panama. [33]Plant-Environment-Interactions Group, Botanical Institute, University of Kiel, Kiel, Germany. ✉e-mail: timo.metz@googlemail.com; bluethgen@bio.tu-darmstadt.de

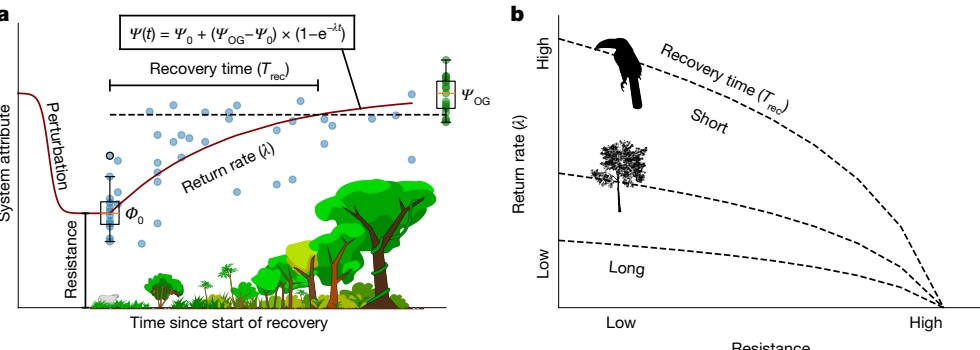

**Fig. 1 | Quantifying resistance, return rates and recovery times of biodiversity in a tropical rainforest. a**, For any recovering system attribute (for example, the diversity of a certain animal group), the resistance, defined as the amount remaining during perturbation (clear cut and agriculture) and the return rate towards the predisturbance reference state (old-growth forests) determine the recovery time. We fit a negative exponential function to the empirical data (**a**) to interpolate between the values in the agricultural plots $\Psi_0$ and the old-growth forest plots $\Psi_{OG}$ with the return rate $\lambda$ (Methods). The recovery time ($T_{rec}$) is calculated as the time difference between the intercept of this function and the value at 90% of the median of old-growth forest plots, represented by dotted lines. Blue dots represent species composition data for bees (independent samples from $n = 62$ plots). Boxplots mark active agriculture ($n = 12$) and old-growth forest ($n = 17$) plots (orange line shows median, boxes show data in 25th and 75th quartile and whiskers indicate 1.5× the interquartile range). An example calculation for recovery time, resistance and the return rate for the bee dataset can be found in Supplementary Note 1. **b**, The resistance and the return rate independently decrease the time until recovery to 90% of reference conditions (recovery time $T_{rec}$). We predict that communities of mobile animals recover faster, particularly as a result of a high return rate, than most trees, which have a much later age at first reproduction. **a**, Forest image reproduced with permissions from ref. 53, Ecological Society of America, under a CC BY-NC 4.0 licence. **a,b**, Silhouettes were reproduced from PhyloPic (https://www.phylopic.org/): frugivorous bird, created by E. Price under a CC BY 4.0 licence; tree, created by T. M. Keesey under a CC0 1.0 Universal Public Domain licence.

anthropogenic perturbations in tropical forests around the globe[2,13–15]. Several studies have shown the remarkable potential for natural regeneration of biomass, diversity and species composition of trees across the tropics[3,4,17–19]. The recovery of animal and microbe communities remains poorly studied. Studies indicate that species composition of different animal groups recovers within decades and that animal species richness may recover more rapidly than species composition[20–23]. However, these results are mostly based on small samples with few replicates and information across taxa is scattered among different studies, regions and forest types that cannot be compared quantitatively[24]. Understanding the recovery of several animal taxa alongside trees is essential to allow a holistic and robust estimation of the potential of secondary forests for biodiversity conservation. Improving our understanding of how quickly different taxonomic groups recover could enable more informed decisions about when to use natural regeneration as a cost-effective restoration tool or where assisted restoration measures may be required[8,15,19,24,25].

The recovery trajectories and recovery times of ecosystems following a perturbation depend on two components: resistance, defined as the ability to withstand disturbance; and recovery, which is the process of returning to the reference state as measured by the return rate[26,27] (Fig. 1). A common definition conceptualizes the combination of resistance and recovery as the resilience of the system[4,27–29] whereas other works define resilience more narrowly as the speed of return alone[30–34]. Resistance is related to attributes that confer tolerance to perturbation such as physiological or behavioural plasticity within taxa or features that provide protection against change[26,35,36]. High return rates are related to low trophic levels[37] and to life-history strategies that allow swift recovery after a perturbation, such as rapid recolonization or regrowth[26,38,39], rapid reproduction with many offspring, early reproductive age and short generation time[40–44]. The resistance and return rates of animal taxa that provide key functions, such as flower pollination or seed dispersal, may be essential for successful tropical forest recovery as 90% of the tree species are animal dispersed and 94% are pollinated by animals[45,46]. Disentangling the contribution of resistance and return rates to the recovery time of various taxa in a tropical rainforest provides a mechanistic understanding of the reaction of ecosystems to disturbances and may help in clarifying which measures

and conditions are necessary to facilitate short recovery times of an entire complex tropical rainforest ecosystem[31].

Here we calculate resistance, return rates, recovery trajectories and recovery times to 90% similarity of old-growth forest conditions of species composition, as well as the underlying species diversity and abundance, of 16 taxonomic groups with 10,856 species or morphospecies plus 23,590 bacteria sequences (amplicon sequence variants), all measured in a well-resolved chronosequence in the Chocó lowland rainforest in Ecuador[47]. We additionally calculate the relative recovery after 30 years to provide a robust estimate of recovery success within a timescale relevant to reach conservation goals[1]. The landscape consists of a mosaic of secondary forest, old-growth forest and agriculture with relatively high forest cover (approximately 75%; ref. 47) that is representative for many neotropical regions (median approximately 85% of 56 sites[17]).

Our 62 study plots are each 0.25 ha in size and include 6 actively used pastures and 6 cacao plantations, 33 secondary forests of variable age (1–38 years) recovering from previous agricultural use (16 pasture/17 cacao) and 17 old-growth forests as a reference (Supplementary Table 6). Our samples thereby provide snapshots across a full disturbance cycle before (old-growth forest), during (agriculture) and after perturbation (secondary forest). The recovery trajectory was modelled with a negative exponential function (Fig. 1a; equation (5)) with the return rate $\lambda$. It assumes a nonlinear recovery and an asymptotical approach to old-growth forest conditions with increasing time since land abandonment. Return rates often depend on the legacies of past land use[48,49]. We therefore compare the legacies of cacao and pasture, which are the two primary land uses in the region. We furthermore test whether ecological differences among taxa, that is life-history strategies, predominant mode of dispersal (aerial or terrestrial) or trophic level explain the order of recovery times, resistance and return rates. We assess the robustness and generality of our results and compare our findings to other tropical forest regions by calculating recovery times, resistance and return rates for data from literature.

## Recovery times of several taxa

Predicted recovery times (Fig. 2 and Extended Data Fig. 3) and trajectories (see Fig. 3 for composition and Extended Data Figs. 1 and 2 for

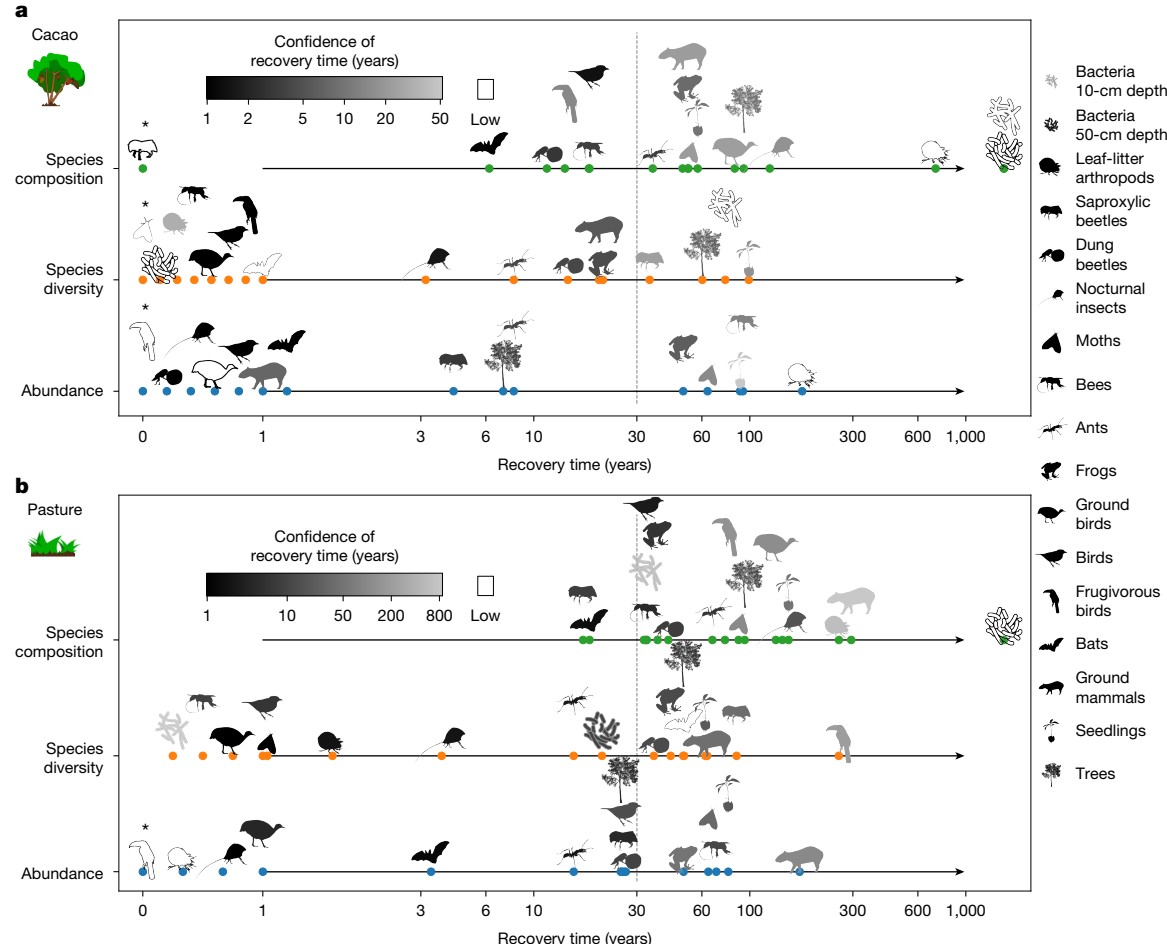

**Fig. 2 | Variation in predicted recovery time to 90% of old-growth forest values. a,b,** Recovery time is shown across taxa and community attributes for cacao (**a**) and pasture (**b**) legacies. Recovery time estimates the time span until species composition (Bray–Curtis similarity), species diversity (Shannon diversity, that is exponential Shannon entropy, Hill number of order 1) and total abundance of each investigated taxon group in secondary forests reaches 90% of old-growth forest reference (Fig. 1 and Methods). The dotted vertical line marks 30 years, the time at which the relative recovery was evaluated (Table 1). Coloured dots indicate different metrics. The 95% CIs for recovery times range from white (>50/820 years for cacao/pasture) and light-grey silhouettes (<50/820 years for cacao/pasture) to black (0 years) on a logarithmic scale; exact values of CIs of recovery times can be found in Table 1, Supplementary Table 1 and Extended Data Fig. 3. The 95% CIs were estimated using a jackknife procedure (Fig. 3 and Methods) based on $n-1$ iterations with $n$ being the number of independent plots sampled per taxon and legacy, with $n=39$ plots for cacao (**a**) and $n=40$ for pasture plots (**b**) in all cases, except: bacteria in 10-cm depth, $n=18/20$; bacteria in 50-cm depth, $n=14/14$; leaf-litter arthropods, $n=19/19$; frogs, $n=23/23$; seedlings, $n=24/24$; and nocturnal insects, $n=39/38$; given for cacao/pasture). Thereby, CIs indicate the range in which 95% of all jackknife curves are located. Taxa marked with an asterisk were set to a recovery time of 0 years (Supplementary Note 2). Bacteria showed extremely long

recovery time which could not be shown on the same scale. Taxa are represented by symbols and described in the legend; they are grouped by their sampling method and may partly overlap. For instance, 'nocturnal insects' represent one dataset based on light traps; other insects (for example, moths and bees) may contain nocturnal species as well. Bird data are based on sound trap recordings, whereas frugivorous birds were recorded by direct observations. The taxa in the legend are ordered according to their generation time ranging from low (bacteria) to high (trees). **a,b,** Cacao (**a**) and pasture (**b**) icons reproduced with permissions from ref. 53, Ecological Society of America, under a CC BY-NC 4.0 licence. Silhouettes of saproxylic beetle, bee, moth, dung beetle, nocturnal insect, ant, bird and ground bird were created by G. Brehm under a CC BY-SA 4.0 licence. The following silhouettes were reproduced from PhyloPic (https://www.phylopic.org/): frog and ground mammal, created by M. Michaud under a CC0 1.0 Universal Public Domain licence; bat, created by Y. Wong under a CC0 1.0 Universal Public Domain licence; frugivorous bird, created by E. Price under a CC BY 4.0 licence; seedling, created by M. Hofstetter under a CC BY 3.0 licence; tree, created by T. M. Keesey under a CC0 1.0 Universal Public Domain licence; leaf-litter arthropod, created by B. Lang under a CC BY 3.0 licence; bacteria 10-cm depth and bacteria 50-cm depth, created by L. Simons under a CC0 1.0 Universal Public Domain licence.

abundance and diversity) varied strongly across taxa. Abundance and diversity (Shannon diversity, that is Hill number of order 1) of many taxa remained high even during agricultural use and early recovery stages and recovered faster (between 0 years and 258 years, median cacao/pasture: 4.3/25.5 years (abundance), 3.2/20.7 (diversity)) than species composition (between 0 years and 724 years except for bacteria, which took substantially longer, median cacao/pasture: 51.6/76.7 years). This trend was significant across taxa (Wilcoxon signed-rank test for cacao/pasture: abundance versus composition−$V=9/12$, $P=0.008/0.009$, $n=15$ taxa excluding bacteria; diversity versus composition−$V=22/27$,

$P=0.01/0.02$, $n=17$ taxa). However, despite a relatively stable diversity, there was a pronounced turnover in species composition during succession and a delayed recovery of old-growth species. This result shows that dissecting attributes of ecosystem recovery is important for a comprehensive understanding of tropical forest recovery. The fast recovery of abundance and species diversity are important to obtain high levels of productivity and other ecosystem functions. Focusing on them, however, may underestimate the much longer time necessary for a complete community composition and thus the typical old-growth forest species to return.

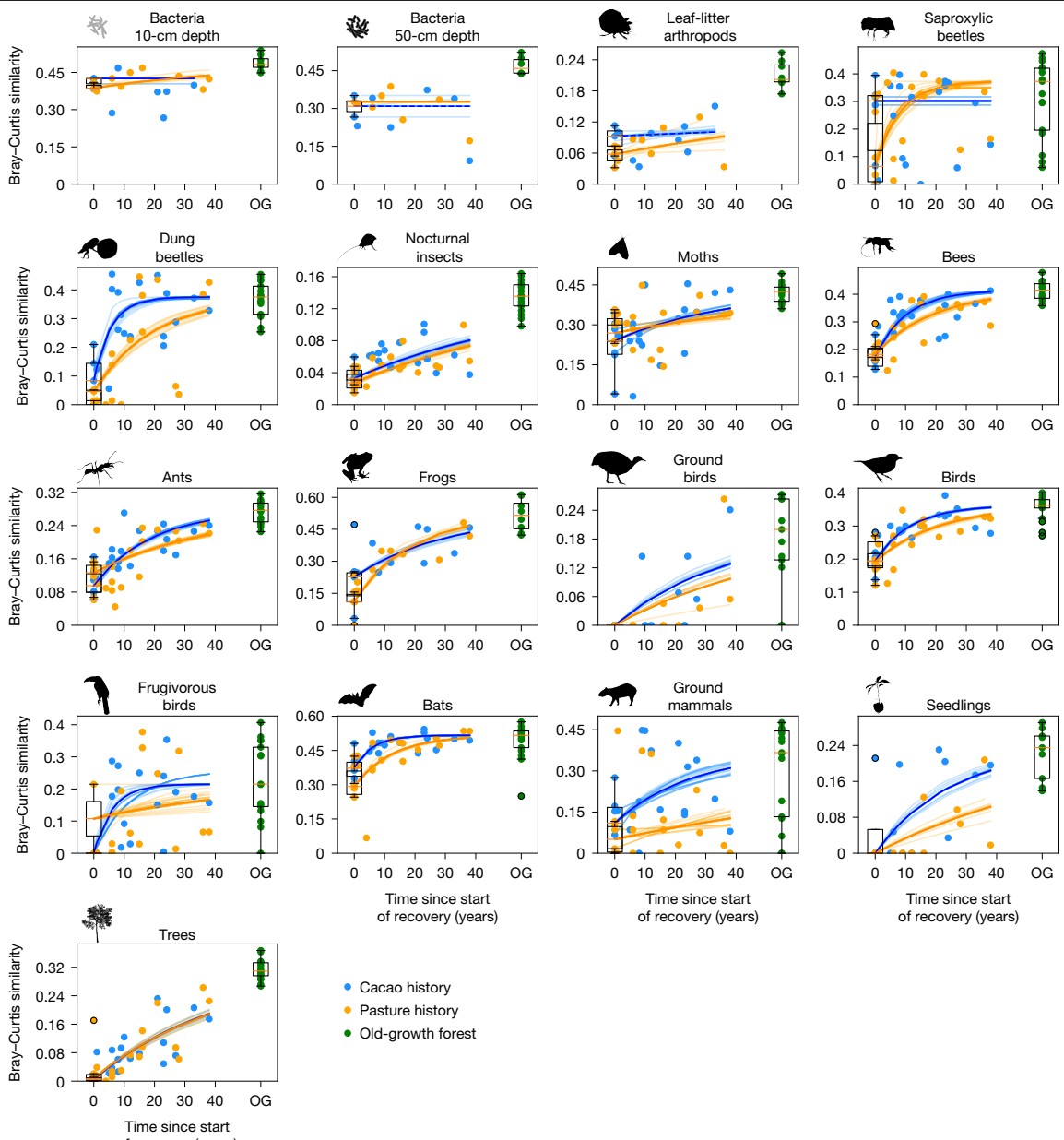

**Fig. 3 | Recovery trajectories of species composition.** Blue and orange dots represent the species composition similarity between cacao or pasture plots (either still in use at time $t = 0$ or recovering) and old-growth forest plots. Green dots indicate species composition similarity among old-growth forest plots. The blue and orange curves represent the estimated recovery trajectories for cacao and pasture legacy plots according to equation (5) (Methods). Dashed lines indicate curves with $\lambda$ not significantly different from 0. The light blue and orange curves indicate 95% CIs estimated using a jackknife procedure (Methods) based on $n - 1$ iterations with n being the number of sampled plots per taxon and legacy (caption of Fig. 2). Note that the $y$ axis has different ranges for different taxa. Boxplots are provided for $n = 6$ active cacao plots and $n = 6$ active pasture plots (except bacteria in 10-cm depth, $n = 3/3$; bacteria in 50-cm depth, $n = 2/1$; leaf-litter arthropods, $n = 3/3$; frogs, $n = 6/6$; seedlings, $n = 4/4$; given for cacao/pasture plots). Boxplots for old-growth (OG) forest plots are based on $n = 17$ plots (except for bacteria in 10-cm depth, $n = 8$; bacteria in

50-cm depth, $n = 6$; leaf-litter arthropods, $n = 8$; frogs, $n = 8$; seedlings, $n = 9$). Orange line in boxplots shows median, boxes show data in 25th and 75th quartile and whiskers indicate 1.5× the interquartile range. Silhouettes of saproxylic beetle, bee, moth, dung beetle, nocturnal insect, ant, bird and ground bird were created by G. Brehm under a CC BY-SA 4.0 licence. The following silhouettes were reproduced from PhyloPic (https://www.phylopic.org/): frog and ground mammal, created by M. Michaud under a CC0 1.0 Universal Public Domain licence; bat, created by Y. Wong under a CC0 1.0 Universal Public Domain licence; frugivorous bird, created by E. Price under a CC BY 4.0 licence; seedling, created by M. Hofstetter under a CC BY 3.0 licence; tree, created by T. M. Keesey under a CC0 1.0 Universal Public Domain licence; leaf-litter arthropod, created by B. Lang under a CC BY 3.0 licence; bacteria 10-cm depth and bacteria 50-cm depth, created by L. Simons under a CC0 1.0 Universal Public Domain licence.

The percentage of recovery after 30 years ranged between 37% and 100% (median cacao/pasture 100%/93%) for abundance, between 50% and 100% (median cacao/pasture 98%/95%) for diversity and between 31% and 100% (median cacao/pasture 81%/75%) for species composition (Table 1). This shows that secondary forests reach strong

similarity to old-growth forests quickly across all studied groups, even for species composition, underlining the high natural regeneration potential of mosaic landscapes typical of unplanned smallholder-driven deforestation. However, the time until full recovery of all taxa and attributes may take several decades and up to centuries.

**Table 1 | Predicted recovery times, resistance, return rates and relative recovery after 30 years for species composition for cacao and pasture legacies**

| Taxon | Resistance (%) | | Return rate (×10⁻³ per year) | | Relative recovery after 30 years (%) | | Predicted recovery time to 90% of old-growth forest (years) | |
|---|---|---|---|---|---|---|---|---|
| Legacy | C | P | C | P | C | P | C | P |
| Bacteria 10-cm depth | 88 (84–89) | 80 (80–82) | 0 (0–0) | 20 (14–33) | 88 (84–89) | 89 (87–92) | >300 | 33 (22–46) |
| Bacteria 50-cm depth | 67 (60–75) | 71 (68–74) | 0 (0–4) | 0 (0–0) | 67 (60–75) | 71 (68–74) | >300 | >300 |
| Leaf-litter arthropods | 46 (39–49) | 29 (24–31) | 2 (0–8) | 8 (3–17) | 50 (47–53) | 43 (35–55) | 725 (242->300) | 258 (127–886) |
| Saproxylic beetles | 81 (77–86) | 17 (9–26) | 0 (0–0) | 124 (96–315) | 81 (77–86) | 98 (95–100) | >300 | 17 (9–22) |
| Dung beetles | 22 (11–30) | 12 (8–13) | 178 (142–212) | 52 (43–62) | 100 (99–100) | 81 (76–86) | 12 (10–14) | 42 (35–50) |
| Nocturnal insects | 24 (24–25) | 20 (18–22) | 16 (15–19) | 15 (13–16) | 54 (51–57) | 49 (46–51) | 124 (106–140) | 142 (129–158) |
| Moths | 56 (46–67) | 63 (56–70) | 28 (16–40) | 15 (6–23) | 81 (79–84) | 76 (74–79) | 52 (43–73) | 89 (63–195) |
| Bees | 42 (38–47) | 44 (43–45) | 97 (81–116) | 52 (46–57) | 97 (95–98) | 88 (86–90) | 18 (15–22) | 33 (30–37) |
| Ants | 34 (30–39) | 46 (43–49) | 53 (46–60) | 25 (22–28) | 87 (85–88) | 75 (73–76) | 36 (33–39) | 67 (61–73) |
| Frogs | 44 (42–46) | 22 (21–22) | 34 (29–40) | 55 (49–65) | 79 (76–83) | 85 (82–89) | 52 (44–60) | 37 (32–42) |
| Ground birds | 0 (0–0) | 0 (0–0) | 27 (22–32) | 18 (12–20) | 56 (48–61) | 41 (28–46) | 85 (73–108) | 132 (113–251) |
| Birds | 54 (51–56) | 53 (52–55) | 85 (78–93) | 48 (44–52) | 96 (96–97) | 89 (88–90) | 18 (17–19) | 32 (30–35) |
| Frugivorous birds | 0 (0–0) | 50 (34–59) | 165 (80–257) | 21 (7–47) | 99 (91–100) | 73 (60–93) | 14 (9–29) | 77 (24–219) |
| Bats | 72 (70–74) | 57 (55–58) | 165 (150–178) | 81 (74–88) | 100 (100–100) | 96 (95–97) | 6 (6–7) | 18 (17–20) |
| Ground mammals | 30 (18–42) | 14 (5–23) | 40 (25–56) | 7 (2–13) | 79 (72–85) | 31 (25–38) | 49 (37–71) | 295 (176->300) |
| Seedlings | 0 (0–0) | 0 (0–0) | 40 (34–48) | 15 (12–18) | 70 (64–76) | 37 (30–42) | 57 (48–67) | 151 (128–202) |
| Trees | 1 (1–1) | 2 (1–3) | 25 (23–27) | 24 (22–26) | 53 (50–56) | 53 (49–55) | 94 (85–101) | 95 (88–105) |

C, cacao legacy; P, pasture legacy. Values in brackets indicate 95% CIs estimated with a jackknife procedure. Exact values can be found in Supplementary Table 1. Return rates given in the table need to be multiplied by 10⁻³ to obtain the actual values.

Long timescales of recovery were predicted for nocturnal insects (meta-barcoded bulk samples), leaf-litter arthropods and bacteria (Table 1 and Fig. 2). The slow recolonization of leaf-litter arthropods was consistent with their relatively low mobility (many taxa are wingless). Soil bacteria species composition showed no recovery at all, indicating that agricultural practices and strongly altered climatic conditions had a long-term legacy impact on the bacterial community. By contrast, animal taxa that recovered comparatively fast include important primary seed dispersers (birds and bats) and pollinators (bees). The positive effect of these animal groups on forest regeneration was evident in large-scale defaunation studies, particularly for the role of primates and birds[50]. However, the composition of old-growth trees recovered slowly compared with most animal taxa, owing to the long generation time of slow-growing tree species. Moreover, many old-growth tree species are rare, limiting their dispersal capacities. Moths, ground birds and ground mammals recovered on a similar timescale as trees. Because hunters target ground-dwelling large vertebrates, their slow recovery in species composition may indicate more disturbances but also more specialized requirements for resources. It remains to be studied whether the large variation in recovery times across taxa corresponds to an asynchrony, delay or decoupling of ecosystem functions and specific interaction partners with temporal succession. Our result showing that the species composition fully recovers for most taxa, but at different timescales, indicates that eventually all interactions may potentially rewire.

Recovery times of bacteria, saproxylic beetles, dung beetles, bees, ants, ground birds, birds, bats and ground mammals, as well as tree seedlings were shorter in former cacao plantations than in pastures (confidence intervals (CIs) in Table 1, Fig. 3 and Extended Data Figs. 1, 2 and 3), possibly facilitated by a combination of more favourable abiotic conditions (for example, shade and humidity) and available resources. Recovery times of trees and a few other taxa showed no consistent response to land-use legacy. For trees, this finding is in line with recent findings across several chronosequences that also did not identify an effect of previous land use[17].

We suggest that legacies have a stronger impact on animal than tree communities, as abandoned cacao plantations provide more available resources than pastures, thereby facilitating the recovery of diverse animal groups. For trees, however, we hypothesize that cacao trees may limit rather than facilitate key resources such as light and nutrients.

Differences in recovery times were largely unaffected by the choice of metrics, for example when using the series of Hill numbers for alpha- and beta-diversity and also applying rarefaction/extrapolation methods to standardize data for unequal sample coverage (Extended Data Fig. 4 and Supplementary Table 9).

For quickly recovering taxa, CIs were usually small and indicate high confidence in the estimated recovery time. Although our model assumes an ecologically plausible recovery trajectory, recovery times may change when using alternative models such as exponential, logarithmic or linear models (for example, refs. 17,47). We predict that this effect will be strongest for taxa with long recovery times, in which the recovery time is based on the model prediction rather than being covered by measurements in the chronosequence. For slowly recovering taxa, we suggest that the percentage of recovery after 30 years provides the most robust and useful measure for restoration guidelines.

## Resistance and return rates

Resistance values ranged from 0% to 100% for abundance (median cacao/pasture 29%/32%) and also for diversity (median cacao/pasture 64%/54%) and from 0% to 88% for species composition (median cacao/pasture 42%/29%).

Adult trees and tree seedlings had very low resistance levels in abundance, diversity and species composition (Fig. 4 and Extended Data Fig. 5), which mirrored the common practice that trees were mostly felled and removed for agriculture, except for a few remnant trees. By contrast, mobile animal taxa providing key mutualistic services to trees (bats, birds, moths and bees) were common in the agricultural

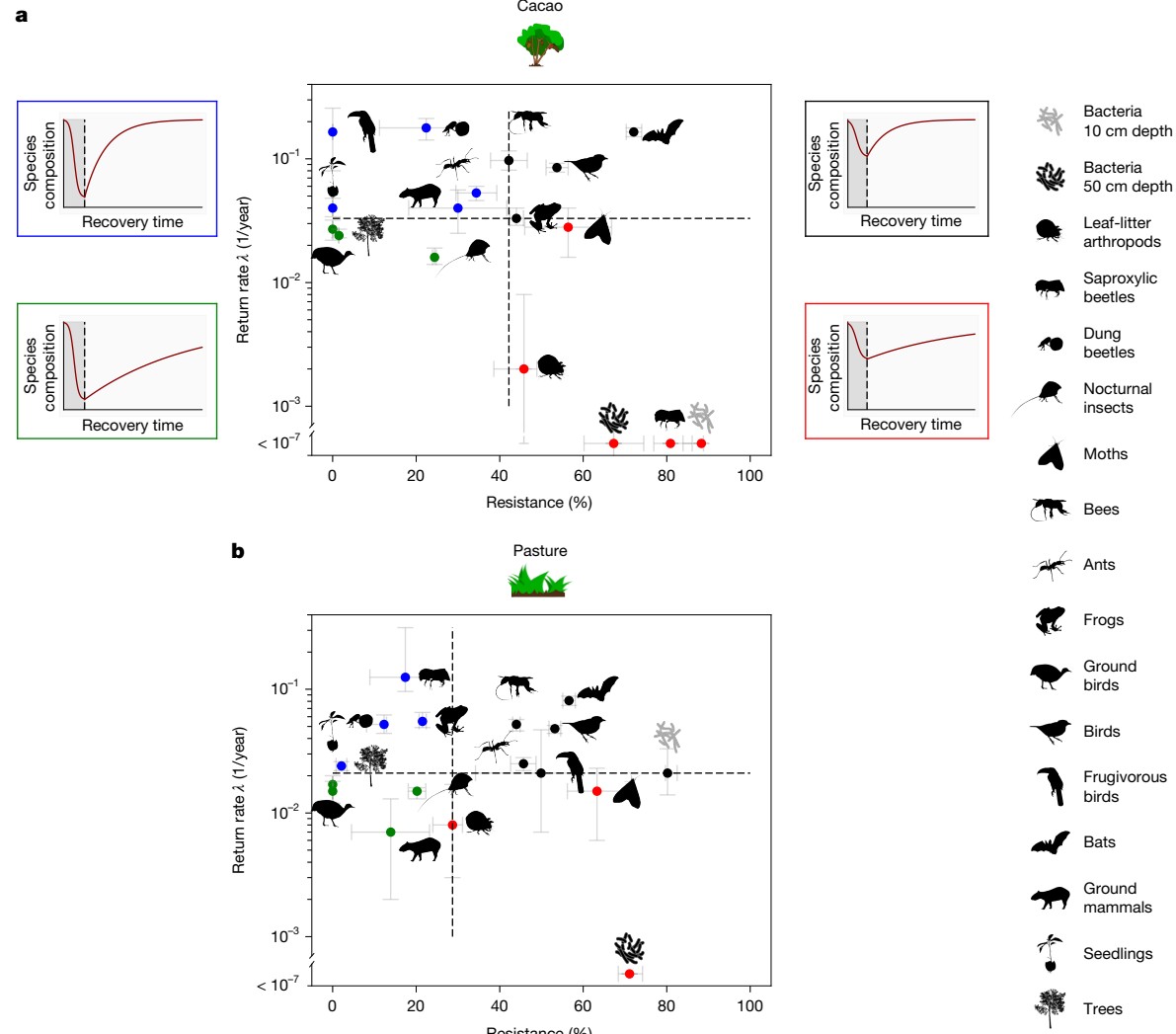

**Fig. 4 | Return rate and resistance combinations of species composition. a,b**, Results are shown for cacao (**a**) and pasture (**b**) legacies. Taxa with low resistance and low return rates (green dots) have long recovery times to the undisturbed reference state. Lowest recovery times are achieved when both resistance and return rate are highest (black dots). Blue dots indicate low resistance and high return rate and red dots high resistance and low return rate. Dashed lines mark median values of resistance and return rate across taxa. A characteristic trajectory (red curve) is shown for each scenario (different colours of dots indicate different scenarios and each scenario is framed with the same colour). Dark-grey shading in pictograms indicates perturbation. Dashed lines in pictograms represent the end of perturbation and the onset of recovery. Resistance and return rates for total abundance and species diversity are shown in Extended Data Fig. 5. The *y* axis is log-transformed. Whiskers represent 95% CIs estimated with a jackknife procedure based on $n - 1$ iterations

with *n* being the number of independent plots sampled per taxon and legacy (Fig. 2). **a,b**, Cacao (**a**) and pasture (**b**) icons reproduced with permissions from ref. 53, Ecological Society of America, under a CC BY-NC 4.0 licence. Silhouettes of saproxylic beetle, bee, moth, dung beetle, nocturnal insect, ant, bird and ground bird were created by G. Brehm under a CC BY-SA 4.0 licence. The following silhouettes were reproduced from PhyloPic (https://www.phylopic.org/): frog and ground mammal, created by M. Michaud under a CC 0 1.0 Universal Public Domain licence; bat, created by Y. Wong under a CC 0 1.0 Universal Public Domain licence; frugivorous bird, created by E. Price under a CC BY 4.0 licence; seedling, created by M. Hofstetter under a CC BY 3.0 licence; tree, created by T. M. Keesey under a CC 0 1.0 Universal Public Domain licence; leaf-litter arthropod, created by B. Lang under a CC BY 3.0 licence; bacteria 10-cm depth and bacteria 50-cm depth, created by L. Simons under a CC 0 1.0 Universal Public Domain licence.

and early successional stages and thus had high resistance levels especially for abundance (median cacao/pasture 46%/56%, range 25–75%) and diversity (median cacao/pasture 89%/78%, range 45–99%) but also for composition (median cacao/pasture 54%/53%, range 0–72%). Their high resistance in agricultural sites is a combination of resident species, but also those that actively forage there but breed elsewhere. Although both residents and spillovers from the forest pollinate or disperse seeds, the latter disperse genes among habitats and therefore facilitate seedling recovery. We hypothesize that the high resistance of these taxa accelerates the recovery of fast-growing pioneer trees (Supplementary Table 1) that provide nectar and fruit resources[51]. These resources further attract pollinators and seed dispersers, leading to a positive feedback loop for fast mutualistic network assembly.

Consistent with the forest-dependency of many species, the return rates of trees and seedlings were close to the median of all investigated taxa. However, frugivorous birds, bats and bees still had higher return rates than trees. We therefore posit that high resistance and return rates of seed dispersers and pollinators coupled with high productivity already during early succession initiate rapid recovery of tree species and forest-dependent taxa.

Remarkably, bacteria communities for both shallow and deep soil layers showed high resistance (88%/80% for 10-cm depth and 67%/71% for 50-cm depth for cacao/pasture), but very low return rates. This result may represent a case of arrested regeneration of a community.

For several animal taxa the return rates for abundance and diversity were about two orders of magnitude greater than species composition,

explaining the much shorter recovery times of abundance and diversity (Supplementary Table 1 and Extended Data Fig. 5). Accordingly, a random-forest analysis shows that return rates to reference conditions after perturbation are approximately 1–2.5 times (depending on the metric and land-use legacy; Supplementary Table 7) as important as resistance for recovery times across all taxa and kingdoms.

Our results further show that resistance and return rates were not correlated (Spearman's rank correlation cacao/pasture: abundance—$r_s$ = 0.24/0.06, $P$ = 0.45/0.84, $n$ = 15/15; diversity—$r_s$ = 0.23/0.35, $P$ = 0.5/0.24, $n$ = 17/17; composition—$r_s$ = −0.41/−0.06, $P$ = 0.14/0.84, $n$ = 17/17; Supplementary Table 9). Therefore, high resistance does not necessarily translate into fast recovery (for example, bacteria). We hypothesize that return rates are driven by a multiplicative effect of a species' mobility and the landscape context (for example, forest cover in the surrounding), whereas resistance is related to factors of land use itself, such as land-use intensity and duration. Dissecting resistance and return rates across many distinct taxa is thus fundamental to understanding the patterns of tropical forest recovery.

To highlight the general applicability of our findings for tropical forest restoration we analysed resistance, return rates and recovery times from 32 studies (Supplementary Table 5 and Extended Data Fig. 6) identified by a literature search (Methods). Consistent with our study, species composition, measured as similarity to the old-growth forest reference by means of the Jaccard index, took much longer to recover than species diversity, measured by means of number of species, that is Hill number of order 0 (Wilcoxon signed-rank test $V$ = 330, $P$ = 0.004, $n$ = 30) across all taxa and regions. Even though most studies included only one taxon per region or were poorly replicated, these fragmentary data also suggest that return rates are approximately 1.5–2 times as important (depending on the metric; Supplementary Table 8) than resistance for recovery times of species diversity and species composition.

The support of our findings has profound implications for conservation and the goals of the UN Decade on Ecosystem Restoration. We hypothesize that forest recovery in regions with low forest cover will be extremely slow or potentially arrested because of low return rates, whereas regions with high and diverse forest cover will quickly regain the original biotic community.

## Potential drivers of recovery times

Differences in environmental responses and recovery dynamics across taxa may correspond to a fundamental divergence in their complex life-history strategies and underlying trade-offs[41,42]. For example, species with longer generation time often show weaker responses to land-use intensity in grasslands[43], as well as a slower demographic post-disturbance recovery[44]. We thus tested whether age at first reproduction explained the order of recovery across taxa (for classification of taxa see Supplementary Table 4). Although taxa differed fundamentally in generation times, this variation could not explain their variation in recovery time, resistance and return rate of community composition (Supplementary Table 2). Even within five taxonomic groups (frogs, trees, birds, bees and saproxylic beetles) these life-history strategies had no consistent effect across species (Supplementary Table 2). Dispersal modes (aerial versus terrestrial) did not predict recovery time, resistance or return rates either. Finally, we categorized all recorded organisms into five trophic levels (detritivore, autotroph, herbivore, omnivore and carnivore) to test whether the dependency of higher-level consumers on biotic resources from lower levels slows down recovery, but again found no consistent support for this theory (Supplementary Table 2, corrected for multiple comparisons based on false discovery rate).

Overall, our results indicate that recovery times depend on more complex interactions that are not captured by simple metrics such as trophic levels, dispersal mode and age of reproduction. Consumer–resource interactions from predation, competition and facilitation to ecological fitting and mutualism may be more important for influencing the order of community recovery. However, a more fine-grained classification of species' life-history strategies also within taxa and including other trade-offs might reveal further insights. We suggest that resolving the temporal sequence of the order of species' establishment both within and across taxa is a promising way to understand functional ecosystem recovery.

## Conclusion

We show that the conservation of secondary forests rapidly and substantially restores biodiversity, as taxa recovered on average more than 90% of their abundance and diversity and approximately 75% of their compositional similarity to old-growth forests within only 30 years. The high resistance of mobile seed dispersers and pollinators, coupled with a high return rate of most taxa, contributes to tropical forests quickly regaining their diversity. We posit that high resistance and return rates of bees, bats and frugivorous birds play key roles at the onset of succession and these groups are drivers rather than passengers of tree recovery. However, full recovery of all taxa, also for composition, is predicted to take several decades. Because most secondary tropical forests worldwide are younger than 10 years and cleared in short intervals[52] they do not unfold their potential as conservation assets and biodiversity reservoirs. Extending turnaround times of forest management plans to several decades is essential to meet goals for biodiversity conservation. Recovery times for many animal groups were shorter in former cacao plantations compared with pasture, indicating that they should be prioritized for natural regeneration.

Overall, our results underline that cost-effective natural regeneration through abandonment of agricultural land is a powerful restoration strategy for tropical landscapes with smallholder agriculture to meet the UN Decade on Ecosystem Restoration goals.

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

## Methods

### Data

Data were sampled in the context of the Research Unit 'Reassembly', studying a chronosequence of a rainforest ecosystem in the Chocó tropical rainforest (see ref. 47 for more details).

### Study area

The study area is located in the reserves Canandé and Tesoro Escondido in the province of Esmeraldas in northwestern Ecuador (0.5° N, 79.2° W). Both reserves consist of a mosaic of actively used cacao and pasture plantations, secondary forest of varying age and primary forest. In this area, logging and agricultural land use started approximately five decades ago. Forests were logged for timber industry and subsequently mainly transformed into cacao plantations or pasture. Annual rainfall at the site is 4,700–6,000 mm. Mean temperature range 21–28 °C and relative humidity 90–100% (ref. 54).

### Chronosequence

The study design represents a chronosequence approach that substitutes space for time, that is several forest plots at various successional stages were studied in a short time span instead of monitoring the same patch over a long time span. This approach assumes that the plots follow similar successional trajectories and had comparable starting points. Information about previous land use and environmental conditions for each site are important and need to be representative and unbiased across the successional stages. This has been tested in detail for trees (see ref. 47 for analyses of spatial autocorrelation and environmental bias). We explicitly distinguished actively used cacao and pasture plots as well as secondary forest plots regenerating from previous use as cacao or pasture. Several plots for each category were distributed over the landscape (200 km²) to acknowledge variation in forest attributes (mean distances between plots in each category were around 5 km and did not differ across successional stages[47]). Our study comprised a total of 62 plots (each 0.25 ha) of which 12 plots represent active agriculture (6 cacao plantations and 6 pastures) and 33 plots secondary forest varying in age between 1 year and 38 years since start of recovery (17 recovering from use as cacao plantation and 16 from use as pasture, both with a similar time span). A total of 17 plots serve as old-growth forest reference and have neither been cut nor used. Supplementary Table 6 provides information on the ages of each plot. The topography is hilly and plots range in elevation from 159 m to 615 m. There was no significant difference in mean elevation between active, recovering or old-growth forest plots and no significant correlation between elevation and plot age[47]. Regeneration plots were located 58 m ± 7 m from the nearest forest edge and old-growth plots 388 m ± 72 m to the nearest forest edge[47].

Pastures were grazed extensively by low densities of cattle and occasionally by horses and were dominated by aggressive pasture grasses such as *Brachiaria* or *Axonopus scoparius*. Cacao plantations were monocultures of sun-exposed *Theobroma cacao* trees which were spaced 2–4 m apart and grew to heights of 5–10 m. These plantations were regularly treated with herbicides. Whereas cacao plantations generally lacked shade trees, pastures had some remnant trees or palms, or shrubs along creeks. These characteristics were also typical of the pastures and cacao plantations that were used several decades ago and which now represent regenerating forests in our study area. Old-growth forests contained large, slow-growing trees with potential for timber use and showed no signs of harvesting. Before they regenerated as secondary forests, pastures and cacao plantations had a similar duration of land use (mean 11.4 years, range 1–30 years). Regenerating pastures were larger (11.3 ha, range 1.2–46.7 ha) than cacao plantations (2.0 ha, 0.3–5.7 ha). All plots were located in a relatively intact landscape with a mean forest cover of 74% ± 2.8% (24–100%) within a 1-km radius. For further details on the study site, see ref. 47. Sampling methods for each taxon along the chronosequence are summarized in Supplementary Methods.

### Calculation of biodiversity metrics

For all taxa and plots we calculated the total abundance (number of individuals of each taxon) per plot and alpha-diversity Hill numbers (that is, effective number of species) for the orders $q = 0, 1$ and 2 (species diversity, Shannon diversity and Simpson diversity). We defined the similarity of species composition to old-growth forests as the mean of all pairwise comparisons between an agricultural or secondary forest plot to each of the 17 old-growth forest plots. Note that old-growth forests were distributed across the entire study area and represent natural spatial variation in biodiversity and composition. To define the old-growth forest reference value for species composition similarity, we thus compared each old-growth forest plot against all other 16 old-growth forest plots and calculated the mean similarity per plot. The median value of these 17 mean old-growth forest similarities was then used as the asymptotic reference for the recovery ($\Psi_{Ref}$ in equations (1) and (5)). Therefore, full recovery of species composition refers to the point at which the compositional similarity of an agricultural or secondary forest plot to an old-growth forest cannot be differentiated from the similarity among old-growth forests. We calculated the pairwise Bray–Curtis similarity as well as the beta-diversity Hill numbers for orders $q = 0, 1$ and 2. Beta-diversity Hill numbers were calculated as the fraction of gamma- and alpha-diversity Hill numbers of orders $q = 0, 1$ and 2 according to ref. 55:

$$q_{D_\beta} = q_{D_\gamma} / q_{D_\alpha} \tag{1}$$

where the alpha-diversity is calculated from the joint assemblage of both plots (by summing over the relative abundance of each species in each of the plots after joining both plots; ref. 55) and the gamma-diversity was defined for each pair of plots by pooling data from both communities. The beta-diversity Hill numbers were subsequently transformed to measures of species composition similarity according to ref. 56. The community overlap of order $q$ of 2 communities is there given by:

$$\text{overlap of order } q = \frac{(1/q_{D_\beta})^{(q-1)} - \left(\frac{1}{2}\right)^{q-1}}{1 - \left(\frac{1}{2}\right)^{q-1}} \tag{2}$$

This formula does not converge for $q = 1$. However, as $q$ approaches 1 the overlap is given by:

$$\text{overlap of order } 1 = (\ln 2 - H_{\beta,Shan}) / \ln 2 \tag{3}$$

with $H_{\beta,Shan}$ the Shannon index. As derived in ref. 57 the above formulas yield for $q = 0$ the Sorensen index, for $q = 1$ the Horn index and for $q = 2$ the Morisita–Horn index.

For Bray–Curtis similarity, species abundances were first transformed to relative abundances for increasing robustness against differences in total abundances among plots. We used the R package vegan v.2.6-4 to calculate the Bray–Curtis dissimilarity using the function vegdist and subsequently transformed it to the Bray–Curtis similarity (1 − Bray–Curtis dissimilarity).

We applied rarefaction/extrapolation methods to standardize the alpha- and beta-diversity Hill numbers of orders $q = 0, 1$ and 2 of each plot to the same sampling coverage according to refs. 58,59,60 (coverage values for alpha- and beta-diversity are reported in Supplementary Table 10).

For the alpha-diversity Hill numbers, we used the R package iNEXT[61] v.3.0.0 for standardization. For each dataset we first extrapolated the Hill numbers of each sampled plot to twice the observed number of individuals, as this is the value at which diversity can still be accurately

predicted[59,60]. We then noted the coverage value at this extrapolated point for each sampled plot. The minimum coverage value among the extrapolated samples was then used to standardize all samples (Supplementary Table 10). We thereby ensure that no plot is extrapolated beyond a point where predictions are still reliable.

For beta-diversity, we used the R package iNEXT.beta3D[60,62] v.1.0.1 for standardization. As for the non-standardized beta-diversity Hill numbers, we did a pairwise pooling of each combination of two plots. We then extrapolated the beta-diversity Hill numbers of each pooled pair of plots to twice the observed number of individuals by extrapolating the alpha- and gamma-diversity component separately. We then noted the minimum coverage value at this extrapolated point. We then rarefied or extrapolated each pooled pair of plots to this minimum coverage value (Supplementary Table 10). We subsequently transformed the extrapolated value of beta-diversity to a measure of species composition similarity as given by equations (2) and (3) and then calculated the mean value of each of these pairwise species composition similarity values between the active/regeneration plots and all old-growth forest plots. Because the beta standardization method requires a sufficient amount of data, the standardized beta value may be less than 1 (or the standardized dissimilarity may be negative) because of data sparsity and/or sampling variation. For seedlings for one plot a negative value for standardized beta-diversity ($q = 0$) was obtained. The same was true for frogs ('fast' strategists) for standardized beta-diversity ($q = 0$ and $q = 1$). For leaf-litter arthropods, a negative value for standardized beta-diversity ($q = 0$) was produced for two plots. These plots were thus excluded from further analysis for these taxa.

We focused on the Shannon diversity (Hill number of order 1) and Bray–Curtis similarity in the main text and the alternative metrics in the Supplementary Information to examine the robustness of our conclusions with respect to sampling completeness. The results of the above analysis for each taxon can be found in Supplementary Table 1 for results presented in the main text and Supplementary Table 3 for results with standardized metrics.

**Calculation of resistance, return rate and recovery time**
Resistance, as used here, measures the proportion of an ecological metric, such as species diversity, after a disturbance in relation to its value before the disturbance. We thus calculated the resistance using the median of the active agriculture plots, representing conditions after disturbance and the old-growth forest plots, representing conditions before disturbance. In some instances, the median value of an ecological metric of the active agriculture plots had a similar or even higher level than the median value of the old-growth forest plots. This was found for abundance or diversity of some taxa, confirming their high number of disturbance-tolerant species and species that can use a recovering habitat already early during succession. However, abundance and diversity levels above the old-growth reference can still be considered a disturbance effect. To make the resistance of those attributes that have a higher or a lower value than reference conditions comparable and obtain a generally applicable measure of similarity to reference conditions, we calculated the negative exponential of the Kullback–Leibler divergence similar to ref. 4. This measure bounds the resistance between 0% and 100% (0% meaning that the active agriculture plots lost all species compared with the old-growth forest plots and 100% implying that the active agriculture plots are completely similar in abundance, species diversity or species composition, to the old-growth forest plots) and was calculated as follows:

$$\text{Resistance} = \exp\left(-\left|\ln\left(\frac{\psi_0}{\psi_{\text{Ref}}}\right)\right|\right), \tag{4}$$

where $\psi_0$ and $\psi_{\text{Ref}}$ are the median of the respective index for the active agriculture and old-growth forest plots, respectively. This similarity measure is not completely symmetrical around 1. This

means that if the median of active agriculture plots is only 70% of the median of old-growth forest plots, the resistance is calculated as $\exp(-|\ln(0.7)|) = 0.7$, but if the median of the active agriculture plots is 130% of the median of old-growth plots (and thus also 30% different from it), the resistance is calculated as $\exp(-|\ln(1.3)|) = 0.77$. The values given by equation (1) are multiplied by 100 to obtain a percentage. We chose to use the median value as default as the median is more robust to outliers, unless the median of the old-growth forest was 0. In these cases, we chose to use the mean value for all calculations because we deemed a recovery to a value of 0 to not be biologically meaningful.

Recovery often occurs in a nonlinear fashion (for example, see ref. 4). Therefore, to calculate recovery time and the return rate, we fitted a negative exponential function, similar to the one used by ref. 4, as given by the equation

$$\psi(t) = \psi_0 + (\psi_{\text{Ref}} - \psi_0)(1 - e^{-\lambda t}), \tag{5}$$

where $\psi(t)$ is the value of the respective index at time $t$ and $\lambda$ is the constant return rate to the old-growth forest level (termed 'intrinsic recovery rate' in ref. 4). This equation models the return to reference conditions such that the difference between the value of the ecological property at time $t$ and the old-growth forest reference decreases with the constant rate $\lambda$. We used the function optimize.curve_fit from the scipy package v.1.10.0 in Python to fit equation (5) to the data. We made two separate models for those plots that had a legacy of cacao or pasture, respectively.

As the initial active level could be higher or lower than the level of old-growth forests, the function given by equation (5) could asymptotically reach the old-growth level from higher or lower values. We therefore solved equation (5) for $\psi(t) = 0.9\psi_{\text{Ref}}$ (for the case when recovery happens from lower than reference values) and $\psi(t) = 1.1\psi_{\text{Ref}}$, (for the case when recovery happens from higher than reference values) to allow a tolerance limit of ±10%. For most cases, recovery happened from lower than reference values for which the above condition then relates to the recovery time to 90% of the total value of the old-growth forest reference. This value was chosen (in line with ref. 4) because it allows to account for natural variation in old-growth forests but is still close enough to assume full recovery.

We then obtained the following formula with which we calculated the recovery time:

$$T_{\text{rec}} = \frac{-\ln(0.1\psi_{\text{Ref}}/|\psi_{\text{Ref}} - \psi_0|)}{\lambda} \tag{6}$$

Resilience is often either defined as the ability to maintain a qualitatively similar state in the face of disturbance and not switch to an alternative state (ecological resilience[27,28,63]) or the process of recovery of the system to reference conditions after a disturbance (engineering resilience[26–28]). The concept of engineering resilience is related to a single stability regime (which is the view we follow in this work), whereas ecological resilience assumes several alternative states. Recently it was proposed to consider resistance and recovery (measurable as the rate of return to reference conditions) as two components of resilience[27–29].

To avoid confusion with various definitions of resilience in the literature[26–34], throughout the manuscript we use the term 'return rate' for the specific measure represented by $\lambda$ and 'recovery time' for describing the time until 90% of the reference value is reached.

An example calculation for one specific group (bees) can be found in Supplementary Note 1. If the resistance of a taxon was above 90% it was considered to be undisturbed or instantaneously recolonized. Recovery time was then set to 0 years and the return rate could not be calculated.

In a few cases, the estimated recovery time was very large (much larger than 300 years). For the statistical tests, we therefore applied a criterion to set these values to 0, if the estimated recovery time was greater than 300 years and the standard deviations of the mean of active

agriculture plots and old-growth forest plots overlapped, as in these cases we assumed the long recovery time to be an artefact of a large spread of the data. This was the case for the abundance of frugivorous birds, the Shannon diversity of bats in cacao and the Bray–Curtis similarity of saproxylic beetles in cacao (Extended Data Fig. 7). For these datasets, which are also shown in Fig. 2, we show a detailed justification for this procedure in Supplementary Note 2. If the estimated recovery time was greater than 300 years, but the standard deviations of the mean of active agriculture plots and old-growth forest plots did not overlap, we assumed a case of arrested recovery and the large value of recovery time was kept. This was the case for the Bray–Curtis similarity of bacteria in 10-cm depth (cacao) and bacteria in 50-cm depth (cacao and pasture).

Several cases (21/24 for cacao/pasture) out of the 98 calculated recovery times presented in the main text for cacao and pasture had a recovery time longer than the 38 years covered by the chronosequence. The recovery times of taxa not recovering within 38 years may be considered less reliable than those of the taxa that recover within the time span covered by the chronosequence. However, even such taxa are important to include in the overall comparison despite their low return rate, high uncertainty of predictions and low $R^2$ (Supplementary Table 1) for our negative exponential model. This represents an unbiased characterization of different attributes of forest ecosystems and avoids a bias towards quickly recovering taxa. Despite possible alternative models to describe recovery, we focused on a single model to facilitate a direct comparison of recovery time and return rate. To obtain a comparable measure of recovery that is similarly robust for all taxa, we calculated the percentage of recovery after 30 years. The percentage of recovery after 30 years was calculated using equation (4) but instead of $\psi_0$ we inserted the modelled value $\psi_{30}$ of the respective index at 30 years after beginning of recovery, which was obtained by evaluating equation (5) after fitting it to the data at time $t = 30$ years. This time was chosen because it represents a timescale that is relevant for many restoration projects but was still within a time span well covered by our chronosequence to provide robust values.

Calculations of resistance, return rate, recovery time and the percentage of relative recovery after 30 years for our data were done in Python 3.11.4 and for the literature data in Python v.3.10.9 in a Jupyter notebook on a notebook server of v.6.5.2. All used code can be found in a digital repository.

### Error estimation

Errors for recovery times, resistance and return rate were estimated using a jackknife procedure[64]. This was preferred over bootstrapping, as bootstrapping omits on average approximately 31% of the data and for small datasets this leads to substantial issues in the estimation of the fit as only a substantially lower amount of information is available[65,66]. From each dataset and for each metric, separated for plots with cacao and pasture legacy but including all old-growth forests for both legacies, we removed each plot one time from the dataset, thereby creating $n$ jackknife samples with $n$ being the number of sampled plots for each taxon. Each of these jackknife samples therefore consisted of $n - 1$ plots. For each resulting jackknife sample, we calculated recovery time, resistance and return rates as explained before. CIs for recovery times, resistance and return rates reported in the main text and in Supplementary Table 1 represent the interval in which 95% of all curves obtained with this procedure were located. Obtained curves using the jackknife procedure for species composition (Bray–Curtis similarity) are visualized, together with the values of Bray–Curtis similarity to the old-growth forest plots and the fitted recovery trajectory, in Fig. 3 and in Extended Data Fig. 1 for abundance and Extended Data Fig. 2 for Shannon diversity.

### Relative importance of resistance and return rates

We calculated the importance of resistance and return rates to predict recovery time across taxa for the different studied metrics using random-forest regression. This approach has the advantage that non-linear effects of resistance and return rates are directly incorporated in the analysis and it has a clear way of interpreting the contributions of interacting predictors. We used the R package ranger v.0.16.0 with resistance and return rate as the predictor variables and the recovery time as the target variable. We used 'impurity' as importance measure as it is the default way to analyse feature importance.

For taxa with a resistance greater than 90%, return rates could not be calculated as 90% represents the threshold for full recovery in our study. We thus omitted these datapoints in further analyses. The random-forest model first creates bootstrapped samples from the original dataset with replacement. It then splits these bootstrapped samples into subsets, using certain values of the predictor variables (in this case the resistance and return rate) as split points. The split points are chosen such that the impurity (the variance of values in each node) is minimal. The impurity itself can then be used as a measure for the importance of the predictor variables in determining the response variable.

### Effect of life-history strategies, predominant dispersal mode and trophic level

To test predictions for recovery sequences, all taxa were a priori grouped into (1) 'slow–fast' life-history strategies, (2) dispersal modes and (3) trophic levels.

(1) Life-history strategies: corresponding to a 'slow–fast' continuum according to the life history of these taxa[43,44], taxa were ordered by an increasing age at first reproduction into four levels: (1) bacteria, (2) invertebrates, (3) vertebrates and (4) trees. This coarse rank order is also obviously consistent with their life span and body mass. A more fine-graded continuum[41–44] is prevented by large variability in each taxon and thus would require data at species level for all taxa.

(2) Dispersal modes were summarized into two levels: terrestrial dispersal was assigned for all non-flying vertebrate and invertebrate taxa (most species in the litter); aerial dispersal was assigned for birds, bats, ants (as they are dispersed by winged females and males), bees, beetles, moths, all nocturnal insects and other arthropods in hanging light traps (dominated by flying insects at their adult stage), but also for trees, as virtually all species are dispersed by birds, bats and wind (with few exceptions and those that are dispersed by larger mammals only).

(3) Trophic levels were assigned into five levels: (0) detritivores (dung beetles, saproxylic beetles, leaf-litter arthropods and all bacteria), (1) autotrophs (all plants), (2) herbivores (all frugivorous birds and bats, nectar or pollen-feeding bees and bats, herbivorous moths and other herbivorous insects and herbivorous mammals), (3) omnivores (ants, omnivorous birds and bats and omnivorous mammals) and (4) carnivores (insectivorous birds and bats and predatory or parasitic insect taxa and carnivorous mammals). To achieve this resolution, bird species were split into different trophic levels according to the Trophic Niche taken from AVONET v.3[67]. Bats were split into different trophic levels according to expert knowledge (M.T. and S.E.) and mammals and nocturnal insects were split into different groups according to the literature. The assigned groups for each taxon can be found in Supplementary Table 4.

Five of the taxa (frogs, trees, birds, bees and saproxylic beetles) have been split into two groups of species each, representing fast versus slow life-history strategies in each taxon. Frogs were split according to fast versus slow reproduction strategies, trees into fast- and slow-growing ones, bees and saproxylic beetles were split into social/non-social groups. This categorization was based on expert knowledge (frogs, M.-O.R.; trees, J.E.G.A.; bees, U.M.D and S.D.L.; saproxylic beetles, J.M.). Birds were split into short- and long-lived species according to their generation time. For each bird species we took the generation time from a database[68] and calculated the median of all species in the dataset.

Species with a generation time larger than the median were put into the 'long-lived species' group and the others into the 'short-lived species' group.

We calculated whether the different groups can explain the variation in recovery time, resistance and return rates for the different metrics by performing a Kruskal–Wallis test. Test criteria for parametric tests were not always met so, for consistency and robustness of the conclusions drawn, we used non-parametric rank tests throughout our study. To test for multiple comparisons, we applied a false discovery rate correction. All calculations were conducted with R v.4.3.2 and all used scripts can be found in a digital repository.

### Literature analysis

We performed a literature analysis to identify studies that were performed in tropical rainforests and contained data of one or more taxa for old-growth forest plots, actively used agriculture plots or other kinds of disturbance and secondary forest plots with a known age. These conditions were necessary for us to calculate resistance, return rates and recovery time and allow comparison to our data.

We performed a literature search including key reviews[20,69,70] that were already done in the field of forest restoration until the year 2016 and screened literature reported therein for the conditions stated before. We then performed a literature search in the Web of Science for the years 2016–2023 using the keywords tropical forest* + (primary forest* OR old-growth*) + (regeneration* OR secondary forest OR succession*) + (pasture* OR cacao* OR plantation* OR agricultur*). This search yielded a total of 432 studies of which the abstract was read to identify whether the criteria mentioned above were fulfilled.

Overall, we identified a total of 32 studies[71–103] for 12 different taxa fulfilling our criteria and for which species composition data were readily available. We calculated resistance, return rates and recovery time in a similar way as for our data and as described above for species diversity (number of species, that is Hill number of order 0) and species composition similarity (Jaccard index) without taking species relative abundances into account, as not all studies reported species abundances. We also calculated the importance of resistance and return rates for variation in recovery time across taxa using a random-forest regression as described above for our data.

### Reporting summary

Further information on research design is available in the Nature Portfolio Reporting Summary linked to this article.

## Data availability

Raw data are available in an online CodeOcean repository (https://doi.org/10.24433/CO.1040081.v4).

## Code availability

All code is available in an online CodeOcean repository (https://doi.org/10.24433/CO.1040081.v4).

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

**Acknowledgements** This work was supported by the Deutsche Forschungsgemeinschaft funded Research Unit REASSEMBLY (FOR 5207, coordination funds BL 960/12). We thank the Fundación Jocotoco and Fundación Tesoro Escondido (particularly C. Morelos-Juarez) for logistic support and permission to do research on their reserves. We would like to especially acknowledge local support from the staff in the Canandé and Tesoro Escondido reserves: K. Krauth (manager of the Chocó Lab); J. Carbajal (manager assistant) B. Tamayo (plot manager); Lady Condoy, L. de la Cruz, F. Quintero, J. Tacuri, J. Ninabanda, S. Vélez, I. Castellano, F. Cedeño (parabiologists); A. Zambrano (Canandé reserve staff); Y. Giler, P. Encarnacion, C. Morelos-Juarez, A. Villigu, P. Paredes and A. Argoti (Tesoro Escondido reserve staff). We acknowledge the Ministerio del Ambiente, Agua y Transición Ecológica for granting collection and research permits under the Genetic Resources Access Agreement number MAATE-DBI-CM-2021-0187. We thank J. Muñoz and K. Römer for project coordination and administration. We thank M. Sánchez-Pinillos and W. Castillo for comments on an earlier version of the paper. We thank F. Newell for discussions that improved this work.

**Author contributions** N.B., N.F., C.F.D. and T.M. conceived the idea of this paper. J.E.G.A., G.B., S.B., D.A.D., M.-J.E., H.F., K.H., M.H., A.K., S.D.L., D.M.-A., J.M., E.L.N., M.-O.R., M. Schleuning, T.S., B.A.T., M.T., S.U., U.M.D., S. Erazo, S. Escobar, A.F.-L., M.G.V., N.G., A.R.L., E.T.L., K.N.-S., K.M.P. and A.T. acquired data. R.L.C. and R.K.C. assisted the interpretation of the results. C.J.T. and E.V.-G. coordinated field research. N.B. and T.M. wrote the initial draft of the manuscript with considerable input from J.M. and H.M.S. T.M. performed data analysis and visualization. A.C. provided parts of the code used in this study and assisted parts of the statistical analysis. M. Staab assisted the statistical analysis of the data. All authors commented on the paper.

**Competing interests** The authors declare no competing interests.

**Additional information**
**Correspondence and requests for materials** should be addressed to Timo Metz or Nico Blüthgen.

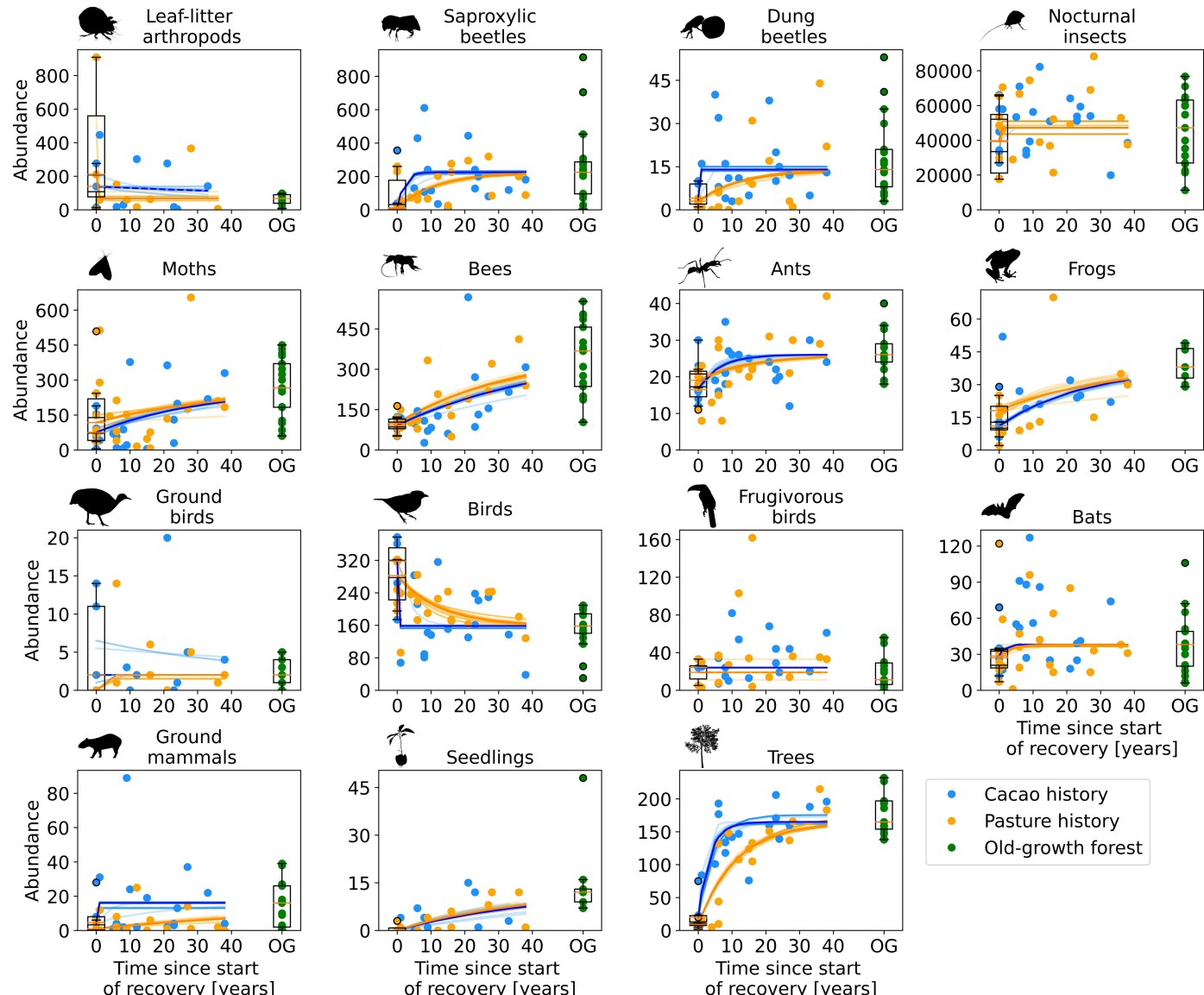

**Extended Data Fig. 1 | Recovery trajectories of abundance.** Blue and orange dots represent abundance (number of all individuals of each taxon group) in agriculture plots or recovering plots with cacao or pasture land use legacy. Green dots indicate abundance in old-growth forest plots. The blue and orange curves represent the estimated recovery trajectories for cacao and pasture legacy plots according to Eq. 5 (see Methods). Dashed lines indicate curves with $\lambda$ not significantly different from 0. The light blue and orange curves indicate 95% confidence intervals estimated using a jackknife procedure (see Methods) based on n-1 iterations with n being the number of independent plots sampled per taxon and legacy, with n = 39 for cacao and n = 40 for pasture plots in all cases, except bacteria in 10 cm depth: n = 18/20; bacteria in 50 cm depth: n = 14/14; leaf-litter arthropods: n = 19/19; frogs: n = 23/23; seedlings: n = 24/24; nocturnal insects: n = 39/38 for cacao/pasture plots. Boxplots are provided for n = 6 active cacao plots and n = 6 active pasture plots (except for bacteria in 10 cm depth: n = 3/3; bacteria in 50 cm depth: n = 2/1; leaf-litter arthropods: n = 3/3; frogs: n = 6/6; seedlings: n = 4/4 for cacao/pasture). Boxplots for old-growth (OG) forest plots are based on n = 17 plots (except for bacteria in 10 cm depth: n = 8; bacteria in 50 cm depth: n = 6; leaf-litter arthropods: n = 8; frogs: n = 8; seedlings: n = 9). Orange line in boxplots shows median, boxes show data in 25th and 75th quartile and whiskers indicate 1.5x the interquartile range. Silhouettes of saproxylic beetle, bee, moth, dung beetle, nocturnal insect, ant, bird and ground bird were created by G. Brehm under a CC BY-SA 4.0 licence. The following silhouettes were reproduced from PhyloPic (https://www.phylopic.org/): frog and ground mammal, created by M. Michaud under a CC0 1.0 Universal Public Domain licence; bat, created by Y. Wong under a CC0 1.0 Universal Public Domain licence; frugivorous bird, created by E. Price under a CC BY 4.0 licence; seedling, created by M. Hofstetter under a CC BY 3.0 licence; tree, created by T. M. Keesey under a CC0 1.0 Universal Public Domain licence; leaf-litter arthropod, created by B. Lang under a CC BY 3.0 licence; bacteria 10-cm depth and bacteria 50-cm depth, created by L. Simons under a CC0 1.0 Universal Public Domain licence.

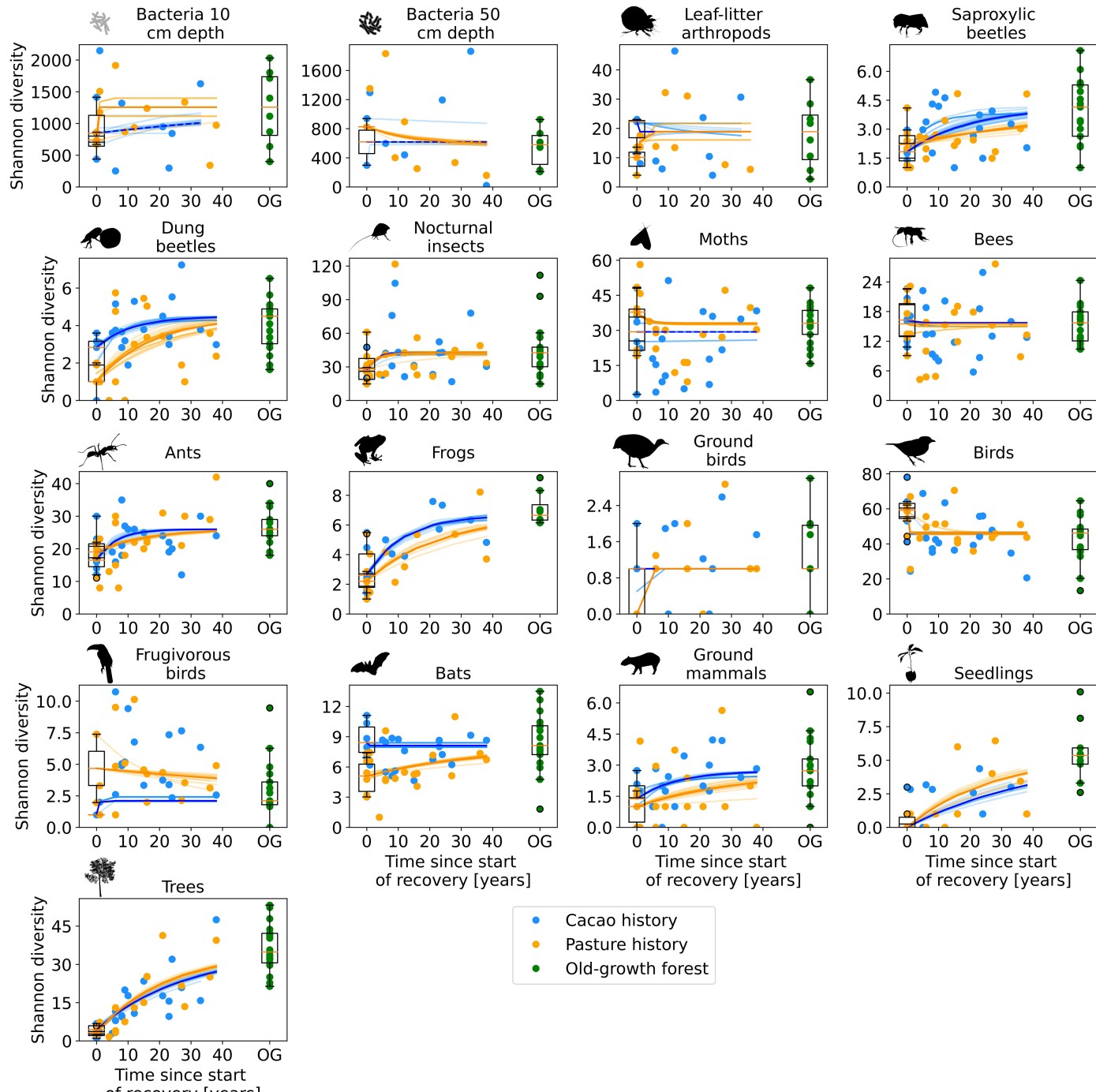

**Extended Data Fig. 2 | Recovery trajectories of species diversity.** Blue and orange dots represent species diversity (alpha-diversity Hill-number of order q = 1, i.e. Shannon diversity) in agriculture plots or recovering plots with cacao or pasture land use legacy. Green dots indicate species diversity in old-growth forest plots. The blue and orange curves represent the estimated recovery trajectories for cacao and pasture legacy plots according to Eq. 5 (see Methods). Dashed lines indicate curves with $\lambda$ not significantly different from 0. The light blue and orange curves indicate 95% confidence intervals estimated using a jackknife procedure (see Methods) based on n-1 iterations with n being the number of independent plots sampled per taxon and legacy. See caption of Extended Data Fig. 1 for sample sizes. Orange line in boxplots shows median, boxes show data in 25th and 75th quartile and whiskers indicate 1.5x the interquartile range. Silhouettes of saproxylic beetle, bee, moth, dung beetle, nocturnal insect, ant, bird and ground bird were created by G. Brehm under a CC BY-SA 4.0 licence. The following silhouettes were reproduced from PhyloPic (https://www.phylopic.org/): frog and ground mammal, created by M. Michaud under a CC0 1.0 Universal Public Domain licence; bat, created by Y. Wong under a CC0 1.0 Universal Public Domain licence; frugivorous bird, created by E. Price under a CC BY 4.0 licence; seedling, created by M. Hofstetter under a CC BY 3.0 licence; tree, created by T. M. Keesey under a CC0 1.0 Universal Public Domain licence; leaf-litter arthropod, created by B. Lang under a CC BY 3.0 licence; bacteria 10-cm depth and bacteria 50-cm depth, created by L. Simons under a CC0 1.0 Universal Public Domain licence.

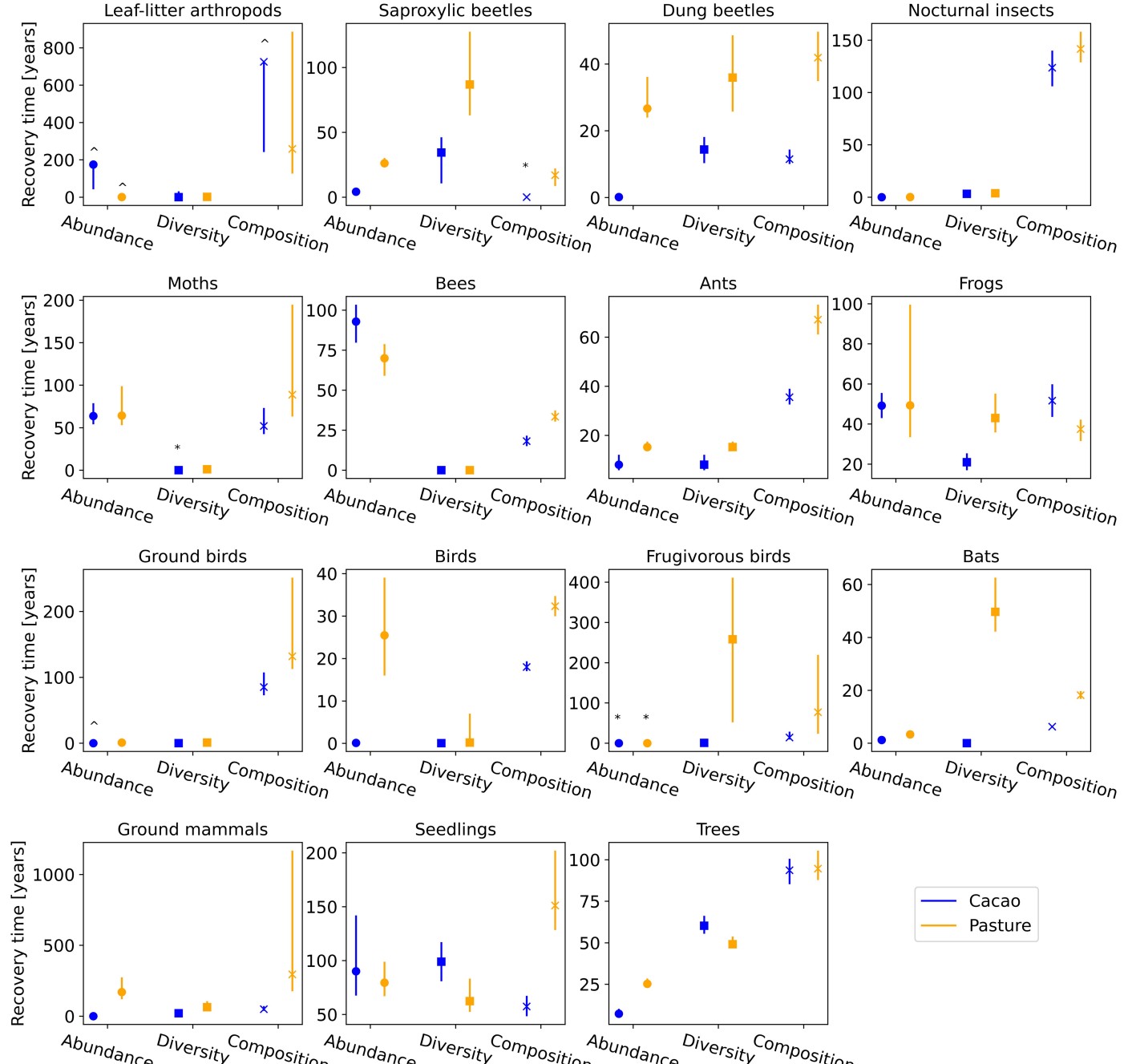

**Extended Data Fig. 3 | Predicted recovery times to 90% of old-growth forest values.** Dots refer to the predicted value of the recovery time to 90% of old-growth forest values. Whiskers represent 95% confidence intervals calculated with a jackknife-procedure based on n-1 iterations with n being the number of sampled plots per and legacy (bacteria in 10 cm depth: n = 18/20; bacteria in 50 cm depth: n = 14/14; leaf-litter arthropods: n = 19/19; frogs: n = 23/23; seedlings: n = 24/24; nocturnal insects: n = 39/38; all other taxa: n = 39/40 for cacao/pasture plots). Asterisks indicate predicted recovery times that were manually set to 0 (see Supplementary Note 2). Recovery times of bacteria were very long and could not be displayed, but are presented in Supplementary Table 1. Recovery times marked with a "^" had very large upper confidence intervals, which were not displayed but can be seen in Supplementary Table 1.

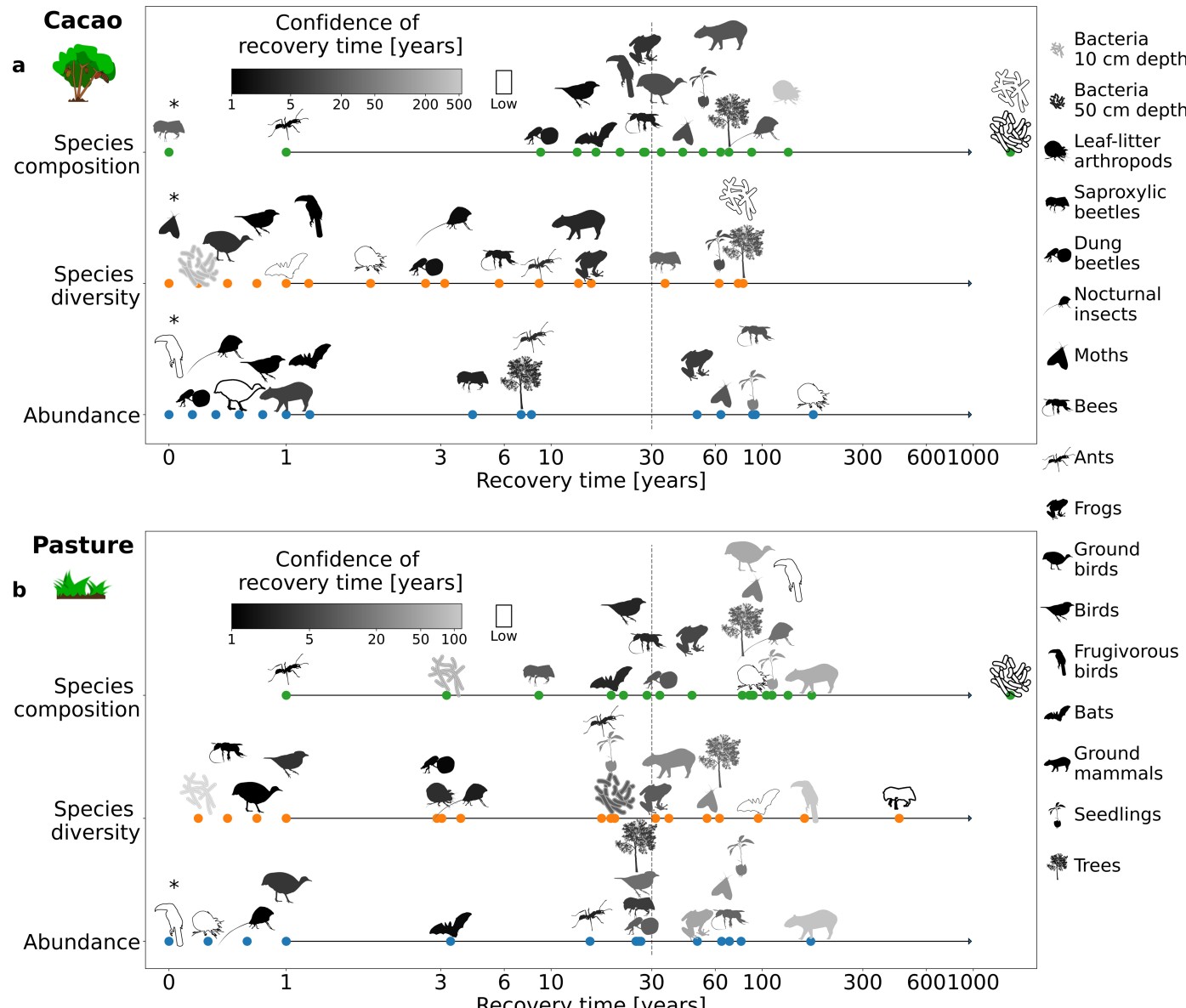

**Extended Data Fig. 4 | Variation in recovery time estimates across taxa and community attributes based on standardized diversity metrics (compare with Fig. 2).** Recovery time estimates the time span until species composition similarity, species diversity and total abundance of secondary forests reaches 90% of the median of old-growth forest plots as a reference. Species composition was calculated using the Horn-index (transformation of the beta-diversity Hill-number of order q = 1). Species diversity was calculated using the standardized alpha-diversity Hill-number of order q = 1 (Shannon diversity). Colored dots on the axis represent the estimated recovery time on the time arrow. Confidence intervals of recovery times range from white (»450 years) and light-grey silhouettes (<450 years) to black (0 years) on a logarithmic scale; exact values of confidence intervals of recovery times can be found in Supplementary Table 1. Exact values of confidence intervals of recovery times can be found in Supplementary Table 1. 95% confidence intervals were estimated using a jackknife procedure (see Fig. 3 and Methods) based on n-1 iterations with n being the number of sampled plots per taxon and legacy. See Extended Data Fig. 1 for sample sizes. Taxa marked with an asterisk were set to a recovery time of 0 years (see Supplementary Note 2). Species compositions of bacteria showed extremely long recovery times which could not be displayed on the same axis. Taxa symbols are described in the legend. **a**,**b**, Cacao (**a**) and pasture (**b**) icons reproduced with permissions from ref. 53, Ecological Society of America, under a CC BY-NC 4.0 licence. Silhouettes of saproxylic beetle, bee, moth, dung beetle, nocturnal insect, ant, bird and ground bird were created by G. Brehm under a CC BY-SA 4.0 licence. The following silhouettes were reproduced from PhyloPic (https://www.phylopic.org/): frog and ground mammal, created by M. Michaud under a CC0 1.0 Universal Public Domain licence; bat, created by Y. Wong under a CC0 1.0 Universal Public Domain licence; frugivorous bird, created by E. Price under a CC BY 4.0 licence; seedling, created by M. Hofstetter under a CC BY 3.0 licence; tree, created by T. M. Keesey under a CC0 1.0 Universal Public Domain licence; leaf-litter arthropod, created by B. Lang under a CC BY 3.0 licence; bacteria 10-cm depth and bacteria 50-cm depth, created by L. Simons under a CC0 1.0 Universal Public Domain licence.

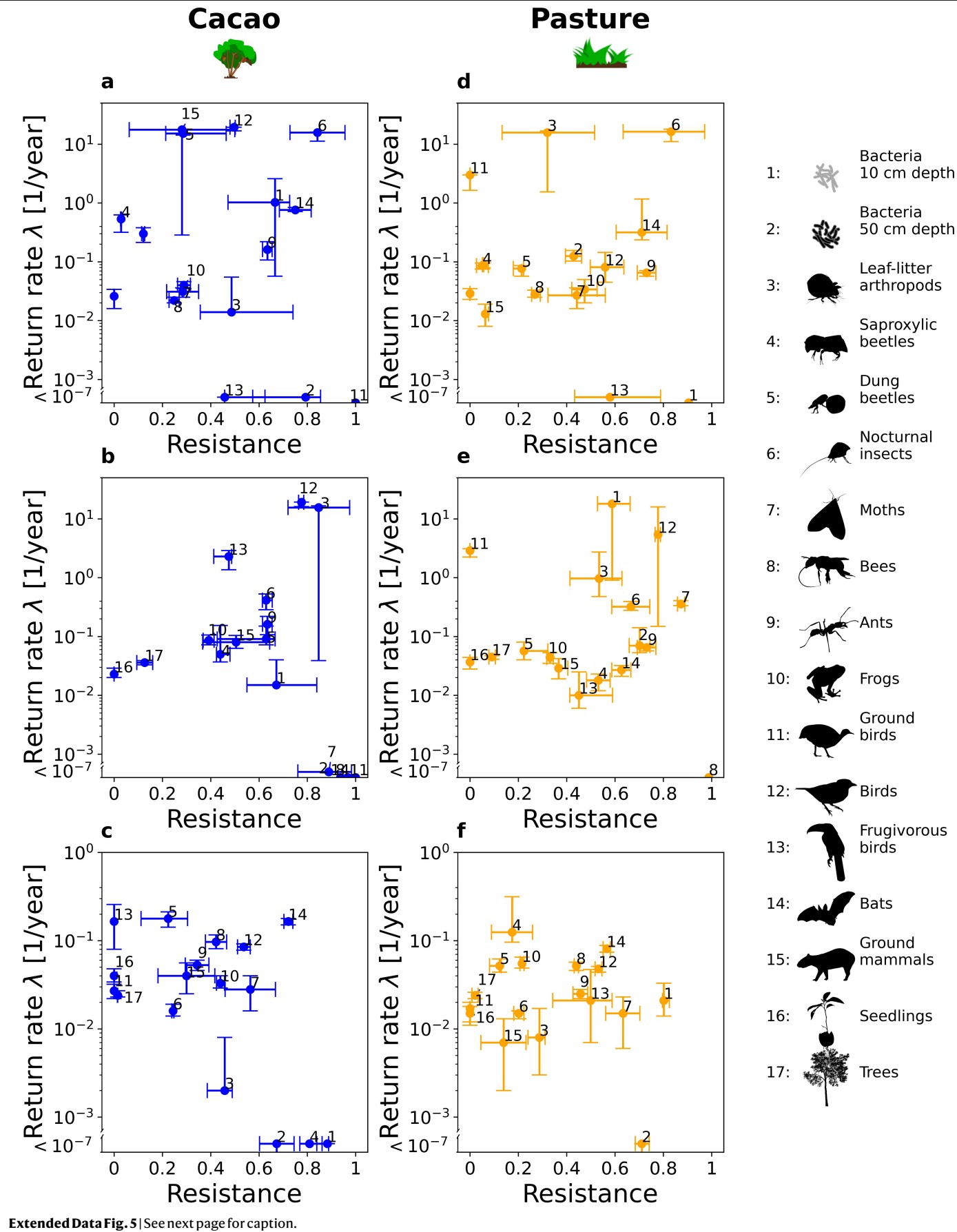

**Cacao**

**Pasture**

**Extended Data Fig. 5 |** See next page for caption.

**Extended Data Fig. 5 | Resistance and return rates of various taxa for abundance (A/D), species diversity (B/E) and species composition (C/F).** Species diversity here refers to Shannon diversity (i.e. Hill-number of order 1), while species composition similarity here refers to Bray-Curtis similarity. Ordering of taxa in the legend is according to their generation time from high (bacteria) to low (trees). Whiskers represent 95% confidence intervals estimated with a jackknife procedure based on n-1 iterations with n being the number of sampled plots per taxon and legacy. See Extended Data Fig. 1 for sample sizes. Silhouettes of saproxylic beetle, bee, moth, dung beetle, nocturnal insect, ant, bird and ground bird were created by G. Brehm under a CC BY-SA 4.0 licence. The following silhouettes were reproduced from PhyloPic (https://www.phylopic.org/): frog and ground mammal, created by M. Michaud under a CC0 1.0 Universal Public Domain licence; bat, created by Y. Wong under a CC0 1.0 Universal Public Domain licence; frugivorous bird, created by E. Price under a CC BY 4.0 licence; seedling, created by M. Hofstetter under a CC BY 3.0 Universal Public Domain licence; tree, created by T. M. Keesey under a CC0 1.0 Universal Public Domain licence; leaf-litter arthropod, created by B. Lang under a CC BY 3.0 Universal Public Domain licence; bacteria 10-cm depth and bacteria 50-cm depth, created by L. Simons under a CC0 1.0 Universal Public Domain licence.

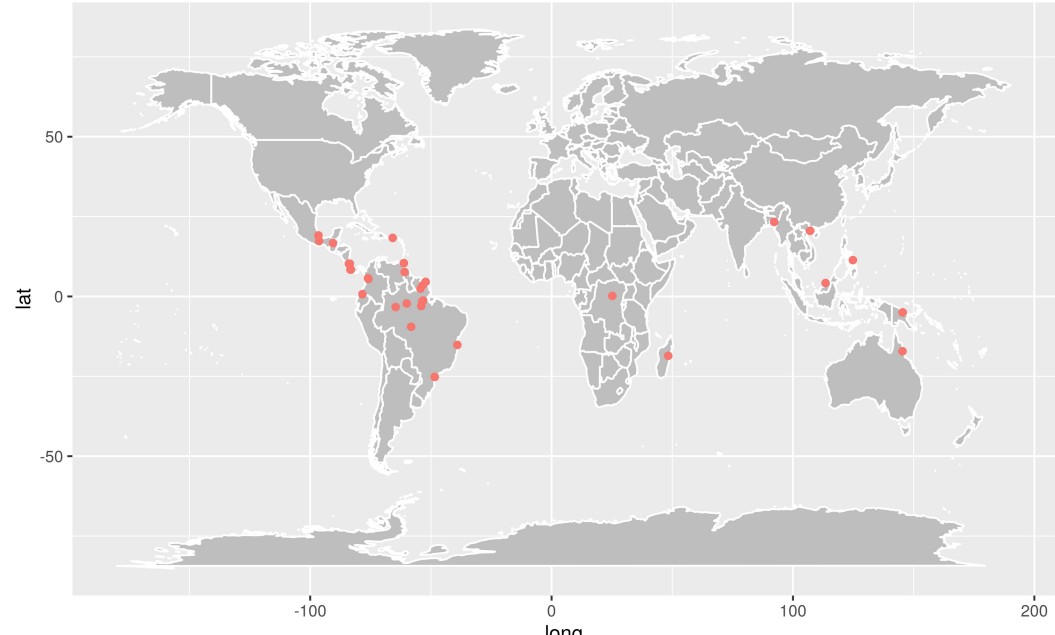

**Extended Data Fig. 6 | Locations of studies identified in the literature analysis.** References and exact location can be found in Supplementary Table 5. World map data were obtained from the maps R package[104].

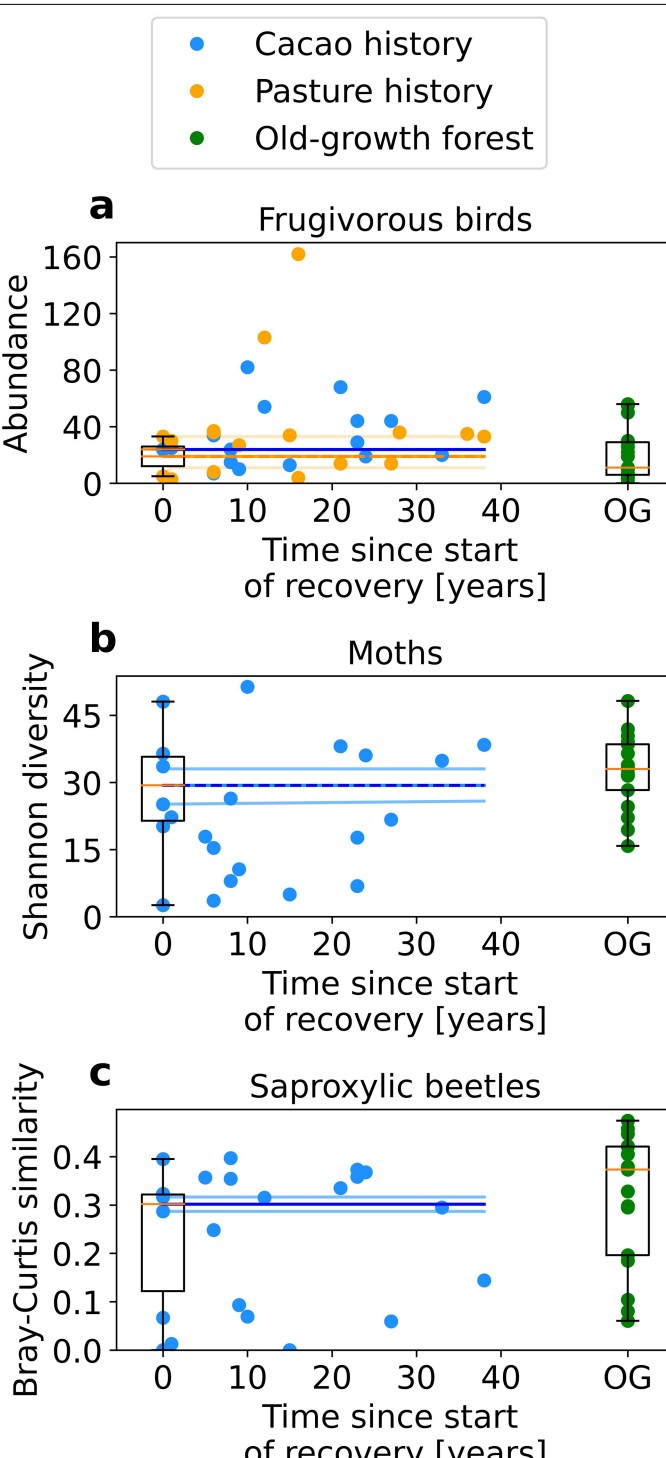

**Extended Data Fig. 7 | Recovery trajectory of the abundance of frugivorous birds (a) the Shannon diversity of bats (b) and the Bray-Curtis similarity of saproxylic beetles (c).** Blue and orange dots indicate values of active and recovering plots with cacao and pasture legacy, respectively, green dots indicate old-growth forest plots. The blue and orange lines indicate the recovery trajectory according to Eq. 5 for plots with cacao and pasture legacy, respectively. Dashed lines indicate curves with $\lambda$ not significantly different from 0. Light blue and orange curves indicate 95% confidence intervals estimated using a jackknife procedure based on n-1 iterations with n being the number of sampled plots per taxon and legacy (n = 39/40 for cacao/pasture). Boxplots are provided for active agriculture plots (n = 6/6 for cacao/pasture) and old-growth forest plots (n = 17). Orange line in boxplots shows median, boxes show data in 25th and 75th quartile and whiskers indicate 1.5x the interquartile range).

# Reporting Summary

## Statistics

For all statistical analyses, confirm that the following items are present in the figure legend, table legend, main text, or Methods section.

| n/a | Confirmed | |
|---|---|---|
| ☐ | ☒ | The exact sample size (*n*) for each experimental group/condition, given as a discrete number and unit of measurement |
| ☐ | ☒ | A statement on whether measurements were taken from distinct samples or whether the same sample was measured repeatedly |
| ☐ | ☒ | The statistical test(s) used AND whether they are one- or two-sided *Only common tests should be described solely by name; describe more complex techniques in the Methods section.* |
| ☐ | ☒ | A description of all covariates tested |
| ☐ | ☒ | A description of any assumptions or corrections, such as tests of normality and adjustment for multiple comparisons |
| ☐ | ☒ | A full description of the statistical parameters including central tendency (e.g. means) or other basic estimates (e.g. regression coefficient) AND variation (e.g. standard deviation) or associated estimates of uncertainty (e.g. confidence intervals) |
| ☐ | ☒ | For null hypothesis testing, the test statistic (e.g. *F*, *t*, *r*) with confidence intervals, effect sizes, degrees of freedom and *P* value noted *Give P values as exact values whenever suitable.* |
| ☒ | ☐ | For Bayesian analysis, information on the choice of priors and Markov chain Monte Carlo settings |
| ☒ | ☐ | For hierarchical and complex designs, identification of the appropriate level for tests and full reporting of outcomes |
| ☒ | ☐ | Estimates of effect sizes (e.g. Cohen's *d*, Pearson's *r*), indicating how they were calculated |

*Our web collection on statistics for biologists contains articles on many of the points above.*

## Software and code

Policy information about availability of computer code

| Data collection | No software was used for data collection |
|---|---|
| Data analysis | Data and code both are shared in a fully reproducible CodeOcean repository (https://doi.org/10.24433/CO.1040081.v1). We used R version v4.3.2 with the packages "ranger v0.16.0" for random forest analysis and iNEXT v3.0.0 and iNEXT.beta3D v1.0.1 for standardization of alpha- and beta-diversity Hill-numbers. Other R packages used are:lme4 v1.1-36, car v.3.1-3, viridis v. 0.6.5, vegan 2.6-8, MuMIn v. 1.48.11, dplyr v2.5.0, doSNOW v1.0.20, progess v1.2.3, visreg v2.7.0, reshape2 v1.4.4, tidyverse v2.0.0. We used Python version 3.11.4 for analysis of data that has been collected in the context of this work and Python version 3.10.9 in a Jupyter notebook on a notebook server of v6.5.2 for analysis of the literature data. Other python-packages: matplotlib v3.7.0, numpy v1.23.5, pandas v1.5.3, uncertainties v3.1.7, Scipy v1.10.0, adjustText v1.2.0. We used the scipy package v1.10.0 in Python for fitting Eq. 2 in the manuscript to the data. All software and packages are listed in detail in the Methods. Methodological details are also provided in the README file. |

For manuscripts utilizing custom algorithms or software that are central to the research but not yet described in published literature, software must be made available to editors and reviewers. We strongly encourage code deposition in a community repository (e.g. GitHub). See the Nature Portfolio guidelines for submitting code & software for further information.

## Data

Policy information about availability of data

All manuscripts must include a data availability statement. This statement should provide the following information, where applicable:

- Accession codes, unique identifiers, or web links for publicly available datasets
- A description of any restrictions on data availability
- For clinical datasets or third party data, please ensure that the statement adheres to our policy

> Data availability:Raw data is available in an online CodeOcean repository (DOI: https://doi.org/10.24433/CO.1040081.v1). Code availability: All code is available in an online CodeOcean repository (DOI: https://doi.org/10.24433/CO.1040081.v1).

## Research involving human participants, their data, or biological material

Policy information about studies with human participants or human data. See also policy information about sex, gender (identity/presentation), and sexual orientation and race, ethnicity and racism.

| | |
|---|---|
| Reporting on sex and gender | N/A |
| Reporting on race, ethnicity, or other socially relevant groupings | N/A |
| Population characteristics | N/A |
| Recruitment | N/A |
| Ethics oversight | N/A |

Note that full information on the approval of the study protocol must also be provided in the manuscript.

# Field-specific reporting

Please select the one below that is the best fit for your research. If you are not sure, read the appropriate sections before making your selection.

☐ Life sciences ☐ Behavioural & social sciences ☒ Ecological, evolutionary & environmental sciences

For a reference copy of the document with all sections, see nature.com/documents/nr-reporting-summary-flat.pdf

# Ecological, evolutionary & environmental sciences study design

All studies must disclose on these points even when the disclosure is negative.

| | |
|---|---|
| Study description | The study are of our collaborative Research Unit "Reassembly" is located in the lowland tropical rainforest within the Ecuadorian Chocó, a biodiversity hotspot that is highly threatened by deforestation. The main aim of this research is to understand the mechanisms, resistance and resilience of a naturally recovering rainforest ecosystem following deforestation.<br>The study design is a chronosequence with a total of 62 plots (50x50m) in cacao plantations and pastures, early and late regeneration (0 - 38 year old secondary forests that were previosly used as cacao plantation or pasture), and mature old-growth forests for reference. The study area (ca. 200 km2) offers unique opportunities unmatched by any other study at a similar scale: a highly resolved chronosequence of spatially independent plots of variable age with clear land-use history, maintained and made accessible to research by a conservation foundation. |
| Research sample | We studied an broad spectrum of animal, plant and bacteria taxa representative for different ecological functions in the forest ecosystem, surveyed im all plots: Ants, Bacteria, Bats, Bees, Dung beetles, Frogs, Frugivorous birds, Ground birds, Leaf-litter arthropods, Mammals, Moths, Nocturnal insects, Saproxylic beetles, Tree seedlings, Trees (Total: 10840 species or morphospecies plus 23590 bacteria sequences). |
| Sampling strategy | The sample size for our analysis of recovery, resistance and resilience was the number of plots of the chronosequence (62 plots in total). Pilot studies on ants and trees (see references) confirmed the feasability of the study design, the lack of spatial autocorrelation for the selected plots as well as the lack of bias by elevation and landscape parameters (see Escobar et al. 2024, cited in the manuscript). |
| Data collection | Each taxon was sampled with specific state-of-the-art sampling techniques by the authors. Sampling methods include traps, sound recorders, wildlife cameras, mist netting, standadized observations, extraction methods of soil, litter or deadwood. Details are provided in the Methods. |
| Timing and spatial scale | Data collection methods were predetermined before the onset of the study, and field work was generally conducted over a long time |

| | |
|---|---|
| Timing and spatial scale | within two years untiul allplots have been sampled. All taxa were recorded simultaneously in this time by different authors. All worked wirthing the same set of plots to allow direct comparability. |
| Data exclusions | No data were excluded. |
| Reproducibility | The data in this study are observational and contain no experimental manipulation (except different forms of land use history, namely pasture and cacao plantation that were considered in the analysis). Methods are completely reproducible based on the detailed description in the paper, and can be comparted to our open data. |
| Randomization | The study design (location of 62 plots) was defined and established prior to the collection of data. Potential spatial biases were controlled for (i.e. variation in elevation and landscape features) and are decribed and analysed in detail in our site description paper (Escobar et al. 2024, cited in the manuscript). Each taxon sampled had a responsible expert principal investigator and a PhD researcher familiar with this taxon; all of them are included as authors. |
| Blinding | Blinding was not possible in our study. Blinding methods are not established for field studies on biodiversity and not relevant for sampling data of species composition in different plots, since there are no subjective judgements involved that may bias the results across sites. |

Did the study involve field work? ☒ Yes ☐ No

## Field work, collection and transport

| | |
|---|---|
| Field conditions | Climatic conditions are typical for moist tropical forests with mean annual temperature of 23°C and mean annual precipitation of 3000–6000 mm. |
| Location | All 62 plots of the study area are located within a large region of ca. 200 km2. The study is located at 0.5°N 79.2°W, the range of elevation is 130–540 m asl. |
| Access & import/export | Sample collection, transport and export permits are regulated with a General Contract (Contrato Marco) for all the subprojects with the Ecuadorian Ministry of Environment via the involved institutions (Universidad de las Américas and Pontificia Universidad Católica del Ecuador in Quito). Export permits are Nagoya compatible and are registred via a Due Diligence Declaration in the European Commission (Project ID 158479). |
| Disturbance | As sampling collections do not target vulnerable species and occur at a small spatial scale, the established survey methods of our study do not represent a significant disturbance to the forest ecosystem nor a threat to regeneration and conservation of this habitat. |

# Reporting for specific materials, systems and methods

We require information from authors about some types of materials, experimental systems and methods used in many studies. Here, indicate whether each material, system or method listed is relevant to your study. If you are not sure if a list item applies to your research, read the appropriate section before selecting a response.

### Materials & experimental systems

| n/a | Involved in the study |
|---|---|
| ☒ | ☐ Antibodies |
| ☒ | ☐ Eukaryotic cell lines |
| ☒ | ☐ Palaeontology and archaeology |
| ☐ | ☒ Animals and other organisms |
| ☒ | ☐ Clinical data |
| ☒ | ☐ Dual use research of concern |
| ☐ | ☒ Plants |

### Methods

| n/a | Involved in the study |
|---|---|
| ☒ | ☐ ChIP-seq |
| ☒ | ☐ Flow cytometry |
| ☒ | ☐ MRI-based neuroimaging |

## Animals and other research organisms

Policy information about studies involving animals; ARRIVE guidelines recommended for reporting animal research, and Sex and Gender in Research

| | |
|---|---|
| Laboratory animals | No laboratory animals were used in the study |
| Wild animals | Two of the taxa involved wild vertebrates that were hand collected: bats from mist nets and frogs from the forest floor.<br>We used mist nets to capture bats. They were handled by experts, removed from the nets, kept in clean cloth bags until they could be examined for identification and measurement (about 30 minutes), and then released at the same sampling site.<br>During plot searches by experts (authors of this study), detected frogs were gently pushed into a plastic tube (no direct handling). In order to avoid potential transmission of diseases, each frog was then immediately transferred and temporarily kept in a separate, |

clean plastic bag, until plot search was finished. Subsequently, each frog was identified (in the field on the respective plot) to species level, based on external morphology (BioWeb; Ron et al., 2024), sexed (based on species- and sex-specific characters, e.g. vocal sacs, nuptial pads, eggs visible through skin etc.) and measured. To avoid potential transmission of diseases or toxic secretions, we used for each frog a new pair of laboratory gloves. After that procedure the frogs were immediately released on the respective plots. The time from capturing a frog to its release, varied from a few minutes to about an hour. All our work complied with the guidelines for amphibians and reptiles in field research, compiled by the American Society of Ichthyologists and Herpetologists (ASIH), The Herpetologists' League (HL) and the Society for the Study of Amphibians and Reptiles (SSAR) (https://ssarherps.org/wp-content/uploads/2014/07/guidelinesherpsresearch2004.pdf). This established treatment (and collection of specimens) was covered by research and collection permits (MAATE-DBI-CM-2021-0187, add further numbers).
References: Ron, S., A. Merino-Viteri, and A. Ortiz. 2024. "BioWeb, Anfibios Del Ecuador. Versión 2024.0." 2024. https://bioweb.bio/portal/Datos/UsoDatos/.
Invertebrates were sampled by taxon-specific established methods and traps and directly killed and preserved in alcohol or freezer, details of sampling methods are described in the paper. Collection of specimen was covered by research and collection permits.

**Reporting on sex**

No sex oder gender-specific information has been collected in this study.

**Field-collected samples**

For taxa that involved collection of samples (invertebrates, bacteria), samples were stored in alcohol or in the freezer in our research station laboratory until they were transfered to Quito or exported to Germany, particularly for DNA analysis and/or for preservation in insect collections hosted by the institutions and museums.

**Ethics oversight**

Ethical approval or guidance was not required for this study. However, all project participants agreed upon rules of procedure how to handle individual responsibilities and rights such as data ownership and publication ethics, or rules to solve conflicts.

Note that full information on the approval of the study protocol must also be provided in the manuscript.

# Dual use research of concern

Policy information about dual use research of concern

## Hazards

Could the accidental, deliberate or reckless misuse of agents or technologies generated in the work, or the application of information presented in the manuscript, pose a threat to:

| No | Yes | |
|---|---|---|
| ☒ | ☐ | Public health |
| ☒ | ☐ | National security |
| ☒ | ☐ | Crops and/or livestock |
| ☒ | ☐ | Ecosystems |
| ☒ | ☐ | Any other significant area |

## Experiments of concern

Does the work involve any of these experiments of concern:

| No | Yes | |
|---|---|---|
| ☒ | ☐ | Demonstrate how to render a vaccine ineffective |
| ☒ | ☐ | Confer resistance to therapeutically useful antibiotics or antiviral agents |
| ☒ | ☐ | Enhance the virulence of a pathogen or render a nonpathogen virulent |
| ☒ | ☐ | Increase transmissibility of a pathogen |
| ☒ | ☐ | Alter the host range of a pathogen |
| ☒ | ☐ | Enable evasion of diagnostic/detection modalities |
| ☒ | ☐ | Enable the weaponization of a biological agent or toxin |
| ☒ | ☐ | Any other potentially harmful combination of experiments and agents |

## Plants

Seed stocks

N/A

Novel plant genotypes

N/A

Authentication

N/A

