## [Peer Review File · Nature]

Biodiversity resilience in a tropical rainforest

Corresponding Author: Professor Nico Bluethgen

Version 0:

Reviewer comments:

Referee #1

(Remarks to the Author)

I am quite enthusiastic about this study based on impressive data sets from one of the few well-established, replicated and sufficiently long rainforest successional chrono-series. The study brings the concept of resilience vs resistance as two components of recovery. These notions are not entirely new, but the quantitative approach demonstrated here with the recovery resolved into these two components is quite insightful and useful. The analysis is comprehensive, including a wide range of taxa (including bacteria) as well as a range of alpha and beta diversity measures (Hill numbers). Further, it is complemented by the literature search and analysis of other existing data – all of them by far less robust than the current analysis. The study is exceptional considering the data sets and the analytical approach, but also the results. I am surprised – but convinced by the study – that generation time (and generally slow vs fast life history) is not a good explanatory factor for the recovery. Further, this study is absolutely right in emphasizing that while species diversity and abundance could recover often quickly, this is not the case for species composition – and emphasized, quite correctly, the importance of species composition. There have been some over-enthusiastic studies previously, even from the same chrono-sequence studied here, focusing on species richness and abundance and arguing on their basis that the rapid recovery demonstrated high biodiversity value of secondary forests. While secondary forests do have their value, this perspective disregarded the identities of species; the current paper provides a very useful corrective.

(Remarks on code availability)

Referee #2

(Remarks to the Author)

The manuscript by Metz et al evaluates for a landscape in the tropical Colombian Chocoo the resistance of 16 taxonomic group to forest conversion to agriculture land use, and their subsequent recovery ('resilience') towards old-growth forest values, and how much time this takes.

For me their main findings are that 1) recovery is slower for composition than for abundance or diversity, 2) recovery is more driven by 'resilience' than by resistance, 3) resistance and 'resilience' are uncoupled, 4) simple attributes can not explain difference in resistance and resilience of taxa, 5) some unexpected groups do not recover (soil bacteria, leaf-litter arthropods, nocturnal insects)

The strengths of the manuscript are 1) the comprehensive description of recovery across many taxonomic groups belonging to different trophic levels, using a standardized plot network and analytical framework which allow for strong generalizations of the results, 2) the interesting analytical framework by decomposing recovery into its two underlying components, resistance and 'resilience' which are associated with different ecological processes (tolerance versus colonization and regrowing capacity), 3) the surprising result that some short-lived taxonomic groups show arrested succession, and that simple ecological attributes can not explain resistance or recovery.

The weaker parts of the manuscript are that 1) it focuses on recovery time of composition, which is difficult to quantify for several taxonomic groups because it takes more than the age of the oldest secondary plots (40 years), 2) two very different land uses were included (cocoa plantations and pastures) but their affect on resistance and 'resilience' are not explicitly taken into account in the model, 3) sometimes unclear description of the study system and the statistics.

Overall, the manuscript was fluently written, interesting, and a pleasure to read, and it provides in my opinion a novel and

important contribution to science and, potentially, to the application of restoration.

Please find my major and minor comments below. I hope they are of use to improve the manuscript.

MAJOR COMMENTS

1. ACCURATE QUANTIFICATION OF RECOVERY TIME? You have opted to focus your story on recovery time of composition (because that recovers most slowly and you find it most important). From Fig 2 it seems (difficult to guesstimate with the logarithmic axes with few tickmarks) that 21 of your 56 comparisons (3x18), i.e., 38%, have a recovery time longer than the oldest secondary forest plot in your dataset (40 years). If you see the greyscale of the 95% confidence interval, then many taxonomic groups have a confidence interval between 10-120 years. The question therefore is how accurately you can estimate recovery time, and how meaningful this is if the confidence intervals are so large? Hence, if the strength of your data is not there, then put less emphasis on recovery time, and more on resistance and 'resilience', that you probably can estimate more accurately.

2. UNCLEAR VALUES AND COMPARISON OF RECOVERY TIME. You say that abundance and diversity recover faster than species composition (1180-182). For abundance and diversity you give such a wide range (0-100 years) and no median, and for species composition you report no values at all, so that it is not informative anymore. If you want to put this in an ecological or restoration perspective then it is important to know the real values. In Fig 2 you put the values on a log₁₀ scale, and it is really difficult to infer the real values. So please include a table of the 18 groups with the resistance, resilience, recovery time and their corresponding confidence intervals. Fig 2 does not confer much, apart that it varies. Please include in an appendix a figure with 18 groups of 3 bars (one for abundance, one for species diversity, one for composition) (mean and confidence interval), so that it is easier to quantitatively compare groups and recovery time of different attributes

3. INCLUDE THE 2 LAND USE TYPES IN THE FORMAL ANALYSIS. You have two types of land use (cocoa and grasslands) that differ totally in openness, structure, growth form composition, management intensity, and hence are likely to differ in resistance, resilience, and recovery time. You say somewhere that the estimates for resistance did not differ between the two land use types, but I did only see the test statistic and not a graph with the data, and given that it is a confounding factor I think it is better if you include it formally in your statistical analysis (i.e., by allowing the two land use types have different intercepts and slopes). Color code your dots in Fig 3 based on land use type, and show two different lines if they have different intercepts or slopes

4. IMPROVE DESCRIPTION OF YOUR STUDY SYSTEM. Explain better your study system upfront in the main text and main methods, as this matters how representative your site and results are, and what resistance and recovery you can expect. E.g., rainfall, forest type, and soil fertility determine the maximum pace and asymptote of recovery, and the types of plants and animals you are likely to have. Describe the landscape context (how much forest nearby) as this determines the source population and dispersal ability and your 'resilience'. Describe for both land use systems the typical area and duration of use, and management practices, as this previous land use intensity strongly shapes resistance and 'resilience'. E.g., for the pasture in addition exotic grasses, stocking density, and fire use, and for cocoa plantations if this was open grown or shade grown with remnant shade trees, pesticides, etc. Describe the heterogeneity in your landscape (i.e., variation in steepness of slopes, altitude, geology and soil type) as this determines your beta diversity and similarity that one can expect.

5. HOW REPRESENTATIVE ARE OLD-GROWTH FOREST? I was highly surprised that some groups did not recover (soil bacteria, leaf-litter arthropods, nocturnal insects). Normally bacteria are not very responsive to secondary succession or recover soon (maybe because there is a mismatch between the micrometer scale that is relevant for bacteria and the plot scale that is relevant for plants, or because the fast generation time of bacteria). I had expected leaf-litter arthropods not very sensitive because litter production and litter thickness often recover very soon in succession (after ca 10 years?), and as it is brown litter I expect less specificity). And I expected nocturnal insects to be less affected by the surrounding landscape (as they forage at night when it is cooler and more humid) and fly and recover faster. Hence, I wondered how representative are your old-growth forest (OGF). Sometimes they are much further away (as people have secondary forests (SF) closer to their homesteads) than secondary forests. Please provide the average distance and range of SF to OGF plots and amongst SF plots and between land use and OGF plots. Similarly, people often do not use OGF if they are on too steep slopes, or on an unproductive soil type or geological formation. This may explain why you have totally different bacterial communities (or other groups) there.

6. IMPROVE RESILIENCE TERMINOLOGY. Please stay in line with mainstream definitions of resilience. You cite Holling for your definition of resilience. Holling (1973, p. 17) defined ecological resilience as 'resilience determines the persistence of relationships within a system and is a measure of the ability of these systems to absorb changes of state variables, driving variables, and parameters, and still persist. Hence, I think resilience comes most close to your recovery time (with a low recovery time being a high resilience. In tree-ring studies people normally analyze growth resilience (for example to drought) as the product of resistance and recovery. Hence, what you call 'resilience' is in fact 'recovery' for many ecologists.

7. UNDERESTIMATION OF RECOVERY RATES; USE REAL SLOPE. As nearly all measures of recovery of composition show a saturating relationship with time, this means that if you calculate 'resilience' as $(1 - \text{resistance}) / \text{recovery time}$ (L571) you underestimate the 'resilience' rate (i.e., recovery rate) over nearly the whole time interval (see extended data fig 6). Hence, I think it is better to use the real slope. Moreover, given that your recovery times are difficult to estimate (see above), your recovery rate is neither so precise.

8. JACKKNIFE CONFIDENCE INTERVALS. L602-610. I am not familiar with the Jackknife method, but if you leave 1 out of 62 plots out, then the resistance and resilience estimates will change very little. This may give erroneously the feeling that the confidence intervals are very narrow while they are not. Can you calculate a more informative confidence interval, directly

based on the data?

9. JUSTIFY ALL YOUR CHOICES IN THE METHODS. The methods read now as a cookbook. I can repeat what you have done, but for many choices I do not understand why you did it. Please justify, so that the reader can understand and judge your choices

MINOR COMMENTS

Fig 3. Interesting graph! Please let the y axis of ALL graphs start at zero, and put more tickmarks so that it is easier to read the values. Write next to the icons the taxonomic group, make the median value line of OGF and land use systems thicker, provide different systems for the two land use systems, and show them with separate recovery lines if they differ in their slopes and/or intercepts Put similar graphs for abundance and diversity in the appendix.

Your life histories, dispersal modes, and trophic levels are rather coarse. So how informative is it to test how they are related to resistance/resilience, and recovery time? Please tone down a bit your conclusions, and say that this is a first analysis/approximation

LOW SIMILARITY HAMPERS RECOVERY TIME ESTIMATES? If I guesstimate median pairwise similarity amongst OGF plots for your 18 taxa than I get a mean of 0.32 (range 0.12-0.50). This is quite low, perhaps because of a high spatial or temporal heterogeneity, combined with small plot sizes and small monitoring time windows for some taxonomic groups. Then you try to estimate recovery to such a low value, and I wonder how accurate that is.

Line (l) 95. 'Stability' comes out of the blue. Do you need it, given that land use conversion of forests to pasture is by definition not stable.

L97. I think 'recovery' indicates the speed of recovery

L100 'Mobile animal communities .. recovered faster than trees ... suggesting that these functional groups drive the recovery of tree communities'. The alternative explanation is that animals and trees recover independently. I think to support your claim you should do a network analysis between recovery of animal species/groups and plant species/groups

L104 replace 'rules' by 'mechanisms'?

L118-120. This feels a bit like a strawmen's argument. You found 32 studies that analyzed recovery of other taxa, and there are several meta analysis and multi taxon studies. Although they have not measured trees alongside they can for sure inform how different taxa recover. Please indicate the state of the art and summarize briefly what they found

Dent, D.H. and Wright, S.J., 2009. The future of tropical species in secondary forests: a quantitative review. *Biological conservation*, 142(12), pp.2833-2843.

Díaz-Vallejo, E.J., Seeley, M., Smith, A.P. and Marín-Spiotta, E., 2021. A meta-analysis of tropical land-use change effects on the soil microbiome: Emerging patterns and knowledge gaps. *Biotropica*, 53(3), pp.738-752.

Oliveira, P. S., Falcão, L. A., Almeida, J. S., Fernandes, G. W., Reis Júnior, R., Nunes, Y. R., ... & do Espírito Santo, M. M. (2024). Diversity patterns along ecological succession in tropical dry forests: a multi-taxonomic approach. *Oikos*, 2024(4), e09653.

L124-125 'Uncertainty about the time scales needed for natural recovery .. hampers effective decisions to achieve restoration goals'. Explain here or in the discussion how, because that is not so clear.

L134. Not only plasticity, but also protection against change (i.e., the ability to persist belowground as rhizomes or seeds)

L137 SMALL-SCALE disturbances (add small scale)

L138. I would say 'are an inherent feature of the system'

L143. Add a quantitative justification: 'successful tropical forest recovery AS MOST ARE INSECT- OR ANIMAL POLLINATED (..%) OR DISPERSED (..)'

L165. Write ASV in full

L168 define what you mean with 'dispersal ability'. To cover a large distance? in space? Time?

L168-169. Better describe your pasture and cocoa plantation system here (see my main comments)

L171. Rewrite. This suggests that you monitor your taxa over time, but you use a chronosequence approach. Write in the methods the assumptions of the chronosequence approach (all plots started from the same situation and followed the same trajectory) and why you think this assumption is met

L175-178. Make explicit how info on recovery times is needed for conservation and policy development

L181. Provide median, min and max, recovery time for abundance, diversity, and composition

L182. Why a signed rank test and not a paired t-test?

L188. Please add: 'OR COMPLEX MEASURES MAY OVERESTIMATE SIMPLE ATTRIBUTE RECOVERY' Recovery can happen in terms of multiple attributes, of which some go slow, others go fast. It is unclear why you focus on the slowest recovering attribute (only when that is recovered it is 'good'?). Please justify and explain better. To me that feels like a quite reductionist and not a holistic view on recovery. I think it is important and informative that abundance and diversity recover fast, as they are also important ecosystem components. Focusing on the slowest component suggests that everything goes slow and little is gained, which is not true.

L193. 'in combination with slow recovery of tree biomass'. Litter production rates recover really very fast (<10 years?), and that matters for litter dwelling insects, I would say.

L193 Slow litter arthropod recovery because of slow recovery in tree composition. Can you give a reference how specific litter arthropods are for species-specific tree litter? I would expect that with brown litter insect-plant relationships are less specific than with green leaves

L193. See above. My guess is that bacteria do not recover because the SF are somewhere else than OGF or under different environmental conditions

L201 'Once these mutualists and other groups recover.. they can support succession towards old-growth tree communities'. Do you have some literature references for that? I know that that is what most of us biologists believe or hope, but I guess there is a lot of redundancy in those relationships, and that different frugivore species can take over the seed dispersing roles of others. See Dent et al who calculated redundancy in seed dispersal mechanisms and showed that it increased during succession.

Estrada-Villegas, S., Stevenson, P. R., López, O., DeWalt, S. J., Comita, L. S., & Dent, D. H. (2023). Animal seed dispersal

recovery during passive restoration in a forested landscape. *Philosophical Transactions of the Royal Society B*, 378(1867), 20210076.

L208. Many trees also recover slowly simply because they occur in low abundances and dispersal is a chance event. Please mention this as well

L210. Why do moths, ground birds, and ground mammals recover on a similar time scale as trees. Why? Because they depend on the trees as a fruit source (I am not sure how specific this is, as cassowaries eat not only large seeded species but also opportunistically any plant with fruits)? Or because they are hunted out and sensitive to fragmentation?

L214. Surprising that deadwood beetles is slowed down by deadwood volume, as there is a lot of thinning and mortality of short-lived pioneer species going on during succession. Maybe you mean that they need specific tree species or thick stem diameters?

L215. Surprising that land use did not affect resistance or 'resilience'. Did you explicitly model it?

L216., I can not see this in Supplementary Table 2

L254-257. Please also say that bats and birds may spill over from the surrounding agricultural landscape to the OGF,

L266-281. If bacteria have a resistance of 82% to 71% (why is the resistance lower at deeper soil layers, by the way), which means 71-82% similarity with OGF, then I would not call that an 'alternative stable state'. This is definitely a case where the glass is more than half full! ;)

L270. Surprising that resilience and resistance are not correlated across taxa! Please come up with a reason why this would be the case

L314-328. This discussion could be much stronger. Please try, as your results are so cool

L330-347. Your conclusions should be way more stronger for Nature. Please try

L335-337. Unclear

338-342 I do not understand

L344. Replace biodiversity by composition

L345-347. This is a bit thin, and strange to end your conclusions with it, as you did not test for the role of corridors, remnant trees 'or other elements' (what do you mean?)

L496. What is the Shannon diversity. Is this the traditional Shannon diversity or the Hill 1 diversity (i.e., effective number of species)

L499. Why did you have to transform to relative abundances?

L508. What is the average coverage in your dataset (mean and range)? It feels more accurate if you use that one rather than the minimum coverage of the extrapolated data to twice the plot size. Please explain and justify

L498. Can you explain exactly what it is. I thought it was Bray-Curtis DISsimilarity???

L512. Twice the size of what? Species? Number of individuals? Unclear why you do this. Please justify

L516-520. I do not understand. Please provide for poor people like me a definition of beta diversity. How can you extrapolate beta diversity if that is calculated as similarity in species composition?

L535-539. Please test and report the differences between pasture and coca in resistance and recovery for all your taxonomic groups and your 3 attributes.

L545 Say that this is, in fact, a measure of similarity to OGF values. Explain above why values that overshoot OGF values (more biodiversity than in OGF) are not 'good'.

L548. Say that this similarity index is not completely symmetrical around 1

L555. Why can't you use an OGF value of zero? Please explain

L566-567. I do not understand

L580. Why do you use SD rather than 95% confidence interval?

L575. Apparently it is recovery time until 0% of OGF values. Please say so explicitly in your abstract and main text, and justify why

L591. Please show in the appendix the pairwise scatterplots for these 3 relationships

594. See above; it does not make sense to report an average rate as recovery changes non-linearly over time

L616. Why do you use random forest rather than a simple multiple regression or a variance partitioning analysis?

L618. Explain why you use impurity

L619-621. Drawing random value for resilience if the resistance was greater than 90% does not make sense to me. I think it means you simply can not estimate it, and I would omit this value

L645 provide a name for group 2

L648 provide a name for group 4

Fig 1. Nice illustrative figure!

Fig 2. Nice illustration, although it is difficult to compare the values and assess the range of the confidence interval (see my comment above). At least put more tickmarks on the x axis, so that it is easier to read. What are blue/orange/green dots/

Fig 4. Why is there a decline in the grey zone, and not an instantaneous decline? Why do your red line stay after the vertical dotted line for some time flat whereas in your Fig 3 recovery starts instantaneously? Please use normal scale for resistance, and not a square root scale. Aldo for resilience. 95% CI of the mean? What are red, blue, green dots?

(Remarks on code availability)

Referee #3

(Remarks to the Author)

This manuscript aims to evaluate recovery times of different taxa after disturbance along a successional gradient using chronosequence data. This is an important question, as understanding how different groups of organisms recover and colonize a tropical forests after land use may bring key insights for landscape management. Additionally, the data set is impressive and reflects a substantial amount of field and lab work. I appreciate the effort of integrating multi-taxa datasets

and combining different statistical methods to address this issue.

However, I have many reservations about the manuscript. Mainly, the concepts mentioned are not explored in sufficient depth, and the authors do not appear to be fully familiar with some of the topics discussed. Below, I provide some general comments.

1. The concept of resilience.

I was surprised by the definition of the term resilience provided here, and the way recovery, resilience and resistance were linked. There is a large body of literature highlighting that resilience has been a concept difficult to grasp, yet many papers provide clear definitions. This is the first time I see recovery defined as the combination of resilience and resistance.

I checked the citations provided in line 130 and none of them defines recovery as the combination of resilience and resistance. In the Holling book, as in many other publications by SR Carpenter (e.g. Ives & Carpenter, *Science* 2007; Peterson et al., *Ecosystems*, 1998), resilience is the most overarching concept from which other related concepts, such as recovery and resistance, are derived. In Poorter et al. 2021 (*Science*) resilience is defined as “ability of a system to absorb disturbances and return to its previous state” and encompasses both recovery and resistance. I also recommend reading Rodrigo Muñoz et al. *J Ecol.* 2021

In this sense, the manuscript is very confusing. Indeed I felt that the term recovery was somewhat polysemic, making the manuscript very difficult to follow. For instance, lines 128-130 mention that recovery can be partitioned into resistance and resilience, but Fig 1 is titled “Two dimensions of stability”, meaning resistance and resilience, which suggests that stability is synonym of recovery. In panel b of Figure 1, there is an equation given to define recovery time, but in Figure 3, one can see the “recovery trajectories” of species composition and this does not seem to be calculated based on the equation in Fig 1, so that recovery trajectories is different from recovery times.

2. Questions and hypotheses

The way the results are presented is somewhat descriptive and I felt the lack of a theoretical framework integrating the questions addressed. Specifically, there is not a conceptual basis to evaluate variation in recovery times across taxa exhibiting different life-history strategies, dispersal modes and trophic levels. What do we know about this? There is no mention in the introduction of any previous work addressing these issues.

In particular, the question “whether differences in recovery times (...) can be explained by simple rules” is weak. The authors expect taxa with the r strategy to recover faster than those with the K strategy based on the fact that r-species exhibit higher reproductive rates. However, there is no explanation on why having lower age at first reproduction may allow species (or taxa) to show a faster recovery (or resilience or resistance). Although the framework of r-K selection continuum has been important to globally understand life history strategies, it is clearly an oversimplification of nature and many species do not fall along this single dimension of variation. Only in tree ecology, recent work has shown that there are many axes of variation in demographic rates, mainly the growth-survival and the stature recruitment trade-offs (Rüger et al, 2020, *Science* 368), and there are other important trade-offs in successional theory such as the colonization-competition trade-off. All these do not necessarily align with the fast-slow continuum mentioned.

The classification of taxa into life-history strategies and dispersal modes was too broad. Within each realm (bacteria, invertebrates, vertebrates and trees) there are species exhibiting a wide array of life-histories so assigning all bacteria the strategy r and all trees the K strategy presents an overly simplistic view of nature. Also, assigning an aerial dispersal to all plants, stating that all plant species are virtually dispersed by birds, bats and wind is also an oversimplification. There is a wide variation in plant dispersal abilities, and my guess is that the same happens for each taxa. These categories are too large and thereby inconsequential. Given such reductionism, I'm not surprised that there are no significant results among categories in the time of recovery.

Overall, I had the sensation throughout the manuscript that the complexity of ecological process is somewhat disregarded and that this study does not sufficiently incorporate previous and more recent research.

3. Successional data

There are many issues that are not clear in the data and the methods used.

First, there is not enough information on the plot data. Of the 62 plots, how many are active cacao plantations, pastures (6 and 6)? Which are the stand ages of the secondary plots? Many studies have addressed the shortcomings of chronosequence data and the importance to have replication within each age class. There is no information on the number of plots for each class age, or on the size of the plots, which also matters a lot.

Also, there is no consideration of the importance of prior land use, see Robin Chazdon and Catarina Jakovac papers on this (eg. Chazdon 2003, *Pers. Plant Ecol. Evol*, Chazdon et al. 2009, *Phil. Trans Royal Society. B*; Jakovac et al. 2021, *Biol. Rev.*). Norden et al. 2015 (*PNAS*) also shows how variable are successional trajectories are based on dynamic data. Figure 3 seems to combine everything, so that you are probably comparing apples and oranges. Drawing conclusions from this

seems unreliable to me.

4. Results presentation

There is not enough description of the analyses performed in the main text. The main text should be self-contained, that is, it shouldn't be necessary to go to the Methods section to understand the results. Mainly, there is a lot of information missing (see point 3) and the figures do not seem to be supported on data. I was particularly puzzled by Figure 2 which is very pretty, yet does not show the data underlying these final results, and the reader needs to go over all the Methods section to understand which data and analyses support this finding.

Also, there are large sections describing results with any reference to a statistical analyses (eg. lines 196-214).

5. Impact and generalization

Although I do recognize that this study relies on an extraordinary amount of data, I don't think that the recovery times reported here can be generalized. Overall, these findings are likely to be anecdotal for this study site. Moreover, as the authors mention, these data were collected in a relatively well conserved landscape, which is not representative of what is happening in most the tropics. The study of Rozendaal et al., for example, was based on chronosequence data from 56 sites (over 1500 plots) distributed in 10 countries and land use was controlled for in each chronosequence dataset. Additionally, based on my understanding of the manuscript, the estimates of recovery times, resistance and resilience were derived from calculations that violate the basic assumptions of chronosequence as they do not distinguish land use (not only separating active cacao plantations from pastures, but also distinguishing successional forests depending upon their previous land use).

Overall, these shortcomings considerably limit the reliability and impact of the results presented in the manuscript.

6. Statistical methods

The authors made a huge effort to write a M&M section as clear as possible, and provide examples that illustrate the different challenges then faced when dealing with such big and heterogenous datasets. I am not up to date in quantitative analyses, so I'm not able to provide a through revision of the statistical methods used. There are however, a few points that were not clear to me:

Abundance refers to the total number of individuals (or sequences) for an entire taxa? How is this number comparable across taxa?

Even if you control by sampling coverage using Chao's methods, I don't think you can extrapolate metrics related with species composition to larger plot sizes (diversity, species composition, beta diversity; line 519). Only metrics such as stem density or biomass increase in a linear trend.

It seems that the authors found many issues with the data and had to adjust using alternative procedures to deal with these cases. How do you think these artifacts affect the analyses?

What is the intrinsic recovery rate (line 591)? How is this different from the recovery times? If this metric is so correlated with resilience, this makes confusing to understand the way you deal with the different terms and concepts.

The section describing the impurity analyses needs clarification (lines 614-626).

7. Literature analysis

The literature analysis comes unexpectedly and is not integrated in the introduction. This makes the motivation of this section unclear to the reader. Also, since studies evaluating recovery from a multi-taxa perspective are so scarce, I wonder why the timeframe of the analysis was so short? For instance, I can think of an important paper of Daisy Dent and J.. Wright (Biol. Conservation 2009) that looks at recovery in different groups, but that is not in TableS5. By looking at this table I was confused by the fact that the search was for the years 2016-2023, yet there are many papers going back to 1990.

8. Restoration

Restoration is mentioned a lot in the Introduction and I think it's used as a hook to provide some relevance to the study, but without support. How exactly do the results found help to restoration strategies? Honestly, this is not clear to me. I recommend the reading of Brancalion & Hall (see Brancalion & Hall, *J Applied Ecological*, 2020 and Hall & Brancalion *Science* 2020) as an overview of the major challenges of restoration from a practical perspective.

Also, there are some statements that are made without sufficient consideration. In particular, I disagree with the sentence "only large-scale restoration of secondary forests can help to achieve global biodiversity conservation and climate change mitigation goals" (l115). As a matter of fact, secondary forests should not need restoration, unless they are in a arrested state. What about abandoned lands? Moreover, and maybe more importantly, this sentence can give a misleading

message: if restoration is the only way to achieve global conservation goals, that means that conserving old-growth forests is worthless.

Please be careful with these kind of statements in the future.

I hope these comments will be helpful during the revision process. I believe that the datasets used and the analyses performed can be the basis for a complete and thorough study evaluating recovery after land use from a multi-taxa perspective. However, I do not think the scope of this study and the limited generalizability of its results are novel enough for a high profile journal like Nature.

(Remarks on code availability)

Version 1:

Reviewer comments:

Referee #1

(Remarks to the Author)

This is a review of the manuscript revision. The revised manuscript is an improvement on the original version. I have found the revision thorough and addressing the concerns of all three reviewers well. I have no further questions/concerns with this manuscript version, after my already positive review of the original manuscript.

(Remarks on code availability)

Referee #2

(Remarks to the Author)

The manuscript by Metz et al evaluates for a landscape in the tropical Colombian Chocoo the resistance of 16 taxonomic group to forest conversion to agriculture land use, and their subsequent recovery ('resilience') towards old-growth forest values, and how much time this takes.

The strengths of the manuscript are 1) the comprehensive description of recovery across many taxonomic groups belonging to different trophic levels, using a standardized plot network and analytical framework with a massive dataset which allow for strong generalizations of the results, 2) the interesting analytical framework by decomposing recovery into its two underlying components, resistance and recovery rate which are associated with different ecological processes (tolerance versus colonization and regrowing capacity), 3) the surprising result that some short-lived taxonomic groups show arrested succession, and that simple ecological attributes can not explain resistance or recovery, 4) its important implications that natural regrowth can be used as a low-cost nature based solution, to scale up forest restoration, and provide tangible outcomes for the conservation of multiple taxa.

This is the second time I review the manuscript. I thank the authors for their openness and willingness to revise their manuscript based on the comments and their thorough revision of the manuscript and replies. I think the manuscript has substantially improved because 1) it now also present data on recovery after 30 years, which is more solid and more policy relevant, 2) the legacy effect of the two previous land uses are now explained, and included in all graphs, 3) the terminology on resistance, return rate and predicted recovery time has been clearly explained and linked to how it is used in the literature, 4) the methods are better explained, justified, the manuscript is now more hypothesis driven, and the main patterns are well explained.

I agree with all replies to my comments and the changes made, and have only two minor remarks. I also have read the comments and replies to reviewer 3, and I think the authors satisfactory addressed the reviewers comments. I disagree with reviewer 3 that the manuscript is not novel, for the four reasons I mentioned above. It is true that the study is carried out at a single site, but the authors made convincingly the point that it is a representative site for large parts in the tropics, and I think the manuscript significantly advances the field because of its broadness and depth by studying succession in a thorough way for so many taxa.

Overall, the manuscript was fluently written, interesting, and a pleasure to read, and it provides in my opinion a novel and important contribution to science and, potentially, to the application of restoration. It is the kind of manuscript I hope and expect to see published in a high impact, high quality journal like Nature.

I congratulate the authors with the result. Please find my last very minor comments below.

MINOR COMMENTS

RECOVERY TIME. I am fine with your reply to my comments, but please mention explicitly in the methods or the results the

caveat that for 21 out of the 56 comparisons the predicted recovery time is longer than the oldest secondary forest plot in the dataset (38 years). This is mitigated to some extent by including a large number of old-growth forest plots as an asymptotic reference, but this leads to a larger uncertainty in the estimates of predicted recovery time larger than 40 years.

PREVIOUS LAND USE. The description of the two land use systems (pasture, cocoa plantation) in the extended methods is now better. Add to the main methods two lines how these land use systems look like (only cocoa, 5 m tall, open pastures, with occasional palms or remnant trees), as it makes it clear to the reader what kind of system it is and how intensive it is. Add to the extended methods for pasture what kind of grass it is? Native/Exotic? What species. And it would be great if you could add 2-3 lines about the size of these patches (Maybe you did it already) and for how long it was ca. used.

Main text L198. You now emphasize that when you know recovery of one component (slow recovery in spp composition) does not inform of recovery of another (fast recovery in abundance and richness). This is fine, but I think it is more informative that some attributes recover in short time, but if you want recovery of typical old-growth forest species, that it takes a long time

Your rebuttal, line 947-965. Include a bit more of this in the extended methods, as with what you now say in the methods (quoted in line 966-970) I could not fully understand and repeat what you have done.

Main text L97. In your abstract, it is unclear if the 100 years of trees and seedlings refer to all 3 components, or just to species composition

Main text L125. This is still not totally clear. Is it decreased uncertainty about time scales that fasters adoption of natural regeneration, or a better knowledge how quickly different taxonomic groups return that fosters the adoption?

L383. Replace 'Timber management plans' by 'forest management plans', or something along those lines

(Remarks on code availability)

Referee #4

(Remarks to the Author)

The study reported in this manuscript makes a relevant contribution to tropical forest taxonomic diversity conservation in an era of accelerated transformation of old-growth forests into different agroecosystems. The study is particularly valuable due to its comprehensiveness, as it provides estimations of recovery times upon disturbance cessation for organisms representing multiple taxa across three kingdoms. Similar holistic efforts to assess the recovery of biological diversity are scanty and urgently needed if we are to soundly assess the possibility of recovering fully functional ecosystems in the future. The study shows that the recovery of three diversity metrics, namely, abundance, diversity (mostly species richness), and composition, exhibits enormous variation among taxa and diversity metrics. Also, it shows the differential effects of different land uses on diversity resilience, although it falls short of finding a relationship between this variation and the broad range of life-histories and ecological strategies represented among the groups examined.

Despite the originality and significance of this study, there are a few issues that still require attention from the authors, especially considering that this paper is meant for such a high-profile journal as Nature, with the expected consequences that this potential publication has for advancing the theoretical and conceptual framework of Biological Conservation Science and Ecology in general. These issues are mainly of a conceptual or theoretical nature, but there is also one comment related to the modeling of diversity return rates, which is an important component of the analysis.

1. Ambiguous standing about the concept of resilience.

This issue was raised by Reviewer 3, but in my view, it was not satisfactorily addressed. The first indication of a potential drawback with this issue appears right in the title of the manuscript. When I first read it, I was intrigued by the two main terms it contains (resistance and resilience), presented in a way that implies that these are different but equally ranking properties of a system. Undoubtedly, the importance of resilience as a key ecological concept has increased tremendously in the face of ecosystem degradation and biodiversity losses occurring on the planet. However, it is important to understand that this conceptual framework has undergone a rapid and interesting development since Holling's (1973) seminal paper, particularly in a direction that makes this property easier to measure, rather than an abstract characteristic based on unquantifiable properties like the "potential to change". We must recall that in his 1973 paper, Holling discussed ecosystem resilience as a "measure of its capacity to absorb changes and continue existing". Along with this concept, he defined stability as the ability of a system to return to the equilibrium state after being temporarily perturbed. Formally, he defined resilience as the size of the stability domain (i.e., the stability basin), or the 'amount' of disturbance a system can tolerate before shifting into an alternative configuration. More than two decades later, Holling (1986) attempted to rid these concepts from their evident subjectivity by conceptually distinguishing between what he called "engineering resilience" (a system's capacity -not necessarily its speed- to return to the stationary state after a disturbance) and "ecological resilience" (the magnitude of the disturbance a system can absorb before shifting from one state to another). The integrated understanding of these two properties led Capdevilla et al. (2021) to think of resilience as a system's ability to face change and cope with it.

Despite these efforts, the measurement of resilience remained loaded with subjectivity for almost two decades, as it required

assessing properties as difficult to grasp (and measure) as the system's "potential energy". In the opinion of many scholars working on this topic and interested in the proper assessment of resilience, a major step forward was the conceptualization of resilience as a complex property that thoroughly describes the entire trajectory of one or more state variables of a system since the moment it is affected by a disturbance until its partial or full recovery after the disturbance ceases.

Along this line of thought, Hogdson and colleagues (2015) proposed the measurement of rates of change in ecosystem properties as a proxy of its "potential energy" and put forward a novel conceptualization of resilience as a bivariate property that can be decomposed into two components, namely 'resistance' and 'recovery'. This conceptualization was taken on by Ingrisch & Bahn (2018), who defined resilience as the capacity of a system to maintain its state and recover from disturbances and highlighted its definition as a complex concept encompassing two components (resistance and recovery). Therefore, I am intrigued by the fact that Fig. 1a in this manuscript virtually reproduces the illustration of Ingrisch and Bahn's idea and, at the same time, it seems to disregard it (the figure legend reads "the return rate towards the pre-disturbance reference state is used to quantify resilience". Ingrisch and Bahn underscore the importance of normalizing both resistance and recovery time to achieve more objective and sounder comparisons among ecosystems. In addition to Muñoz et al.'s (2021) paper mentioned by Reviewer 3, I would like to suggest van der Sande et al.'s (2023) paper on soil resistance and recovery in tropical ecosystems as an example of this currently mainstream approach.

Given the current development of the resilience concept, I would like to invite the authors to adopt this mainstream framework, which is virtually the same one they used; the biggest change required in this regard would be in the title of the paper, for which the following phrase would be appropriate: "Diversity resilience in a tropical rain forest".

References:

Capdevila, P et al. (2021). Reconciling resilience across ecological systems, species and subdisciplines. *J Ecol.*, 109(9), 3102-3113.

Holling CS (1973). Resilience and stability of ecological systems. *Annu. Rev. Ecol. Syst.*, 4, pp. 1-23.

Holling CS (1996). Engineering resilience versus ecological resilience. In: National Academy of Engineering. *Engineering Within Ecological Constraints*. Washington, DC: The National Academies Press.

Hogdson D et al. (2015). What do you mean, 'resilient'? *Trends Ecol. Evol.* 30: 503-506.

Ingrisch, J & Bahn, M (2018). Towards a comparable quantification of resilience. *Trends Ecol. Evol.*, 33(4), 251-259.

Muñoz, R et al. (2021). Autogenic regulation and resilience in tropical dry forest. *J. Ecol.*, 109(9), 3295-3307.

van der Sande, MT et al. (2023). Soil resistance and recovery during neotropical forest succession. *Philosophical Transactions of the Royal Society B, Biological Sciences*, 378(1867): 20210074.

2. Confusion of the concepts of density-dependent r- and K-selection vs. r- and K-strategies.

In searching for evidence of the effect of life history strategies and trophic levels on the observed recovery times for the different groups examined, the authors classified their study organisms as "r- vs. K-strategists" and cite MacArthur and Wilson's (1967) seminal book 'The Theory of Island Biogeography' as the source of this dichotomy, but this is not entirely correct. MacArthur and Wilson developed the notion of density-dependent natural selection with two extremes occurring under highly contrasting demographic conditions (very low or very high population densities). Within this framework, frogs are not necessarily "r-strategists"; for any population of any species, r-selection takes place when population densities are low, so that highly reproductive genotypes have higher fitness than genotypes investing more in parental care but reproducing less. The opposite is true when the same population attains a high density, under which circumstance those genotypes that invest more in parental care have higher fitness than those investing in larger offspring, many of which will not achieve adulthood. If the authors prefer to classify species by strategy not by selection type, they should cite Eric Pianka's (1970) work, as he is responsible for the unfortunate but widely used misinterpretation of MacArthur and Wilson's density-dependent selection model as ecological strategies. In this regard, I invite the authors to review these key references, particularly the first one (Boyce, 1984):

Boyce, MS (1984). Restitution of r-and K-selection as a model of density-dependent natural selection. *Annu. Rev Ecol Syst*, 15, 427-447.

Reznick, D et al. (2002). r-and K-selection revisited: the role of population regulation in life-history evolution. *Ecology*, 83(6), 1509-1520.

Engen, S & Sæther, BE (2017). r-and K-selection in fluctuating populations is determined by the evolutionary trade-off between two fitness measures: Growth rate and lifetime reproductive success. *Evolution*, 71(1), 167-173.

MacArthur, RH & Wilson EO (2001). *The Theory of Island Biogeography* (Vol. 1). Princeton University Press, Princeton.

Pianka, E. R. (1970). On r-and K-selection. *Am. Nat.*, 104(940), 592-597.

3. Statistical analyses and interpretation

The statistical tests used in this manuscript to analyze the huge data set are appropriate. 95% confidence intervals of recovery times were correctly calculated through bootstrapping.

Recovery time and return rate are two metrics to measure recovery estimated by the authors by fitting a negative exponential function (eq. 2) to empirical data. Figure 1a shows this function fitted to a particular set of data (species composition of bees). In this figure, the model presents a good fit to the data; however, the results show that this is not a general trend. The R^2 column in Supplementary_Table_1.csv and Supplementary_Table_3.csv show a huge variation in this goodness-of-fit metric, ranging from -Infinite (i.e., nearly 0 or no variance explained at all) to 0.826, with a mean of 0.105 (excluding the -Infinities) and an extremely low median of 0.066. Among all R^2 values, only 10% are > 0.5 , while 25% of them are > 0.25 . These results imply that, for most cases, a negative exponential function was not a good model to describe the empirical data. Since recovery time and return rate depend on this fit, two possibilities should be considered: either (1) the results on the topic of recovery should be limited to those cases where the negative exponential function was a good representation of the pattern followed by data (i.e., those cases where R^2 was relatively high, preferably > 0.5), or (2) the authors could explore other models describing recovery to consider alternative recovery patterns and then perform model selection, this hopefully leading to higher R^2 values.

4. Visualization

Figure 2 provides an attractive summary of this study's main results. It succeeds in visually conveying the complexity of the analyses, given the number of biological groups involved and the differences among the three diversity metrics examined. Although the horizontal axes of these graphs have logarithmic scales, it is unfortunate that the scale of grays depicting recovery times for the different groups has an arithmetic scale. This restricts the possibility of visually discriminating more finely among taxa with contrasting recovery times during the first three decades of recovery, which are critical given current patterns of land use and known maximum recovery times for tropical rainforests, as convincingly argued by the authors. Therefore, it would be very useful to also use a logarithmic color scale to make differences in recovery times more visible during the first three decades of recovery, while putting less emphasis on these differences for the remaining time.

5. Minor issues.

L 109. There is a small typo here (irreplaceable).

L 134. This line exemplifies the inconsistent or dated use of the resilience concept and the disregard for mainstream current conceptualization.

L 161-162. To match the sequence on the system's trajectory from disturbance to recovery, resistance should go first on this list of resilience components.

L 220-222. I would have liked to see a broader discussion about the potential consequences of the differences in recovery time of species composition between long-lived trees and many animal groups, considering that biological conservation should not only focus on species assemblages but also on the multiple cross-trophic level interactions that should be preserved.

L 262. For the same reason mentioned above regarding the system's trajectory since the disturbance acts upon it (the perturbation period) to the potential end of the recovery, the order of the columns in this table seems odd. However, this could be easily improved by moving to the right end of the table the column with the predicted recovery time to 90% of old-growth forest. Also, on this table, does the asterisk in the title of the return rate column denote a multiplication?

L312-313. In many basic Statistics texts (and also in the basic R package), the Greek letter rho (ρ) is used to indicate the estimated Spearman non-parametric correlation coefficient. Unfortunately, this is not strictly correct. In Statistics, Greek letters are used to denote population parameters, whereas Latin characters are used to denote estimates. Thus, rho (ρ) should not be used to denote the estimated correlation coefficient, but another symbol should be used instead. To respect this statistical usage, many authors use the appropriate symbol r_s (r subscript s), and I invite the authors to do the same.

L 602. The authors know well the difference between Shannon diversity and Shannon index. However, I wonder if calling the Hill number of order $q = 1$ Shannon diversity and the one of order $q = 2$ Simpson diversity is warranted. My fear of the possibility of a reader of this manuscript getting confused between Shannon Index and Shannon diversity is fed by your insistence to clarify in parentheses that you are referring to Hill number of order $q = 1$ every time you mention the term 'Shannon diversity'. This is just a reflection on nomenclature, and I kindly ask the authors to think about it.

(Remarks on code availability)

The authors provide all the necessary code to make the results of the paper reproducible. However, not all the code is clearly described. For example, Calculate_recoverytimes.py and Explain_recoverytimes.R, which are essential to understanding how the recovery metrics were calculated, are not fully documented. It is important that the authors complete the documentation of all the code.

(Remarks to the Author)

I co-reviewed this manuscript with one of the reviewers who provided the listed reports.

(Remarks on code availability)

Version 2:

Reviewer comments:

Referee #1

(Remarks to the Author)

This is a repeated revision of the paper. I am now satisfied with the changes in response to the referees' comments including mine.

(Remarks on code availability)

Referee #4

(Remarks to the Author)

The authors have made an outstanding effort to address the theoretical and conceptual issues raised by my co-reviewer and me in our previous report on this manuscript, and I would like to thank them for their effort. In its current form, this study rests on a solid, up-to-date theoretical framework and, given its novelty and biological breadth, will certainly make a significant contribution to the field of Biological Conservation Science. We have no further comments or questions on these issues.

This said, there is still a minor but relevant issue with the modeling used in this manuscript to analyze the large dataset that warrants attention. We recognize the authors' effort to explore an alternative model describing the return of species to their old-growth forest status. First, the authors state that the alternative model is a linear model with a square-root transformed time-axis; this is not a linear model. The equation $y = b_0 + b_1 \sqrt{x}$ means that the actual model that they fit is:

$$[(y - b_0)/b_1]^2 = x, \text{ i.e., } y^2/b_1^2 - 2*b_0/b_1*y + b_0^2/b_1^2 - x = 0.$$

This is clearly a parabolic, not a linear, model. Second, and more importantly, as their Fig. R1 shows, the parabolic and negative-exponential models differ in their recovery-time estimates when the species has long recovery times. This is a clear example of how model uncertainty plays an important role in estimating ecological metrics from models, a fact many authors have emphasized (e.g., 0.1016/j.tree.2003.08.001, 10.1002/ecm.1309, 10.1111/j.1365-2699.2006.01460.x). A simple ChatGPT search (prompt: "Give alternative monotonic non-linear models that connect point (x1, y1) to (x2, y2)") suggests exploring exponential, power, logarithmic, monotonic rational, sigmoid, cubic and spline functions to describe the type of pattern the authors seek to fit to the data. We are not suggesting that the authors try to fit all these models and compare them formally, but they must recognize in the text, preferably in the Discussion, the model uncertainty they are not addressing in presenting their results. Admittedly, the authors did account for data uncertainty when calculating their jackknife confidence intervals. However, model (not data) uncertainty is also an issue. Nowhere in the Discussion are model uncertainty and data uncertainty addressed. We recognize that this is not a modeling paper, but their conclusions largely rest on estimates from the available dataset and a single model. Therefore, it would be appropriate to devote a couple of lines to reflect on these two aspects of a modeling exercise to study biodiversity resilience.

(Remarks on code availability)

Referee #5

(Remarks to the Author)

I co-reviewed this manuscript with one of the reviewers who provided the listed reports.

(Remarks on code availability)

We are grateful to all three reviewers for their very insightful and helpful
suggestions, to which we respond in detail below. All comments helped to
substantially improve the quality of our manuscript and clarify the message. The
most visible changes – in line with suggestions by the reviewers – are

(1) the differentiation of legacy effects (cacao versus pasture), highlighting the
differences in recovery trajectories for these two contrasting land use types

(2) a much-improved description of the study design, spatial independence and
other prerequisites of the chronosequence,

(3) a refined definition of the return rate, which now better acknowledges the non-
linear behaviour of the recovery trajectory,

(4) a better readability of relevant parameters for biodiversity conservation goals,
e.g. a new table in the main text that also includes the completeness of recovery
after 30 years, and

(5) significantly improved conclusions emphasizing the relevance of our findings for
global biodiversity conservation goals.

Despite the improved differentiation and a new table in the main text (rather than as
an Extended Data Table), our manuscript is still within the length limits according to
the word count; alternative solutions to reduce the size of the manuscript would
include to use the Table 1 as an Extended Data Table.

In addition, we also updated the text in many instances, and, where necessary,
changed the underlying code to align with the suggestions made by the reviewers.

The main results, interpretation and conclusions were unaffected by these changes,
but the readability, quality and robustness of the study has been substantially
enhanced due to the suggestions by the reviewers.

We respond to each comment point-by-point below. All references mentioned here
and additional changes are summarized at the end.

Referee #1 (Remarks to the Author):

I am quite enthusiastic about this study based on impressive data sets from one of the
few well-established, replicated and sufficiently long rainforest successional chrono-

series. The study brings the concept of resilience vs resistance as two components of
recovery. These notions are not entirely new, but the quantitative approach
demonstrated here with the recovery resolved into these two components is quite
insightful and useful. The analysis is comprehensive, including a wide range of taxa
(including bacteria) as well as a range of alpha and beta diversity measures (Hill
numbers). Further, it is complemented by the literature search and analysis of other
existing data – all of them by far less robust than the current analysis. The study is
exceptional considering the data sets and the analytical approach, but also the results.
I am surprised – but convinced by the study – that generation time (and generally slow vs
fast life history) is not a good explanatory factor for the recovery. Further, this study is
absolutely right in emphasizing that while species diversity and abundance could recover
often quickly, this is not the case for species composition – and emphasized, quite
correctly, the importance of species composition. There have been some over-
enthusiastic studies previously, even from the same chrono-sequence studied here,
focusing on species richness and abundance and arguing on their basis that the rapid
recovery demonstrated high biodiversity value of secondary forests. While secondary
forests do have their value, this perspective disregarded the identities of species; the
current paper provides a very useful corrective.

*We thank the reviewer for this positive evaluation of our work, and for highlighting*
*the strengths such as resolving resistance and resilience, robustness of our*
*sample, broad taxonomic coverage, and the focus on community composition*
*rather than species richness alone.*

Referee #2 (Remarks to the Author):

The manuscript by Metz et al evaluates for a landscape in the tropical Colombian Chococo
the resistance of 16 taxonomic group to forest conversion to agriculture land use, and
their subsequent recovery (‘resilience’) towards old-growth forest values, and how much
time this takes.

For me their main findings are that 1) recovery is slower for composition than for
abundance or diversity, 2) recovery is more driven by ‘resilience’ than by resistance, 3)
resistance and ‘resilience’ are uncoupled, 4) simple attributes can not explain difference
in resistance and resilience of taxa, 5) some unexpected groups do not recover (soil

bacteria, leaf-litter arthropods, nocturnal insects)

The strengths of the manuscript are 1) the comprehensive description of recovery across
many taxonomic groups belonging to different trophic levels, using a standardized plot
network and analytical framework which allow for strong generalizations of the results,
2) the interesting analytical framework by decomposing recovery into its two underlying
components, resistance and 'resilience' which are associated with different ecological
processes (tolerance versus colonization and regrowing capacity), 3) the surprising
result that some short-lived taxonomic groups show arrested succession, and that
simple ecological attributes can not explain resistance or recovery.

The weaker parts of the manuscript are that 1) it focuses on recovery time of
composition, which is difficult to quantify for several taxonomic groups because it takes
more than the age of the oldest secondary plots (40 years), 2) two very different land uses
were included (cocoa plantations and pastures) but their affect on resistance and
'resilience' are not explicitly taken into account in the model, 3) sometimes unclear
description of the study system and the statistics.

Overall, the manuscript was fluently written, interesting, and a pleasure to read, and it
provides in my opinion a novel and important contribution to science and, potentially, to
the application of restoration.

Please find my major and minor comments below. I hope they are of use to improve the
manuscript.

*We thank the reviewer for the positive and hugely constructive and insightful*
*evaluation, and for well-justified comments and very detailed suggestions. All*
*points and suggestions were considered in the revised manuscript where*
*possible, helped to greatly improve the quality of the presentation of the study*
*results (all without affecting the main conclusion). We respond to each of the*
*individual major and minor comments below.*

MAJOR COMMENTS

1. ACCURATE QUANTIFICATION OF RECOVERY TIME? You have opted to focus your story
on recovery time of composition (because that recovers most slowly and you find it most
important). From Fig 2 it seems (difficult to guesstimate with the logarithmic axes with
few tickmarks) that 21 of your 56 comparisons (3x18) , i.e., 38%, have a recovery time

longer than the oldest secondary forest plot in your dataset (40 years). If you see the
greyscale of the 95% confidence interval, then many taxonomic groups have a
confidence interval between 10-120 years. The question therefore is how accurately you
can estimate recovery time, and how meaningful this is if the confidence intervals are so
large? Hence, if the strength of your data is not there, then put less emphasis on recovery
time, and more on resistance and 'resilience', that you probably can estimate more
accurately.

*We thank the reviewer for addressing these shortcomings of recovery time*
*estimates for cases where recovery takes very long.*

*We agree that recovery times of those properties or taxa that fully recover within*
*the time span represented by our chronosequence are more reliable than those*
*that remain incomplete in the oldest secondary forest plots.*

*We now additionally calculated the relative recovery (in percent) after 30 years as*
*an additional measure, i.e. a time span well covered by the plots. A similar*
*approach was also taken by Poorter et al. (2021) for trees and ecosystem*
*functions. 30-year values are now summarized in Table 1 for the community*
*composition of all taxa (for cacao and pasture legacies separately, with*
*confidence intervals), together with the recovery time to 90% of the old-growth*
*forest reference values, resistance and return rates. They are also mentioned in*
*the main text. 30 years represents a more relevant time frame for conservation*
*targets than longer times for full recovery. These values are surprisingly high for*
*species composition of many taxa, due to the non-linear recovery trajectory,*
*which further strengthens our target of providing useful information on when*
*natural regeneration of secondary forests can be expected to provide significant*
*conservation benefits. In the Supplementary Tables 1 and 3 we also report the 30-*
*year values for abundance and diversity, but since many taxa reach full recovery*
*earlier, this metric is less informative regarding the differences across taxa. We*
*also generally strengthened the discussion of results obtained for recovery times*
*of abundance and diversity, as these attributes are also very important for a*
*holistic understanding of forest recovery (see also answer to minor comment 21)*
*and for many taxa these attributes recover significantly within a time frame*
*covered by our chronosequence.*

Despite the fact that recovery times, especially for species composition, can lie beyond the time covered by our chronosequence, we decided to also keep all results for recovery times in the main text. To better acknowledge the limitations of forward predictions, mostly relevant for composition, we changed our wording in figures and text to “Predicted recovery time” instead of “recovery time”. We think that our finding that species composition takes much longer than abundance and alpha diversity – consistent across taxa – is an important message of our study (also highlighted by reviewer #1) and has fundamental conservation implications, questioning the use of simple proxies such as species richness for more specific target such as conservation of vulnerable species.

We think these results are generalizable, as our chronosequence is representative regarding its regeneration time range (0–38 years) compared with other Neotropical studies for tree communities (oldest plots: median 40 years, range: 15–100 years; values computed from Rozendaal et al., 2019) and our 62 plots are highly resolved in space and time (spatially independent and without e.g. elevational bias, see the discussion of the study design in the answer to minor comment 3). The values for the return rate (λ) would have the same accuracy as the recovery time, because the recovery time and λ are calculated with the same function. A potential alternative solution to using the asymptotic function is to fit a linear model (see e.g. Rozendaal et al., 2019, Hoenle et al., 2022) with a square-root or log-transformed time-axis through the recovering plots (secondary forests) only. However, we agree with Poorter et al. (2021) that ignoring solid data for the reference state and thus endpoint of recovery may yield less accurate estimates; we thus use the asymptotic function and include the reference stage, even if predicted recovery times are in some cases beyond the time covered by the chronosequence. This is especially because the old-growth forest values are unusually well represented in our study compared to many other chronosequence studies, as we have 17 replicates that are all spatially spread across the entire study region. This suitability of the design and importance of well-represented old-growth forests are discussed in the answer to major comment 4 in the context of an improved description of the study

*area. We are therefore confident that the predicted recovery times are valid within*
*their respective confidence intervals and extrapolation beyond 40 years is reliable*
*using the asymptotic function due to the high availability of old-growth forest*
*reference plots.*

2.UNCLEAR VALUES AND COMPARISON OF RECOVERY TIME. You say that abundance
and diversity recover faster than species composition (l180-182). For abundance and
diversity you give such a wide range (0-100 years) and no median, and for species
composition you report no values at all, so that it is not informative anymore. If you want
to put this in an ecological or restoration perspective then it is important to know the real
values. In Fig 2 you put the values on a log10 scale, and it is really difficult to infer the real
values. So please include a table of the 18 groups with the resistance, resilience,
recovery time and their corresponding confidence intervals. Fig 2 does not confer much,
apart that it varies. Please include in an appendix a figure with 18 groups of 3 bars (one
for abundance, one for species diversity, one for composition) (mean and confidence
interval), so that it is easier to quantitatively compare groups and recovery time of
different attributes

*We thank the reviewer for addressing these important points. Whereas these*
*values had been reported with confidence intervals in Supplementary Table 1 and*
*3 in the previous version, we now additionally highlighted them (for species*
*composition) and included the requested table as “Table 1” in the main text. The*
*values were calculated for pasture and cacao legacy plots separately (in response*
*to the suggestion in major comment 3), hence this distinction is now included in*
*the table and main text as well. We agree that an enhanced visibility of such values*
*(not only their position in a figure) is helpful. We also included more tickmarks on*
*the x-axis of Fig. 2 to improve readability and the usefulness of that figure. We also*
*included the requested figure with the different groups of 3 bars with mean and*
*confidence intervals as Extended Data Fig. 3 for cacao and pasture legacy*
*separately. We additionally report more informative values for recovery times in*
*the main text (p.6, l.189-193):*

**Abundance and diversity (Shannon diversity, i.e. Hill number of order 1) of**
**many taxa remained high even during agricultural use and early recovery**

**stages and recovered faster (between 0 and 258 years, median**
**cacao/pasture: 4.3/25.5 years (abundance), 3.2/20.7 (diversity)) than species**
**composition (between 0 and 724 years except for bacteria, which took**
**significantly longer, median cacao/pasture: 51.6/76.7 years).**

3.INCLUDE THE 2 LAND USE TYPES IN THE FORMAL ANALYSIS. You have two types of
land use (cocoa and grasslands) that differ totally in openness, structure, growth form
composition, management intensity, and hence are likely to differ in resistance,
resilience, and recovery time. You say somewhere that the estimates for resistance did
not differ between the two land use types, but I did only see the test statistic and not a
graph with the data, and given that it is a confounding factor I think it is better if you
include it formally in your statistical analysis (i.,e., by allowing the two land use types
have different intercepts and slopes). Color code your dots in Fig 3 based on land use
type, and show two different lines if they have different intercepts or slopes

*We thank the reviewer for raising this important point (also raised by reviewer #3*
*in comment 3.2). Indeed, we studied the two different land-use types as a main*
*part of our study design, and thus calculated separate recovery trajectories for*
*plots with cacao legacy and pasture legacy besides those for all plots combined.*
*An effect of land-use legacy was part of our main objectives (see Guariguata &*
*Ostertag, 2000, Jakovac et al., 2021, now included in the references), and in fact a*
*reason for us to include equal shares of cacao and pasture land use legacy in the*
*study design with a very similar age distribution (see Escobar et al. (2025)*
*<https://esajournals.onlinelibrary.wiley.com/doi/full/10.1002/ecs2.70157> and the*
*revised method section in the manuscript).*

*Whereas these legacy effects had only been included in Supplementary Table 1*
*and Supplementary Table 3 in the initial version of our manuscript for*
*simplification purposes, we now summarize and highlight them in the main text,*
*hence we included the values in Table 1 and modified Figures 2, 3 and 4 in the*
*main text and also included the new Extended Dada Figures 1-5 to show this*
*differentiation. Indeed, some taxa show a legacy effect, i.e. differences in*
*responses to cacao and pasture. Interestingly, seedlings showed a strong legacy*
*effect, but adult trees did not – the latter was consistent with earlier studies that*

*showed no clear effect of land-use legacy (see Rozendaal et al., 2019). However,*
*several animal taxa showed a clear advantage for cacao regeneration over*
*pastures, which are now discussed in the section “Recovery times of multiple taxa*
*in a tropical forest ecosystem”. Therefore, we agree that the differences in land-*
*use legacy are highly interesting (for ecologists and practitioners), and showing*
*the differences explicitly improves our message.*

4.IMPROVE DESCRIPTION OF YOUR STUDY SYSTEM. Explain better your study system
upfront in the main text and main methods, as this matters how representative your site
and results are, and what resistance and recovery you can expect. E.g., rainfall, forest
type, and soil fertility determine the maximum pace and asymptote of recovery, and the
types of plants and animals you are likely to have. Describe the landscape context (how
much forest nearby) as this determines the source population and dispersal ability and
your ‘resilience’. Describe for both land use systems the typical area and duration of use,
and management practices, as this previous land use intensity strongly shapes
resistance and ‘resilience’. E.g., for the pasture in addition exotic grasses, stocking
density, and fire use, and for cocoa plantations if this was open grown or shade grown
with remnant shade trees, pesticides, etc. Describe the heterogeneity in your landscape
(i.e., variation in steepness of slopes, altitude, geology and soil type) as this determines
your beta diversity and similarity that one can expect.

*We agree that a better description of the study area and properties of our spatial*
*design (e.g. the independence of the plots) should be provided in such a synthesis*
*paper as a ‘stand-alone’ section in the methods, despite a key reference in which*
*we described the study system (Escobar et al., 2025). We thus now described the*
*study system in much more detail, particularly focusing on the spatial distribution*
*and independence of the plots. The suitability of the design is an important*
*prerequisite for the recovery analysis, particularly the lack of environmental*
*biases associated with the chronosequence. Our substantially revised methods*
*(section: "Chronosequence") now covers each of these important aspects*
*addressed by the reviewer. We also improved the short description of the study*
*site in the main text (p.5, l.167-169):*

**The landscape consists of a mosaic of secondary forest, old-growth forest**

**and agriculture with relatively high forest cover (~75%⁴⁵) that is representative**
**for many neotropical regions (median ~85% of 56 sites¹⁷).**

*and (p.5, l.170-173):*

**Our 62 study plots are each 0.25 ha in size and include 6 actively used**
**pastures and 6 cacao plantations, 33 secondary forests of variable age (1–38**
**years) recovering from previous agricultural use (16 pasture/17 cacao), and**
**17 old-growth forests as a reference (see Supplementary Table 6 for details).**

5.HOW REPRESENTATIVE ARE OLD-GROWTH FOREST? I was highly surprised that some
groups did not recover (soil bacteria, leaf-litter arthropods, nocturnal insects). Normally
bacteria are not very responsive to secondary succession or recover soon (maybe
because there is a mismatch between the micrometer scale that is relevant for bacteria
and the plot scale that is relevant for plants, or because the fast generation time of
bacteria). I had expected leaf-litter arthropods not very sensitive because litter
production and litter thickness often recover very soon in succession (after ca 10 years?),
and as it is brown litter I expect less specificity). And I expected nocturnal insects to be
less affected by the surrounding landscape (as they forage at night when it is cooler and
more humid) and fly and recover faster. Hence, I wondered how representative are your
old-growth forest (OGF). Sometimes they are much further away (as people have
secondary forests (SF) closer to their homesteads) than secondary forests. Please
provide the average distance and range of SF to OGF plots and amongst SF plots and
between land use and OGF plots. Similarly, people often do not use OGF if they are on
too steep slopes, or on an unproductive soil type or geological formation. This may
explain why you have totally different bacterial communities (or other groups) there.

*The old-growth forests (OGFs) are now described in much more detail (see*
*methods section: “Chronosequence”). The number of plots is highly*
*representative (17 plots, each 0.25 ha, thus the total area covered is 4.25 ha and*
*larger than OGFs of other chronosequences in the Neotropics, see Escobar et al.,*
*2025). Moreover, the OGFs are spread over the entire study area with an average*
*distance of 5 km between OGFs – directly comparable with the secondary forests*
*that have the same distances within each category. While OGFs tend to be higher*

*in elevation than agricultural areas in the broader study region, the selection of the*
*plots aimed to minimize such biases, and thus elevation and topography of the*
*selected OGFs were not significantly different from the secondary forest plots. We*
*now highlight the distribution of the plots and forest cover in the revised methods,*
*as well as the unbiased design (analysed in detail in Escobar et al., 2025).*
*Regarding unexpected patterns: We assume that the altered bacterial*
*communities in active agricultural plots and young secondary forests are an effect*
*of disturbance (e.g. microclimate, grass, pesticides), and a strong dispersal*
*limitation in the soil communities. While this is indeed a particularly surprising*
*finding compared to most other taxa, we do not want to overinterpret the pattern*
*here. Leaf-litter arthropods are poor dispersers as well (mostly represented by*
*flightless taxa), and even if litter accumulates, the entire community composition*
*may then not recover quickly. Nocturnal insects are good dispersers and highly*
*mobile (trapped during flight), and they should be less affected by hot*
*microclimate in young secondary forests, but many of them are forest specialists*
*(e.g. host-specific herbivores) and sensitive to deforestation. Unfortunately, there*
*is no space in such a broad synthesis for discussing each taxon individually in*
*more detail, but details will be covered by individual studies.*

6.IMPROVE RESILIENCE TERMINOLOGY. Please stay in line with mainstream definitions
of resilience. You cite Holling for your definition of resilience. Holling (1973, p. 17) defined
ecological resilience as ‘resilience determines the persistence of relationships within a
system and is a measure of the ability of these systems to absorb changes of state
variables, driving variables, and parameters, and still persist. Hence, I think resilience
comes most close to your recovery time (with a low recovery time being a high resilience.
In tree-ring studies people normally analyze growth resilience (for example to drought)
as the product of resistance and recovery. Hence, what you call ‘resilience’ is in fact
‘recovery’ for many ecologists.

*We thank the reviewer for pointing out inconsistencies and ambiguities in the*
*definition and measurement of resilience. A similar point was raised by reviewer*
*#3 (comment number 1), and the answers to these two suggestions overlap. We*
*agree that resilience is a concept with multiple definitions that needs proper*

*definition in each study. To respond to this comment and to avoid confusion with*
*different definitions of resilience, we now changed the term “resilience” to the*
*undisputed term “return rate” in all cases where we refer to the measurement*
*itself (rather than the broader context of the study). We explicitly use the term*
*“return rate” for the specific measure represented by lambda from Eq. 2, and*
*“recovery time” for describing the predicted time until 90% of the reference value*
*is. We thus adopt “return rate” throughout (e.g. as axes in the figures) as a very*
*intuitive term describing the speed of recovery relative to the amount lost.*

*We put a more concise definition of all used terms into the main text (p.4, l.129-*
*135):*

**The recovery trajectories and recovery times of different components of**
**tropical forests (e.g. the species composition of a certain taxon) depend on**
**resistance - the ability to withstand disturbance - and the return rate to**
**reference conditions after perturbation (Fig. 1)^{26,27}. Resistance is related to**
**attributes that confer tolerance to perturbation such as physiological or**
**behavioral plasticity within taxa or features that provide protection against**
**change^{26,28,29}. Return rates provide a measure of resilience (in the sense of**
**speed of return to reference conditions^{30,31}) already standardized with the**
**amount of change caused by the disturbance³²⁻³⁴.**

*We also provide a much improved explanation on all the terminology along these*
*lines in the methods section (p.24, l.703-710):*

**Resilience is often either defined as the ability to maintain a qualitatively**
**similar state in the face of disturbance and not switch to an alternative state**
**(ecological resilience^{27,59}) or the speed of return to reference conditions**
**(engineering resilience^{26,27}). Here, we follow the engineering resilience**
**concept and define the return rate λ from Eq. 2 (see also Fig. 1) as a measure**
**of resilience standardized by the amount of change caused by the**
**disturbance^{26,28-30}. However, to avoid confusion with other definitions of**
**resilience in the literature, throughout the manuscript we use the term**
**“return rate” for the specific measure represented by λ , and “recovery time”**
**for describing the time until 90% of the reference value is reached.**

*In our manuscript (and in other studies of our research unit) we do not follow*

*Holling’s concept of “ecological resilience” which is often related to alternative*
*stable states and deemed difficult to quantify (Pimm et al., 2019, Van Nes &*
*Scheffer, 2007, Van Meerbeek et al., 2021). We follow the quite commonly used*
*definition of resilience as speed of return to reference conditions in line with the*
*“engineering resilience” concept of Holling (Holling, 1996) and based on Pimm’s*
*definition of resilience (Pimm, 1984). We agree that an alternative useful and*
*measurable definition of engineering resilience would indeed be the total recovery*
*time, but our definition of viewing resilience as the return rate from the baseline*
*(which is “the amount remaining”, i.e. the level defining resistance) to reference*
*conditions (also in line with the “speed of return” definition) follows calls to make*
*resistance and resilience conceptually independent (already mentioned by*
*Pimm, 1984 and more recently by Pimm et al., 2019, Justus, 2007, Hillebrand et*
*al., 2018, White et al., 2020 and Van Meerbeek et al., 2021 and Table 1 therein,*
*also given below).*

[Table Redacted]

[Text Redacted]

[REDACTED]

7.UNDERSTIMATION OF RECOVERY RATES; USE REAL SLOPE. As nearly all measures of
recovery of composition show a saturating relationship with time, this means that if you
calculate “resilience” as $(1 - \text{resistance}) / \text{recovery time}$ (L571) you underestimate the
‘resilience’ rate (i.e., recovery rate) over nearly the whole time interval (see extended data
fig 6). Hence, I think it is better to use the real slope. Moreover, given that your recovery
397 times are difficult to estimate (see above), your recovery rate is neither so precise.

*We thank the reviewer for addressing this point. Our goal was to give a linearized*
*approximation of the return rate in %/year, as we thought this would yield a more*
*intuitive (average) measure that is easier to interpret for applied conservation*
*scientists. However, we agree that this value is then oversimplifying the reality of*
*typical non-linear trajectory while also being difficult to understand. We therefore*
*now report only the return rate λ (similar to the “intrinsic recovery rate”*
*reported in Poorter et al., 2021) instead of the “linearization” $((1 -$
*resistance)/recovery time.**

8.JACKKNIFE CONFIDENCE INTERVALS. L602-610. I am not familiar with the Jackknife
method, but if you leave 1 out of 62 plots out, then the resistance and resilience

estimates will change very little. This may give erroneously the feeling that the
confidence intervals are very narrow while they are not. Can you calculate a more
informative confidence interval, directly based on the data?

*Generally, there are two alternative resampling-based methods to calculate*
*confidence intervals: jackknife and bootstrapping. There is some literature about*
*benefits and limitations of both methods, with no clear recommendation (e.g.*
*Dixon, 2020). However, bootstrapping (i.e., resampling with replacement) omits*
*on average 31% of the data, meaning that we obtain information of a substantially*
*smaller subset of plots per simulation — discussed as a shortcoming in Efron &*
*Tibshirani (1993) or Manly (1997). For a dataset where the extremes (agriculture*
*with age=0 and old-growth forests) are highly represented in the age distribution*
*unlike a in a Gaussian distribution, resampling of plots with such a proportion of*
*omissions can have unexpected consequences. For example, for skewed*
*population data, Meyer et al. (1986) concluded that for a case where the*
*“sampling distribution was negatively skewed [...] Bootstrap and full-sample*
*estimates of r were negatively biased” unlike Jackknife.*

*Jackknife, in contrast, uses the full dataset except one datapoint rather than a*
*substantially reduced subset, and is particularly useful for small datasets. We*
*therefore deemed the confidence intervals calculated with the jackknife*
*procedure to be preferable over bootstrapping. Therefore, we decided to keep*
*calculating the confidence intervals using the Jackknife-procedure, but included*
*a more elaborate explanation (p.25, l.739-742):*

**Errors for recovery times, resistance and return rate were estimated using a**
**jackknife procedure⁶⁰. This was preferred over bootstrapping, as**
**bootstrapping omits on average ~31% of the data and for small datasets this**
**leads to substantial issues in the estimation of the fit as only a substantially**
**lower amount of information is available^{60,62}.**

9. JUSTIFY ALL YOUR CHOICES IN THE METHODS. The methods read now as a cookbook.
I can repeat what you have done, but for many choices I do not understand why you did
it. Please justify, so that the reader can understand and judge your choices

*We thank the reviewer for this comment. We agree that our methods section*

*lacked proper justification regarding why methods were chosen in some*
*instances. The description of the calculation of Hill-numbers and species*
*composition similarity is now much improved. Additionally, the jackknife method*
*is now better justified in the methods (references provided). We also now*
*improved the description of the spatial design, the lack of bias and independence*
*of the study plots in the methods. We additionally justified, in many instances, why*
*we chose the methodology.*

MINOR COMMENTS

1. Fig 3. Interesting graph! Please let the y axis of ALL graphs start at zero, and put more
tickmarks so that it is easier to read the values. Write next to the icons the taxonomic
group, make the median value line of OGF and land use systems thicker, provide different
systems for the two land use systems, and show them with separate recovery lines if the
differ in their slopes and/or intercepts Put similar graphs for abundance and diversity in
the appendix.

*We thank the reviewer for this positive comment. We changed the graph to*
*incorporate all mentioned suggestions. We also included similar graphs for*
*abundance and diversity in the appendix as Extended Data Fig. 1 and 2.*

2. Your life histories, dispersal modes, and trophic levels are rather coarse. So how
informative is it to test how they are related to resistance/resilience, and recovery time?
Please tone down a bit your conclusions, and say that this is a first
analysis/approximation.

*Thanks for pointing this out. We now add (p. 14, l.363-370):*

**Overall, our results suggest that recovery times depend on more complex**
**interactions that are not captured by simple metrics such as trophic levels,**
**dispersal mode and age of reproduction. Consumer-resource interactions**
**from predation, competition and facilitation to ecological fitting and**
**mutualism may be more important for influencing the order of community**
**recovery. However, a more fine-grained classification of species' life-history**
**strategies also within taxa and including other trade-offs might reveal further**
**insights. We suggest that resolving the temporal sequence of the order of**

**species' establishment both within and across taxa is a promising way to**
**understand functional ecosystem recovery.**

3. LOW SIMILARITY HAMPERS RECOVERY TIME ESTIMATES? If I guesstimate median
pairwise similarity amongst OGF plots for your 18 taxa than I get a mean of 0.32 (range
0.12-0.50). This is quite low, perhaps because of a high spatial or temporal
heterogeneity, combined with small plot sizes and small monitoring time windows for
some taxonomic groups. Then you try to estimate recovery to such a low value, and I
wonder how accurate that is.

*This low similarity level (i.e. the high beta-diversity) of most communities studied*
*here is indeed a characteristic of our rainforest and involves most taxa, but this is*
*not untypical for tropical moist forests. The plot size of 0.25 ha (for mapping trees)*
*has been defined to allow for a high number of plots (total N=62, covering 4.25 ha*
*old-growth and 8 ha regenerating forests, which exceeds other chronosequence*
*studies; see Fig. 1 below from our "study design" paper: Escobar et al. (2025). In*
*particular, this study design allowed us to better represent the entire study region,*
*avoiding any spatial bias (e.g. there was no elevational bias, no clustering and*
*spatial autocorrelation etc.) (Fig. 2 and Fig. 3 below). The accuracy of our*
*statistical model predictions depends strongly on the assumptions of (spatially)*
*independent plots. This is particularly important for mobile animals, for which the*
*plot size as such (for tree measurements) is not as relevant as the patch size of*
*the (land-use) unit. Patch sizes are on average 9-12 ha for current and previous*
*pastures and 2-8 ha for current or previous cacao plantations, see Escobar et al.*
*(2025). Only one plot was selected per patch, to avoid pseudoreplication. Old-*
*growth forest patches are much larger, of course. We now considerably improved*
*the site description in the revised version and added this important information*
*independent of the cited paper.*

[Figure Redacted]

- [Redacted]
- [Redacted]
- [Redacted] [Text Redacted]
- [Redacted]
- [Redacted]
- [Redacted]

[Figure Redacted]

█

█

█

█

█

█

█

█

█

█

█

█

█

█

█

█

█ [Text Redacted] █

█

█

█

█

█

█

[Figure Redacted]

[Redacted]

[Redacted]

[Redacted]

[Redacted]

[Text Redacted]

[Redacted]

[Redacted]

[Redacted]

[Redacted]

4. Line (l) 95. ‘Stability’ comes out of the blue. Do you need it, given that land use
conversion of forests to pasture is by definition not stable.

*We agree and excluded the term.*

5. L97. I think ‘recovery’ indicates the speed of recovery

*Thanks. To avoid confusion of different definitions in the literature, we improved*
*the definitions of all used concepts and terms (see answer to major comment 6).*

6. L100 ‘Mobile animal communities .. recovered faster than trees ... suggesting that
these functional groups drive the recovery of tree communities’. The alternative
explanation is that animals and trees recover independently. I think to support your claim
you should do a network analysis between recovery of animal species/groups and plant

species/groups

*We thank the reviewer for suggesting this analysis to strengthen our conclusions.*
*A pairwise comparison or meaningful correlational network analysis (such as e.g.*
*done in Poorter et al. 2021) would require data of multiple chronosequences to*
*obtain results, which is not feasible for our study, given that the data from*
*chronosequences elsewhere do not provide sufficient cases for pairwise*
*comparison between trees and animals either (see our analysis of literature data*
*in this manuscript). We however agree that the statement is speculative and thus*
*removed it from the abstract to leave only those findings that are more robust. We*
*wrote (p.3, l.96-97):*

**Mobile animal communities that provide key functions, such as seed**
**dispersal or pollination, had high resistance levels and recovered faster (0-**
**100 years) than trees or tree seedlings (ca. 100 years).**

7. L104 replace ‘rules’ by ‘mechanisms’?

*Thanks, corrected.*

8. L118-120. This feels a bit like a strawmen’s argument. You found 32 studies that
analyzed recovery of other taxa, and there are several meta analysis and multi taxon
studies. Although they have not measured trees alongside they can for sure inform how
different taxa recover. Please indicate the state of the art and summarize briefly what
they found

Dent, D.H. and Wright, S.J., 2009. The future of tropical species in secondary forests: a
quantitative review. *Biological conservation*, 142(12), pp.2833-2843.

Díaz-Vallejo, E.J., Seeley, M., Smith, A.P. and Marín-Spiotta, E., 2021. A meta-analysis of
tropical land-use change effects on the soil microbiome: Emerging patterns and
knowledge gaps. *Biotropica*, 53(3), pp.738-752.

Oliveira, P. S., Falcão, L. A., Almeida, J. S., Fernandes, G. W., Reis Júnior, R., Nunes, Y.
R., ... & do Espírito Santo, M. M. (2024). Diversity patterns along ecological succession in
tropical dry forests: a multi-taxonomic approach. *Oikos*, 2024(4), e09653.

*We thank the reviewer for suggesting these additional studies to include in our*
*introduction. We changed the introduction to incorporate the information*

*presented in the given studies and provided a short section on the state of the art*
*on recovery of animal taxa. We wrote (p.4, l.117-125):*

**The recovery of animal and microbe communities remains poorly studied.**
**Studies suggest that species composition of different animal groups recovers**
**within decades and that animal species richness may recover more rapidly**
**than species composition²⁰⁻²³. However, these results are mostly based on**
**small samples with few replicates and information across taxa is scattered**
**among different studies, regions and forest types that cannot be compared**
**quantitatively²⁴. Understanding the recovery of multiple animal taxa**
**alongside trees is essential to allow a holistic and robust estimation of the**
**potential of secondary forests for biodiversity conservation.**

9. L124-125 ‘Uncertainty about the time scales needed for natural recovery .. hampers
effective decisions to achieve restoration goals’. Explain here or in the discussion how,
because that is not sooo clear.

*We agree that this sentence needs more explanation. We now changed the*
*wording as follows (p.4, l.125-127):*

**Decreased uncertainty about the timescales needed for successful natural**
**recovery of biodiversity in secondary forests may foster natural regeneration**
**as a cost-effective restoration tool and thereby help reaching restoration**
**goals^{8,19,25}.**

10. L134. Not only plasticity, but also protection against change (i.,e., the ability to
persist belowground as rhizomes or seeds)

*Thanks. We added this to the sentence.*

11. L137 SMALL-SCALE disturbances (add small scale)

*Thanks. Corrected.*

12. L138. I would say ‘are an inherent feature of the system’

*Thanks. Corrected.*

13. L143. Add a quantitative justification: ‘successful tropical forest recovery AS MOST
ARE INSECT- OR ANIMAL POLLINATED (..) OR DISPERSED (..)’

*Thanks. We wrote (p.4, l.139-141):*

**The resistance and return rates of animal taxa that provide key functions,**
**such as pollination or seed dispersal, may be essential for successful**
**tropical forest recovery as 90% of the tree species are animal dispersed and**
**94% are pollinated by animals^{43,44}.**

14. L165. Write ASV in full

*Thanks. Corrected.*

15. L168 define what you mean with ‘dispersal ability’. To cover a large distance? in
space? Time?

*With “dispersal ability” we referred to the “predominant mode of dispersal (aerial*
*or terrestrial)”. We rewrote this part of the sentence.*

16. L168-169. Better describe your pasture and cocoa plantation system here (see my
main comments)

*Thanks. We wrote (p.5, l.167-173):*

**The landscape consists of a mosaic of secondary forest, old-growth forest**
**and agriculture with relatively high forest cover (~75%⁴⁵) that is representative**
**for many neotropical regions (median ~85% of 56 sites¹⁷).**

**Our 62 study plots are each 0.25 ha in size and include 6 actively used**
**pastures and 6 cacao plantations, 33 secondary forests of variable age (1–38**
**years) recovering from previous agricultural use (16 pasture/17 cacao), and**
**17 old-growth forests as a reference (see Supplementary Table 6 for details).**

17. L171. Rewrite. This suggests that you monitor your taxa over time, but you use a
chronosequence approach. Write in the methods the assumptions of the
chronosequence approach (all plots started from the same situation and followed the
same trajectory) and why you think this assumption is met

*We now rewrote the section as (p.5-6, l.173-175):*

**Our samples thereby provide snapshots across a full disturbance cycle**
**before (old-growth forest), during (agriculture) and after perturbation**
**(secondary forest).**

*We furthermore included an extensive description of our chronosequence and the*
*assumptions of the chronosequence approach in the methods section (see*
*Methods: Chronosequence, p.20-21, l.571-600).*

18. L175-178. Make explicit how info on recovery times is needed for conservation and
policy development

*Thanks for pointing this out. We merged this section with the section mentioned*
*in the answer to minor comment 9 to increase clarity of that general topic.*

19. L181. Provide median, min and max, recovery time for abundance, diversity, and
composition

*Thanks. We now include these values and write (p.6, l.187-193):*

**Predicted recovery times (see Fig. 2 and Extended Data Fig. 3) and trajectories**
**(see Fig. 3 for composition and Extended Data Fig. 1 and 2 for abundance and**
**diversity) varied strongly across taxa. Abundance and diversity (Shannon**
**diversity, i.e. Hill number of order 1) of many taxa remained high even during**
**agricultural use and early recovery stages and recovered faster (between 0**
**and 258 years, median cacao/pasture: 4.3/25.5 years (abundance), 3.2/20.7**
**(diversity)) than species composition (between 0 and 724 years except for**
**bacteria, which took significantly longer, median cacao/pasture: 51.6/76.7**
**years).**

20. L182. Why a signed rank test and not a paired t-test?

*Test-criteria for parametric tests were not always met so, for consistency and*
*robustness of the conclusions drawn, we used non-parametric rank tests*
*throughout our study. We added this explanation to the methods section (p.27,*
*l.809-811).*

21. L188. Please add: 'OR COMPLEX MEASURES MAY OVERESTIMATE SIMPLE

ATTRIBUTE RECOVERY' Recovery can happen in terms of multiple attributes, of which
some go slow, others go fast. It is unclear why you focus on the slowest recovering
attribute (only when that is recovered it is 'good'?). Please justify and explain better. To
me that feels like a quite reductionist and not a holistic view on recovery. I think it is
important and informative that abundance and diversity recover fast, as they are also
important ecosystem components. Focusing on the slowest component suggests that
everything goes slow and little is gained, which is not true.

*Thanks. We now write (p.6, l.198-203):*

**This result shows that dissecting attributes of ecosystem recovery is**
**important for a comprehensive understanding of tropical forest recovery.**
**Total abundance and species diversity, both linked to the high productivity of**
**the forest, recover quickly. Focusing on them may substantially**
**underestimate the time necessary for more complex ecosystem attributes**
**such as composition to recover. Composition, in turn, would fail to capture**
**the rapid recovery of ecosystem functions.**

*We also highlighted in the conclusion (p.14, l.373-377):*

**We show that the conservation of secondary forests rapidly and significantly**
**restores biodiversity, as taxa recovered on average more than 90% of their**
**abundance and diversity and ~75% of their compositional similarity to old-**
**growth forests within only 30 years. The high resistance of mobile seed**
**dispersers and pollinators, coupled with a high return rate of most taxa,**
**contributes to tropical forests quickly regaining their diversity.**

22. L193. 'in combination with slow recovery of tree biomass'. Litter production rates
recover really very fast (<10 years?), and that matters for litter dwelling insects, I would
say.

*We agree. Our unpublished data (not included here) also confirm such fast*
*recovery (and little disturbance) of litter production and decomposition. Litter*
*decomposition is driven to a large extent by microbes and to a smaller proportion*
*by arthropods. Colonization of arthropod communities (largely flightless taxa)*
*seems to be limited by their dispersal abilities. We removed this part of the*
*sentence.*

23. L193 Slow litter arthropod recovery because of slow recovery in tree composition. Can you give a reference how specific litter arthropods are for species-specific tree litter? I would expect that with brown litter insect-plant relationships are less specific than with green leaves

We thank the reviewer for pointing that out. Yes, specialization of herbivores (phytophages feeding on living plant tissues) is strongly driven by plant defenses, which should play a much lower role in decomposing litter (see e.g. Donoso et al., 2010). The slow leaf-litter arthropod recovery is likely primarily driven by low mobility. We therefore removed the half-sentence related to biomass (see answer to the previous minor comment) and species composition and now simply write (p.7, l.213-214):

The slow recolonization of leaf-litter arthropods was consistent with their relatively low mobility (many taxa are wingless).

24. L193. See above. My guess is that bacteria do not recover because the SF are somewhere else than OGF or under different environmental conditions

Our spatial design was chosen such that influences of environmental conditions and location were minimized. Therefore, we believe that the pattern we observe is not an artifact of the study design. We now significantly improved the description of our study design to clarify this point. We however, agree that the impact on the bacterial community is not severe, and therefore toned down the impact of agriculture on the bacterial community (p.7, l.215-217):

Soil bacteria species composition showed no recovery at all, indicating that agricultural practices and strongly altered climatic conditions had a long-term legacy impact on the bacterial community.

25. L201 ‘Once these mutualists and other groups recover.. they can support succession towards old-growth tree communities’. Do you have some literature references for that?

I know that that is what most of us biologists believe or hope, but I guess there is a lot of
redundancy in those relationships, and that different frugivore species can take over the
seed dispersing roles of others. See Dent et al who calculated redundancy in seed
dispersal mechanisms and showed that it increased during succession.
Estrada-Villegas, S., Stevenson, P. R., López, O., DeWalt, S. J., Comita, L. S., & Dent, D.
H. (2023). Animal seed dispersal recovery during passive restoration in a forested
landscape. *Philosophical Transactions of the Royal Society B*, 378(1867), 20210076.

*We agree. Full recovery of species composition of seed dispersers may be less*
*important for tree species composition recovery due to low specialization in seed*
*dispersal mutualisms. However, in combination with the high resistance of*
*abundance, diversity and also composition of seed dispersing animal groups, the*
*relatively fast recovery in diversity, but also in composition, indicates that services*
*to trees can be provided early during succession, thereby avoiding dispersal*
*limitation. We now framed this point more cautiously and also put more emphasis*
*on the aspect of high resistance in abundance and diversity. We therefore*
*combined the section mentioned here with the discussion of the resistance and*
*now write (p.11, l.284-296):*

**In contrast, mobile animal taxa providing key mutualistic services to trees**
**(bats, birds, moths and bees) were common in the agricultural and early**
**successional stages and thus had high resistance levels especially for**
**abundance (median cacao/pasture 46%/56%, range 25-75%) and diversity**
**(median cacao/pasture 89%/78%, range 45-99%) but also for composition**
**(median cacao/pasture 54%/53%, range 0-72%). Their high resistance in**
**agricultural sites is a combination of resident species, but also those that**
**actively forage there but breed elsewhere. While both residents and**
**spillovers from the forest pollinate or disperse seeds, the latter disperse**
**genes among habitats and therefore facilitate seedling recovery. We**
**hypothesize that the high resistance of these taxa accelerates the recovery of**
**fast-growing pioneer trees (Supplementary Table 1) that provide nectar and**
**fruit resources⁴⁹. These resources further attract pollinators and seed-**
**dispersers, leading to a positive feedback loop for fast mutualistic network**
**assembly.**

25. L208. Many trees also recover slowly simply because they occur in low abundances
and dispersal is a chance event. Please mention this as well

*Thanks. We now mention this effect (p.7, l.222-223).*

26. L210. Why do moths, ground birds, and ground mammals recover on a similar time
scale as trees. Why? Because they depend on the trees as a fruit source (I am not sure
how specific this is, as cassowaries eat not only large seeded species but also
opportunistically any plant with fruits)? Or because they are hunted out and sensitive to
fragmentation?

*That's an interesting question, but also difficult to evaluate based on a single*
*chronosequence. Most ground birds and mammals are known to be forest*
*specialists for various reasons, including fruit availability, shelter from predators*
*and hunting or microclimate. After discussion with people that are engaged in the*
*conservation work at our study site, we believe the hunting pressure may be the*
*predominant cause for slow recovery and now added a hypothesis (p.7, l.224-*
*226):*

**Because hunters target ground-dwelling large vertebrates, their slow**
**recovery in species composition may indicate additional disturbances but**
**also more specialized requirements for resources.**

27. L214. Surprising that deadwood beetles is slowed down by deadwood volume, as
there is a lot of thinning and mortality of short-lived pioneer species going on during
succession. Maybe you mean that they need specific tree species or thick stem
diameters?

*Yes, individual dead wood volume matters as well as the total amount – both*
*increase strongly with forest age (as does the living biomass of trees). Thick*
*branches or trunks are key for most deadwood beetles; some are also taxon-*
*specific. However, when resolving the two different land-use legacies, the*
*recovery trajectory of deadwood beetles changes (and actually differs quite a lot*
*between both legacies). In cacao, there is now nearly no difference between*
*agriculture and old-growth forest visible (although with low credibility), while*

*recovery in pasture is fairly fast. We now removed the sentence stating that*
*recovery of deadwood beetles is slowed down.*

28. L215. Surprising that land use did not affect resistance or ‘resilience’. Did you
explicitly model it?

*Yes, we explicitly modelled the influence of land use on resistance and resilience*
*for each taxon. We could not find a statistically significant trend across all taxa,*
*leading to the statement in our initial manuscript. However, as we now report*
*resistance and resilience (now return rate) values for each taxon for both land*
*uses individually, we are now able to show and discuss that land use has an*
*influence for some taxa and can discuss this topic in more depth.*

29. L216., I can not see this in Supplementary Table 2

*The effect of land use is reported in Supplementary Table 2 in rows marked with*
*“Legacy” in the column “Analysis”. We now changed the entry to “Land-*
*use_history” to avoid confusion.*

30. L254-257. Please also say that bats and birds may spill over from the surrounding
agricultural landscape to the OGF

*Thanks. We now include this direction by making the statement broader to include*
*comparisons between habitats in general (p. 11, l.289-290):*

**Their high resistance in agricultural sites is a combination of resident**
**species, but also those that actively forage there but breed elsewhere.**

31. L266-281. If bacteria have a resistance of 82% to 71% (why is the resistance lower at
deeper soil layers, by the way), which means 71-82% similarity with OGF, then I would
not call that an ‘alternative stable state’. This is definitely a case where the glass is more
than half full! 😊

*True. Without knowing the functional importance of these bacteria species, it*
*seems likely that this deficit has no strong consequences. We thus removed the*

second half of the sentence and deleted:
**“in an alternative state that is different from the old-growth forest”.**

32. L270. Surprising that resilience and resistance are not correlated across taxa! Please
come up with a reason why this would be the case

*We agree. While resistance and resilience (now return rate) are mathematically*
*independent, logical drivers behind the variation in resistance and return rate can*
*differ or be similar and lead to correlations between both, as for example found in*
*Poorter et al. (2021) where resistance and return rate were found to correlate. For*
*instance, climate tolerance may limit the resistance of a taxon, dispersal abilities*
*their return rate. If there are trade-offs between these traits, a negative*
*relationship between resistance and return rate may be expected across taxa. If*
*similar parameters (e.g. proximity to forest) affect both resistance and return rate,*
*they might be positively related. We now include a hypothesis, why resistance and*
*return rates may be uncorrelated in our study (p. 12, l.314-318):*

**We hypothesize that return rates are driven by a multiplicative effect of a**
**species’ mobility and the landscape context (e.g. forest cover in the**
**surrounding), while resistance is related to factors of land-use itself, such as**
**land-use intensity and duration. Dissecting resistance and return rates**
**across many distinct taxa is thus fundamental to understand the patterns of**
**tropical forest recovery.**

33. L314-328. This discussion could be much stronger. Please try, as your results are so
cool

*Thanks, we revised the discussion and tried to strengthen it.*

34. L330-347. Your conclusions should be way more stronger for Nature. Please try

*We thank the reviewer for pointing out that our conclusion needs improvement.*
*We significantly changed the conclusions section of our manuscript to point out*
*the main findings and increase the discussion of the applicability of our results*
*and put them in a broader context.*

35. L335-337. Unclear

*We removed this sentence, as we did not deem it necessary anymore.*

36. L338-342 I do not understand

*We changed that section and hope it is now easier to understand.*

37. L344. Replace biodiversity by composition

*We now significantly rewrote the conclusion, so this sentence does not exist*
*anymore.*

38. L345-347. This is a bit thin, and strange to end your conclusions with it, as you did not
test for the role of corridors, remnant trees ‘or other elements’ (what do you mean?)

*We agree that this ending of our conclusions needed revision. We generally*
*rewrote the conclusions in many parts (accordingly to minor comment 34) and*
*now write at the end of the conclusion (p.15, l.387-389):*

**Overall, our results underline that cost-effective natural regeneration via**
**abandonment of agricultural land is a powerful restoration strategy for large**
**areas of tropical landscapes with small-holder agriculture to meet the UN**
**Decade on Ecosystem Restoration goals.**

39. L496. What is the Shannon diversity. Is this the traditional Shannon diversity or the
Hill 1 diversity (i.e., effective number of species)

*With “Shannon diversity” we always refer to the Hill-number of order $q=1$ which is*
*the exponential Shannon entropy (effective number of species) and normally*
*referred to simply as Shannon diversity. We now clarified this by writing (p.21,*
*l.603-605):*

**For all taxa and plots we calculated the total abundance (number of**

**individuals of each taxon) per plot and alpha-diversity Hill-numbers (i.e.**
**effective number of species) for the orders $q=0, 1$ and 2 (species diversity,**
**Shannon diversity and Simpson diversity).**

*We also included upon the first mentioning of the word diversity in the main text*
*(p.6, l.189-190):*

**“Abundance and diversity (Shannon diversity, i.e. Hill number of order 1) of**
**many taxa remained high even during agricultural use....”**

40. L499. Why did you have to transform to relative abundances?

*If Bray-Curtis distances are computed on raw abundance data, two plots with the*
*same species assemblage, and each species occurring in the same proportion,*
*may have a huge distance value - if they differ only in total abundances of all*
*species (or total number of reads etc.). This is an undesirable property. Other*
*alpha- and beta-diversity indices such as those based on Hill numbers are directly*
*defined for proportions (e.g. Shannon or Simpson diversity equations and*
*derivatives for beta-diversity), but the commonly used Bray-Curtis, Euclidean*
*distance and others are not. This normalization (i.e. standardization as relative*
*abundances, or relative amount) is commonly applied to any distance metrics*
*that aim to express the distance in (relative) composition rather than in the totals*
*(the latter can be tested by univariate statistics).*

*We now write (p.21, l.614-616):*

**For Bray-Curtis similarity, species abundances were first transformed to**
**relative abundances for increasing robustness against differences in total**
**abundances among plots.**

41. L508. What is the average coverage in your dataset (mean and range)? It feels more
accurate if you use that one rather than the minimum coverage of the extrapolated data
to twice the plot size. Please explain and justify

*We now included the mean coverage and the range in Extended Data Table 1. We*
*extrapolated the data of each sampled plot to twice the observed number of*
*individuals, as this is the value at which diversity can still be accurately predicted.*
*We then noted the coverage value at this extrapolated point for each sampled*

*plot. The minimum coverage value among the extrapolated samples was then*
*used to standardize all samples to this coverage value. We thereby ensure that no*
*plot is extrapolated beyond the coverage that can still be accurately predicted.*
*We thereby follow the procedure recommended by our co-author Anne Chao who*
*developed the standardization techniques we used. The rationale behind this is*
*explained in more depth in Chao et al. (2014) and Chao et al. (2023). However, we*
*also added a better justification (p.22, l.628-629):*

**We thereby ensure that no plot is extrapolated beyond a point where**
**predictions are still reliable.**

42. L498. Can you explain exactly what it is. I thought it was Bray-Curtis DISSimilarity???

*Thanks for pointing this out. Yes, we indeed calculated the Bray-Curtis*
*dissimilarity using the package vegan and subsequently calculated 1-Bray-Curtis*
*dissimilarity to obtain the Bray-Curtis similarity. We corrected this in the*
*description (p.21, l.616-618).*

43. L512. Twice the size of what? Species? Number of individuals? Unclear why you do
this. Please justify

*We extrapolate each plot to twice the number of individuals that has been*
*sampled, as this is the point at which extrapolation can still be accurately done.*
*We then look at the lowest coverage value among all plots at this extrapolated*
*point. We then use this coverage value as the point to which all plots are*
*extrapolated, as this is the point at which diversity can be accurately predicted for*
*all plots, including the one with the lowest coverage. We thereby follow the*
*methodology proposed by our co-author Anne Chao in Chao et al. (2014) and*
*Chao et al. (2023). We now clarified (p.22, l.624-626):*

**For each dataset we first extrapolated the Hill-numbers of each sampled plot**
**to twice the observed number of individuals, as this is the value at which**
**diversity can still be accurately predicted^{55,56}.**

44. L516-520. I do not understand. Please provide for poor people like me a definition of

beta diversity. How can you extrapolate beta diversity if that is calculated as similarity in
species composition?

*For standardization of beta-diversity we follow the methodology described in*
*Chao et al. (2023). The beta-diversity of order q (with q being the weight of species'*
*relative abundances in the framework of Hill-numbers) is there defined as the*
*fraction of gamma and alpha-diversity according to Whittaker (1972):*

$${}^qD_{\beta} = {}^qD_{\gamma} / {}^qD_{\alpha}$$

*Thereby, the extrapolation of beta-diversity follows from the extrapolation of the*
*gamma-diversity and alpha-diversity. The beta-diversity is related to measures of*
*community similarity through transformations (Jost, 2007). It is shown there that*
*the community overlap of order q of 2 communities (pairwise comparison of two*
*plots) is given by:*

$$\text{overlap (of order } q) \equiv \frac{({}^qD_{\beta})^{q-1} - (1/2)^{q-1}}{1 - (1/2)^{q-1}}.$$

*For $q=1$ this formula does not converge and therefore for q tends to 1 the overlap*
*is given by:*

$$\text{overlap of order } 1 = (\ln 2 - H_{\beta \text{ Shan}}) / \ln 2$$

*With H_{β} , Shan the Shannon-index. As derived in Jost, 2006, the above*
*formulas give for $q=0$ the Sorensen-index, for $q=1$ the Horn-index and for $q=2$ the*
*Morisita-Horn-index.*

*The standardized overlap is then calculated from the standardized beta-diversity.*
*We now clarified (p.22, l.610-614):*

**We calculated the pairwise Bray-Curtis similarity as well as the beta-diversity**
**Hill-numbers for orders $q=0, 1$ and 2 (calculated as the fraction of gamma- and**
**alpha-diversity Hill-numbers of orders $q=0, 1$ and 2) which we subsequently**
**transformed to measures of species composition similarity (Jaccard-index,**
**Horn-index and Morisita-Horn-index) following refs. 52 and 53.**

45. L535-539. Please test and report the differences between pasture and coca in

resistance and recovery for all your taxonomic groups and your 3 attributes.

*Differences between pasture and cacao in all calculated metrics are given in*
*Supplementary Tables 1 and 3. These tables are quite large in size and*
*unfortunately it is not possible to present them in the main text or as an extended*
*data table due to size restrictions. However, we now report the exact values for*
*cacao and pasture for species composition, which we think are most important,*
*in the newly created Table 1 including the respective confidence intervals.*
*Significant differences can be inferred from confidence intervals. An additional*
*test of a statistically significant different difference across all taxa was performed*
*and results are given in Supplementary Table 2. Again, due to size constraints, this*
*Table was put into the supplements rather than the main text.*

46. L545 Say that this is, in fact, a measure of similarity to OGF values. Explain above why
values that overshoot OGF values (more biodiversity than in OGF) are not ‘good’.

*Thanks for pointing this out. We now write (p.22-23, l.656-662):*

**In some instances, the median value of an ecological metric of the active**
**agriculture plots had a similar or even higher level than the median value of**
**the old-growth forest plots. This was found for abundance or diversity of some**
**taxa, confirming their high number of disturbance-tolerant species and**
**species that can utilize a recovering habitat already early during succession.**
**However, abundance and diversity levels above the old-growth reference can**
**still be considered a disturbance effect.**

47. L548. Say that this similarity index is not completely symmetrical around 1

*Thanks, we now mention this (p.23, l. 672-673).*

48. L555. Why can’t you use an OGF value of zero? Please explain

*We opted for not accepting OGF median values of zero because a value of 0*
*means there is nothing there (with respect to abundance, diversity or*

composition) in the reference and we deemed a recovery back to nothing is not
meaningful. We then switched to using the mean value, which is less robust but at
least provides a non-zero value to which recovery can happen. We explain this
now in the methods section as (p.23, l.678-680):
**We chose to use the median value as default as the median is more robust to**
**outliers, unless the median of the old-growth forest was 0. In these cases we**
**chose to use the mean value for all calculations because we deemed a**
**recovery to a value of 0 to not be biologically meaningful.**

49. L566-567. I do not understand

*The recovery can happen from “below” or “above”, depending on whether the*
*values in the actively used agriculture plots were higher or lower than reference*
*values. In both cases we accepted a deviation of 10% from the reference*
*condition values as the point of full recovery. We now write (p.24, l.692-700):*

**As the initial active level could be higher or lower than the level of old-growth**
**forests the function given by Eq. 2 could asymptotically reach the old-growth**
**level from higher or lower values. We therefore solved Eq. 2 for $\psi(t) = 0.9 \cdot \psi_{Ref}$**
**(for the case when recovery happens from lower than reference values) and**
**$\psi(t) = 1.1 \cdot \psi_{Ref}$, (for the case when recovery happens from higher than**
**reference values) to allow a tolerance limit of $\pm 10\%$. For the majority of cases,**
**recovery happened from lower than reference values for which the above**
**condition then relates to the recovery time to 90% of the total value of the old-**
**growth forest reference. This value was chosen (in line with ref. 4) because it**
**allows to account for natural variation within old-growth forests but is still**
**close enough to assume full recovery.**

50. L580. Why do you use SD rather than 95% confidence interval?

*The 95% confidence interval was calculated for the recovery trajectories which*
*includes all plots. However, in the analysis mentioned there we only wanted the*
*information whether active agriculture plots and old-growth forest plots were*
*significantly different from each other and therefore compared only the means*

*and standard deviations of the active agriculture plots and the old-growth forest*
*plots.*

51. L575. Apparently it is recovery time until 0% of OGF values. Please say so explicitly in
your abstract and main text, and justify why

*Thanks. We now explicitly mention in the methods and in the main text that*
*recovery time was calculated until 90% of old-growth forest values.*

52. L591. Please show in the appendix the pairwise scatterplots for these 3 relationships

*As we now only report the parameter lambda as the “return rate” we do not show*
*the pairwise scatterplots between lambda and the initial calculation of the*
*resilience.*

53. L594. See above; it does not make sense to report an average rate as recovery
changes non-linearly over time

*We agree. We now only report the parameter lambda that describes the non-linear*
*behaviour of the recovery process.*

54. L616. Why do you use random forest rather than a simple multiple regression or a
variance partitioning analysis?

*The main reason for using Random Forest is that we do not have to worry about*
*how to represent non-linearity of the effects of resistance and resilience, and that*
*it has a clear way of interpreting the contributions of interacting predictors. If*
*resistance and return rates interact in a GLM, it is unclear how much variance*
*should be attributed to either. We now added (p.26, l.757-759):*

**This approach has the advantage that non-linear effects of resistance and**
**return rates are directly incorporated in the analysis and it has a clear way of**
**interpreting the contributions of interacting predictors.**

55. L618. Explain why you use impurity

*Impurity is the default measure of importance. We now write (p.26, 761-762):*

**We used “impurity” as importance measure as it is the default way to analyse**

**feature importance.**

56. L619-621. Drawing random value for resilience if the resistance was greater than 90%
does not make sense to me. I think it means you simply can not estimate it, and I would
omit this value

*We thank the reviewer for pointing this out. We now omitted the values and*
*calculated the importance with random forest without these values. We wrote*
*(p.26, l.763-765):*

**For taxa with a resistance greater than 90%, return rates could not be**
**calculated as 90% represents the threshold for full recovery in our study. We**
**thus omitted these datapoints in further analyses.**

57. L645 provide a name for group 2

*Thanks. Corrected.*

L648 provide a name for group 4

*Thanks. Corrected.*

58. Fig 1. Nice illustrative figure!

*We thank the reviewer for this positive comment on our figure.*

59. Fig 2. Nice illustration, although it is difficult to compare the values and assess the
range of the confidence interval (see my comment above). At least put more tickmarks
on the x axis, so that it is easier to read. What are blue/orange/green dots/

*We thank the reviewer for this positive comment. We included more tickmarks.*

*The different colors just indicate different metrics (abundance, diversity,*
*composition). We clarified this in the figure legend.*

60. Fig 4. Why is there a decline in the grey zone, and not an instantaneous decline? Why
do your red line stay after the vertical dotted line for some time flat whereas in your Fig 3
recovery starts instantaneously? Please use normal scale for resistance, and not a

square root scale. Aldo for resilience. 95% CI of the mean? What are red, blue, green
dots?

*We thank the reviewer for these comments on Fig. 4. The decline in the grey zone*
*has aesthetic reasons and aims at making the understanding of the decline of the*
*system attribute during the perturbation easier. The sigmoidal shape of the red*
*curve after recovery also had aesthetic reasons, but we agree it is misleading as*
*the function we fit to the data indeed is not sigmoidal but the recovery starts*
*instantaneously. We now used the correct function to depict the different*
*scenarios. We also now use a normal scale for resistance, but for the resilience*
*(now return rate) we use a logarithmic scale now, as otherwise values were*
*difficult to assess in the graph. 95% confidence intervals were given for the fitted*
*value. The colored dots indicate the taxa belonging to the four different scenarios*
*shown in the pictograms. We clarified this in the figure legend and furthermore put*
*coloured boxed around the scenarios for clarification.*

Referee #3 (Remarks to the Author):

This manuscript aims to evaluate recovery times of different taxa after disturbance along
a successional gradient using chronosequence data. This is an important question, as
understanding how different groups of organisms recover and colonize a tropical forests
after land use may bring key insights for landscape management. Additionally, the data
set is impressive and reflects a substantial amount of field and lab work. I appreciate the
effort of integrating multi-taxa datasets and combining different statistical methods to
address this issue.

However, I have many reservations about the manuscript. Mainly, the concepts
mentioned are not explored in sufficient depth, and the authors do not appear to be fully
familiar with some of the topics discussed. Below, I provide some general comments.

1. The concept of resilience.

I was surprised by the definition of the term resilience provided here, and the way
recovery, resilience and resistance were linked. There is a large body of literature
highlighting that resilience has been a concept difficult to grasp, yet many papers provide
clear definitions. This is the first time I see recovery defined as the combination of
resilience and resistance.

I checked the citations provided in line 130 and none of them defines recovery as the
combination of resilience and resistance. In the Holling book, as in many other
publications by SR Carpenter (e.g. Ives & Carpenter, Science 2007; Peterson et al.,
Ecosystems, 1998), resilience is the most overarching concept from which other related
concepts, such as recovery and resistance, are derived. In Poorter et al. 2021 (Science)
resilience is defined as “ability of a system to absorb disturbances and return to its
previous state” and encompasses both recovery and resistance. I also recommend
reading Rodrigo Muñoz et al. J Ecol. 2021

In this sense, the manuscript is very confusing. Indeed I felt that the term recovery was
somewhat polysemic, making the manuscript very difficult to follow. For instance, lines
128-130 mention that recovery can be partitioned into resistance and resilience, but Fig
1 is titled “Two dimensions of stability”, meaning resistance and resilience, which
suggests that stability is synonym of recovery. In panel b of Figure 1, there is an equation
given to define recovery time, but in Figure 3, one can see the “recovery trajectories” of
species composition and this does not seem to be calculated based on the equation in
Fig 1, so that recovery trajectories is different from recovery times.

*We thank the reviewer for pointing out this ambiguity in our text. Indeed, we agree*
*that the concepts in our paper need further clarification and more elaborate*
*definition because a variety of definitions and concepts of resilience exist (a point*
*raised by reviewer #2 as well, see also our answer to major comment 6 of reviewer*
*#2).*

*We agree that the connection between resilience, resistance and recovery was*
*misleading, as resilience is in many works defined through the contribution of*
*resistance and recovery, e.g. in Poorter et al. (2021), Ingrisch & Bahn (2018). In our*
*work, we follow the definition that sees resilience as speed of return to reference*

conditions related to the “engineering resilience” concept of Holling (Holling,
1996, Pimm, 1984) that is widely followed. We define resilience as the return rate
from the baseline (i.e. the level defining resistance) following calls to disentangle
resistance and resilience (see e.g. Pimm, 1984, Pimm et al., 2019, Justus, 2007,
Van Meerbeek et al., 2021, Hillebrand et al., 2018, White et al., 2020) to identify
their individual contributions to short or long recovery times. We clarified this by
writing in the main text (p.4, l.129-135):

**The recovery trajectories and recovery times of different components of**
**tropical forests (e.g. the species composition of a certain taxon) depend on**
**resistance - the ability to withstand disturbance - and the return rate to**
**reference conditions after perturbation (Fig. 1)^{25,26}. Resistance is related to**
**attributes that confer tolerance to perturbation such as physiological or**
**behavioral plasticity within taxa or features that provide protection against**
**change^{25,27,28}. Return rates provide a measure of resilience (in the sense of**
**speed of return to reference conditions^{29,30}) already standardized with the**
**amount of change caused by the disturbance³¹⁻³³.**

*However, to avoid confusion with different definitions of resilience in the literature*
*and increase the readability of the text we now simply refer to the return rate*
*lambda in the revised manuscript whenever we refer to that specific measure and*
*only use the term resilience in the introduction, when referring to the more general*
*framework of our study. We also improved our definitions in the methods to clarify*
*the reasoning behind our choice of definition. We now write (p.24, l.703-710):*

**Resilience is often either defined as the ability to maintain a qualitatively**
**similar state in the face of disturbance and not switch to an alternative state**
**(ecological resilience^{26,59}) or the speed of return to reference conditions**
**(engineering resilience^{25,26}). Here, we follow the engineering resilience**
**concept and define the return rate λ from Eq. 2 (see also Fig. 1) as a measure**
**of resilience standardized by the amount of change caused by the**
**disturbance^{25,27-29}. However, to avoid confusion with other definitions of**
**resilience in the literature, throughout the manuscript we use the term**
**“return rate” for the specific measure represented by λ , and “recovery time”**
**for describing the time until 90% of the reference value is reached.**

In the original version of our manuscript we also transformed the actual return rate λ used also in Poorter et al. (2021), which is part of Eq. 2 visible in Fig. 3, to a linear return rate (depicted in the initial version of Fig. 1). We did so to obtain a measure of percentage recovery per year which we thought would be easier to interpret. We however agree that this was confusing and oversimplifying the reality of non-linear recovery (see also major comment 6 of reviewer #2). We now omitted this linearization and now simply report the return rate λ , acknowledging the non-linear behaviour of the recovery trajectory. We also now include Eq. 2, which was used to calculate recovery times, rates and trajectories, in the revised version of Fig. 1. Thereby, Fig. 1 and Fig. 3 should align much better and we hope the concepts we used are easier to understand. We provide improved definitions in the methods section. We hope this clarifies our methodology significantly. We thank the reviewer again for these comments, and hope all our definitions are now clearer and easier to understand.

2. Questions and hypotheses

The way the results are presented is somewhat descriptive and I felt the lack of a theoretical framework integrating the questions addressed. Specifically, there is not a conceptual basis to evaluate variation in recovery times across taxa exhibiting different life-history strategies, dispersal modes and trophic levels. What do we know about this? There no mention in the introduction of any previous work addressing these issues. In particular, the question “whether differences in recovery times (...) can be explained by simple rules” is weak. The authors expect taxa with the r strategy to recover faster than those with the K strategy based on the fact that r-species exhibit higher reproductive rates. However, there is no explanation on why having lower age at first reproduction may allow species (or taxa) to show a faster recovery (or resilience or resistance). Although the framework of r-K selection continuum has been important to globally understand life history strategies, it is clearly an oversimplification of nature and many species do not fall along this single dimension of variation. Only in tree ecology, recent work has shown that there are many axes of variation in demographic rates, mainly the growth-survival

and the stature recruitment trade-offs (Rüger et al, 2020, Science 368), and there are
other important trade-offs in successional theory such as the colonization-competition
trade-off. All these do not necessarily align with the fast-slow continuum mentioned.

The classification of taxa into life-history strategies and dispersal modes was too broad.
Within each realm (bacteria, invertebrates, vertebrates and trees) there are species
exhibiting a wide array of life-histories so assigning all bacteria the strategy r and all trees
the K strategy presents an overly simplistic view of nature. Also, assigning an aerial
dispersal to all plants, stating that all plant species are virtually dispersed by birds, bats
and wind is also an oversimplification. There is a wide variation in plant dispersal
abilities, and my guess is that the same happens for each taxa. These categories are too
large and thereby inconsequential. Given such reductionism, I'm not surprised that there
are no significant results among categories in the time of recovery. Overall, I had the
sensation throughout the manuscript that the complexity of ecological process is
somewhat disregarded and that this study does not sufficiently incorporate previous and
more recent research.

*We agree that the introduction of the analysis of variation among taxa in recovery*
*times, resistance and return rates could better incorporate previous work. While*
*the conceptual framework of disentangling recovery times into contributions of*
*resistance and return rates provides, in our opinion, already a step forward into*
*mechanistic insights of the recovery process of several taxa (also highlighted by*
*reviewer #1 and #2), we agree that our analysis of ecological differences leading*
*to variation among taxa needs also a better motivation. We thus extended the*
*introduction as follows (p.4, l.135-139):*

**High return rates are related to low trophic levels³⁴ and life-history strategies**
**that allow swift recovery after a perturbation, such as rapid recolonization or**
**re-growth^{25,35,36}, rapid reproduction with large numbers of offspring, early**
**reproductive age and short generation time³⁷⁻⁴¹.**

*We also thank the reviewer for suggesting several meaningful alternatives to the r-*
*K selection continuum to predict recovery times, e.g. the "growth-survival and the*
*stature recruitment trade-offs" (Rüger et al., 2020), and "colonization-competition*

*trade-off”, and for highlighting that there is a high variation between species within*
*each taxon. Indeed, these two trade-off characteristics would be a very valuable*
*way to understand successional dynamics, and for trees such an analysis is*
*underway. However, in this study, we opted to restrict our analysis to life history*
*strategies that can be assigned to all taxa including plants, invertebrates,*
*vertebrates and bacteria to explain the order of recovery times across taxa. The*
*two trade-offs suggested by the reviewer would be more operable within each*
*taxon than across a broad spectrum of different lifeforms as investigated here.*
*Age at first reproduction was chosen in this study as one simple way towards this*
*goal, but arguably does not cover all aspects of reproduction during adulthood.*
*We think that a convincing general framework of economic spectrum and other*
*life history traits across life-forms is still in its infancy, and works best within taxon,*
*particularly for plants. However, the slow-fast continuum investigated here has*
*been shown to be a very valuable way to understand recovery dynamics and the*
*response of organism groups to disturbances (see e.g. Capdevila et al., 2022, Li et*
*al., 2021) and the assumption that short recovery times are driven by fast*
*population dynamics has been in the ecological literature for a long time (see e.g.*
*Pimm, 1984) but empirical tests of this hypothesis were so far rare. A test of*
*ecological differences such as the slow-fast continuum and different trophic*
*levels explaining the order of recovery time, especially across multiple taxa within*
*a general analytical framework, is in our opinion, a large step forward. Our finding*
*that these simple mechanisms do not explain recovery times, contrary to*
*expectations, is thereby particularly surprising (as pointed out by Reviewers #1*
*and #2). Reviewer #2 also highlighted among the strengths of the study “the*
*surprising result that some short-lived taxonomic groups show arrested*
*succession, and that simple ecological attributes cannot explain resistance or*
*recovery.” We also think that the broad classification chosen here provides a*
*reasonable first approximation, because, while there for sure is within-taxon*
*variation, the variation between taxa should be large enough that our broad*
*classification is able to resolve differences. Additionally, we also did split 5 groups*
*(frogs, trees, birds, bees and saproxylic beetles) into fast- and slow-reproducing*
*species for a more detailed analysis to acknowledge variation in life-history*

*strategies within taxa. However, this analysis did not reveal a significant*
*difference, further strengthening the finding that fast or slow reproduction does*
*not explain variation in recovery times, resistance or return rates among taxa.*

*We now acknowledge the limitations of our approach and point towards the need*
*for more detailed work in the future by writing (p.14, l.367-370):*

**However, a more fine-grained classification of species' life-history strategies**
**also within taxa and including other trade-offs might reveal further insights.**
**We suggest that resolving the temporal sequence of the order of species'**
**establishment both within and across taxa is a promising way to understand**
**functional ecosystem recovery.**

3. Successional data

3.1 There are many issues that are not clear in the data and the methods used.

First, there not enough information on the plot data. Of the 62 plots, how many are active
cacao plantations, pastures (6 and 6)? Which are the stand ages of the secondary plots?
Many studies have addressed the shortcomings of chronosequence data and the
importance to have replication within each age class. There is no information on the
number of plots for each class age, or on the size of the plots, which also matters a lot.

*We thank the reviewer for pointing out the lack of sufficient explanation of our*
*study site which is a point also raised by reviewer #2 (see major comment 4). We*
*previously referred to another work (Escobar et al., 2025) which provides a*
*detailed description. However, we agree that this paper, as a stand-alone work,*
*needs sufficient description of the study site. We therefore included a better*
*description of the study site in the methods section. We additionally improved the*
*description of the study site in the main text and now specifically feature the*
*number of plots for each land use legacy and the size of the plots. The number of*
*replicates for active agriculture (6 cacao/6 pasture), secondary forests (16*
*pasture/17 cacao) and old-growth forests (17) is high and the total investigated*
*area is also exceptionally high compared to many other regions (see Fig. 4 below*
*in the answer to comment 5). Detailed information on each plot is supplied in*

*Supplementary Table 6.*

*We now write in the main text (p.5, l.170-173):*

**Our 62 study plots are each 0.25 ha in size and include 6 actively used**
**pastures and 6 cacao plantations, 33 secondary forests of variable age (1–38**
**years) recovering from previous agricultural use (16 pasture/17 cacao), and**
**17 old-growth forests as a reference (see Supplementary Table 6 for details).**

*We also included a much more detailed explanation and justification of the*
*chronosequence approach in the methods section (Methods: Chronosequence)*
*in line with major comment 4 of reviewer #2.*

3.2 Also, there is no consideration of the importance of prior land use, see Robin
Chazdon and Catarina Jakovac papers on this (eg. Chazdon 2003, Pers. Plant Ecol. Evol,
Chazdon et al. 2009, Phil. Trans Royal Society. B; Jakovac et al. 2021, Biol. Rev.). Norden
et al. 2015 (PNAS) also shows how variable are successional trajectories are based on
dynamic data. Figure 3 seems to combine everything, so that you are probably comparing
apples and oranges. Drawing conclusions from this seems unreliable to me.

*We thank the reviewer and agree that prior land use can be important as pointed*
*out in the named references. This is a major reason why our study design explicitly*
*differentiates between plots of cacao and pasture legacy and has equal shares of*
*plots of both legacies. While in the previous version of our manuscript we opted*
*for keeping the results rather simple, we agree that resolving differences in*
*organism groups for different land-use legacies is a benefit of our study which*
*justifies explicit presentation in the main text and not only in the supplementary*
*tables. We thus now (also in agreement with major comment 3 of reviewer #2)*
*include both types of land use legacy in Figure 2,3 and 4 in the main text and*
*Extended Data Figs. 1-5 and present the underlying results in Table 1. We*
*additionally include a more sophisticated discussion of the effect of land use on*
*the recovery trajectories of the investigated taxa. We now write in the introduction*
*(p.6, l.178-179):*

**Return rates often depend on the legacies of past land use^{46,47}. We therefore**
**compare the legacies of cacao and pasture, which are the two primary land**

**uses in the region.**

*We also write in the main text (p.7, l.227-237):*

**Recovery times of bacteria, saproxylic beetles, dung beetles, bees, ants,**
**ground birds, birds, bats and ground mammals, as well as tree seedlings were**
**shorter in former cacao plantations than in pastures (see confidence**
**intervals in Table 1 and Fig. 3 and Extended Data Fig. 1, 2 and 3), possibly**
**facilitated by a combination of more favorable abiotic conditions (e.g. shade**
**and humidity) and available resources. Recovery times of trees and few other**
**taxa showed no consistent response to land-use legacy. For trees, this finding**
**is in line with recent findings across multiple chronosequences that also did**
**not identify an effect of previous land use¹⁷. We suggest that legacies have a**
**stronger impact on animal than tree communities, as abandoned cacao**
**plantations provide more available resources than pastures, thereby**
**facilitating the recovery of diverse animal groups. For trees, however, we**
**hypothesize that cacao trees may limit rather than facilitate key resources**
**such as light and nutrients.**

4. Results presentation

There is not enough description of the analyses performed in the main text. The main text
should be self-contained, that is, it shouldn't be necessary to go to the Methods section
to understand the results. Mainly, there is a lot of information missing (see point 3) and
the figures do not seem to be supported on data. I was particularly puzzled by Figure 2
which is very pretty, yet does not show the data underlying these final results, and the
reader needs to go over all the Methods section to understand which data and analyses
support this finding.

Also, there are large sections describing results with any reference to a statistical
analyses (eg. lines 196-214).

*We thank the reviewer for pointing out these shortcomings in the presentation of*
*results in the main text. In line also with the answer to major comment 2 made by*
*reviewer #2, we now include a table that details the specific recovery times,*
*resistance, return rates and the newly calculated relative recovery after 30 years*

*(including confidence intervals) separately for plots with cacao or pasture legacy*
*for the species composition. We hope this clarification of the exact results of*
*species composition, which we believe is the most important and enlightening of*
*the measures we studied, improves the usefulness of the results that are*
*presented in the main text. Data for other metrics are given in Supplementary*
*Tables 1 and 3. Both tables are quite extensive, especially when considering the*
*separate legacies for cacao and pasture history. The size of these tables doesn't*
*allow all results to be featured in the main text in a concise way apart from the*
*presentation in Fig. 2. However, we now included more tickmarks in Fig. 2 to*
*improve readability of the values. Additionally, we improved Fig. 1 and its caption,*
*which presents a schematic and a description of how the analysis was performed.*
*As we now omitted the linearization of the calculation of the return rate, Fig. 1 and*
*Fig. 3 should align much better and are hopefully easier to understand. We also*
*added a concise description of the methodology in the main text (p.6, l.175-178):*
**The recovery trajectory was modeled with a negative exponential function**
**(see Fig. 1 a and Eq. 2 in the methods) with the return rate λ . It assumes a non-**
**linear recovery and an asymptotical approach to old-growth forest conditions**
**with increasing time since land abandonment.**
*We also included more references to statistical tests, whenever appropriate, e.g.*
*(p.11-12, l.311-313):*
**Our results further show that resistance and return rates were not correlated**
**(Spearman's rank correlation cacao/pasture: abundance: $\rho=0.24/0.06$,**
**$p=0.45/0.84$; diversity: $\rho=0.23/0.35$, $p=0.5/0.24$, composition: $\rho=-0.41/-0.06$,**
**$p=0.14/0.84$; Supplementary Table 9).**
*We hope these changes made to the main text and figures improves readability*
*significantly and it is now possible to understand all results and how they were*
*derived from reading only the main text.*

5. Impact and generalization

Although I do recognize that this study relies on an extraordinary amount of data, I don't
think that the recovery times reported here can be generalized. Overall, these findings
are likely to be anecdotal for this study site. Moreover, as the authors mention, these

data were collected in a relatively well conserved landscape, which is not representative
of what is happening in most the tropics. The study of Rozendaal et al., for example, was
based on chronosequence data from 56 sites (over 1500 plots) distributed in 10
countries and land use was controlled for in each chronosequence dataset. Additionally,
based on my understanding of the manuscript, the estimates of recovery times,
resistance and resilience were derived from calculations that violate the basic
assumptions of chronosequence as they do not distinguish land use (not only separating
active cacao plantations from pastures, but also distinguishing successional forests
depending upon their previous land use). Overall, these shortcomings considerably limit
the reliability and impact of the results presented in the manuscript.

*We did consider the influence of land-use in the study design and analysis,*
*because each active agriculture was classified as active cacao or pasture and*
*each secondary forest plot as recovering cacao or recovering pasture. We also*
*performed each analysis individually for cacao and pasture legacy only. For*
*simplification purposes, we only presented the results in the main text in which*
*both land uses were pooled in the initial version of our manuscript and presented*
*the results in Supplementary Table 1 and 3. However, we thank the reviewer for*
*pointing out that this was an oversimplification (a criticism shared with reviewer*
*#2 in major comment 3). We now consequently report the results of both land-use*
*legacies individually in Fig. 2, 3 and 4 in the main text and Extended Data Fig. 1-5,*
*so it is possible to see, for each taxon individually, how they recover from different*
*land-uses. As stated in the answer to comment 4, we furthermore included a table*
*(Table 1) in the main text that shows the resistance, return rates, recovery time*
*and the percentage relative recovery after 30 years values for cacao and pasture*
*land-use legacy. We also included a more elaborate discussion of the land-use*
*effects on different taxa in the main text (see answer to comment 3.2). We hope*
*this increases the generalizability and impact of our results.*

*Quite generally, multi-taxon studies such as ours come with an extensive amount*
*of field work (as recognized by the reviewer) and subsequent data analysis, which*
*already requires a substantial infrastructure and funding for each site, making a*
*repetition across multiple sites challenging. This is often more feasible if only a*
*single group is studied as for example in Rozendaal et al. (2019) who studied trees.*

*The systematic analytical framework and plot design were chosen such that the*
*results across taxa are most comparable, increasing the generalizability of the*
*findings made here (a point also raised by reviewer #2 as a major benefit of our*
*study). Additionally, we compared our results with a substantial amount of*
*literature data from all across the tropics, which supported the main findings and*
*thereby also increased the generalizability of the findings of our study.*

*Our study furthermore provides a large plot area that was studied for both*
*regenerating and old-growth forest, increasing the robustness of our results. The*
*recovery time to 90% of the old-growth forest values for species richness of trees*
*in Poorter et al. (2021) is reported to be 37 years across large parts of the tropics,*
*while we report a value of approximately 50 years. These recovery times are thus*
*on a very similar time-scale.*

*With respect to the representativeness of our study site, Escobar et al. (2025)*
*(<https://esajournals.onlinelibrary.wiley.com/doi/full/10.1002/ecs2.70157>)*

*compared the chronosequence presented here with the chronosequences in*
*Rozendaal et al. (2019) and found that the forest cover at our study site*
*(74% ± 2.8% within a 1-km radius) is quite comparable or even less than at many*
*other sites in the neotropics (median 86.3% calculated from Rozendaal et al.,*
*2019), and the landscape thus is representative at least for the neotropics (see*
*figure below, red indicates this study). We now feature this in the introduction and*
*write (p.5, l.167-169):*

**The landscape consists of a mosaic of secondary forest, old-growth forest**
**and agriculture with relatively high forest cover (~75%⁴⁵) that is representative**
**for many neotropical regions (median ~85% of 56 sites¹⁷).**

[Figure Redacted]

[Redacted]

[Redacted]

[Redacted]

[Redacted]

[Redacted]

6. Statistical methods

The authors made a huge effort to write a M&M section as clear as possible, and provide
examples that illustrate the different challenges then faced when dealing with such big
and heterogenous datasets. I am not up to date in quantitative analyses, so I'm not able
to provide a through revision of the statistical methods used. There are however, a few
points that were not clear to me:

*We thank the reviewer for pointing out sources of unclarity in our methodology,*
*and we address the questions in the following point by point.*

6.1 Abundance refers to the total number of individuals (or sequences) for an entire taxa?

How is this number comparable across taxa?

*Exactly, the abundance is the summed-up number of individuals per plot for each*
*taxon. This number is not comparable across taxa, due to the variety of methods*
*used. However, this number also does not need to be comparable, as we only*

calculate recovery times, resistance and return rates within each taxon. The
abundance just needs to be comparable within each taxon across all plots, which
it is due to the use of a standardized methodology for each taxon. We added a
better explanation of the abundance in the methods as (p.21, l.603-605):

**For all taxa and plots we calculated the total abundance (number of**
**individuals of each taxon) per plot and alpha-diversity Hill-numbers (i.e.**
**effective number of species) for the orders $q=0, 1$ and 2 (species diversity,**
**Shannon diversity and Simpson diversity).**

6.2 Even if you control by sampling coverage using Chao's methods, I don't think you can
extrapolate metrics related with species composition to larger plot sizes (diversity,
species composition, beta diversity; line 519). Only metrics such as stem density or
biomass increase in a linear trend.

*The rarefaction/extrapolation methods – including the modifications in the script*
*provided by Anne Chao who is a coauthor – were not mainly employed to provide*
*real values for larger spatial units such as km² grids, but to make our 50x50m plots*
*with variation in sample coverage more comparable and to correct potential*
*biases by unequal sample coverage. The methodologies for standardization of*
*alpha-diversity (see Chao et al., 2014) and beta-diversity (see Chao et al., 2023)*
*do not assume that extrapolation happens in a linear trend but the estimates*
*are based on species accumulation curves. While the standardization techniques*
*we used represent the current state-of-the-art, we agree that the prediction of*
*species composition and beta-diversity can be more complex than the number of*
*stems and we thus used this approach carefully. To be comparable across taxa,*
*we present the unmodified metrics per plot as the main result and used the*
*sample-coverage corrected metrics in the Appendix to test the robustness of our*
*results only.*

6.3 It seems that the authors found many issues with the data and had to adjust using
alternative procedures to deal with these cases. How do you think these artifacts affect
the analyses?

*There is a small number of cases in which the recovery time was surprisingly large*

*and a manual assessment revealed that this finding was implausible based on the*
*data (see Extended Data Fig. 7). We decided, for transparency, to report and justify*
*our choice of setting the recovery times, that were calculated to be very large by*
*our method, manually to 0 in these few cases. We believe such a manual curation*
*inside a single framework that aims at obtaining comparable results across such*
*a wide range of taxa is a reasonable way to deal with these cases. We do not think*
*that this choice impairs any other calculated values or analysis or conclusions*
*drawn from the results.*

*Additionally, for some datasets (seedlings, r-strategist frogs and leaf-litter*
*arthropods) calculation of standardized values for beta-diversity of some plot*
*combinations was not possible due to the low number of species. This is a known*
*problem for standardization techniques (see Chao et al. (2023)) and results for*
*these taxa may be less robust for the standardized values. This is one reason why*
*we opted for presenting these results only in the supplementary material, rather*
*than the main text. The results presented in the main text are not impaired by this*
*because there we present the unstandardized metrics. None of the general*
*conclusions in the main text will be influenced by this. We now added the*
*following paragraph to the methods to make readers aware of potential difficulties*
*when working with standardization of beta-diversity (p.22, l.638-641):*

**Because the beta standardization method requires a sufficient amount of**
**data, the standardized beta value may be less than 1 (or the standardized**
**dissimilarity may be negative) due to data sparsity and/or sampling variation.**

6.4 What is the intrinsic recovery rate (line 591)? How is this different from the recovery
1552 times? If this metric is so correlated with resilience, this makes confusing to understand
the way you deal with the different terms and concepts.

*The intrinsic recovery rate is the fitting parameter lambda in Eq. 2, which we now*
*simply term “return rate” for clarity and simplicity in the revised manuscript. We*
*previously calculated a “linearized” return rate based on resistance and recovery*
*time, because we thought the units of that measure in % recovery to reference*
*conditions per year would provide a more intuitive measure of the return rate.*
*However, we agree to major comment 7 of reviewer #2 that the non-linear fashion*

*of recovery should be acknowledged and now simply report the parameter*
*lambda as the return rate.*

6.5 The section describing the impurity analyses needs clarification (lines 614-626).

*Thanks for pointing this out. We now write (p.26, l.767-770):*

**The split points are chosen such that the impurity (the variance of values in**
**each node) is minimal. The impurity itself can then be used as a measure for**
**the importance of the predictor variables in determining the response**
**variable.**

7. Literature analysis

The literature analysis comes unexpectedly and is not integrated in the introduction. This
makes the motivation of this section unclear to the reader. Also, since studies evaluating
recovery from a multi-taxa perspective are so scarce, I wonder why the timeframe of the
analysis was so short? For instance, I can think of an important paper of Daisy Dent and
1575 J.. Wright (Biolo. Conservation 2009) that looks at recovery in different groups, but that is
1576 not in TableS5. By looking at this table I was confused by the fact that the search was for
the years 2016-2023, yet there are many papers going back to 1990.

*We thank the reviewer for pointing out this point of confusion. We used 3 key*
*reviews that were done until 2016 and that already contain a large database of*
*studies. The mentioned study (Dent & Wright (2009)) is among the reviews that*
*was already incorporated in two reviews that we screened (Curran et al., 2014,*
*Crouzeilles et al., 2016) which is why we did not screen it ourselves anymore.*

*The conditions we had for studies to be included in our analysis were more*
*restrictive than the conditions the authors used in their reviews, which is why we*
*decided to use their databases and use a subset of studies reported there that met*
*our conditions. We then performed a literature search on our own only for the time*
*of 2016-2023 to also include more recent works in our analysis. We clarified this*
*in the methods section and now write (p.27-28, l.821-823):*

**We performed a literature search including key reviews^{20,63,64} that were**
**already done in the field of forest restoration until the year 2016 and screened**
**literature reported therein for the conditions stated before.**

*We also introduce the literature search better in the introduction by writing (p.6,*
*l.182-184):*

**We assess the robustness and generality of our results and compare our**
**findings to other tropical forest regions by calculating recovery times,**
**resistance and return rates for data from literature.**

8. Restoration

8.1 Restoration is mentioned a lot in the Introduction and I think it's used as a hook to
provide some relevance to the study, but without support. How exactly do the results
found help to restoration strategies? Honestly, this is not clear to me. I recommend the
reading of Brancalion & Hall (see Brancalion & Hall, J Applied Ecological, 2020 and Hall
& Brancalion Science 2020) as an overview of the major challenges of restoration from a
practical perspective.

*We thank the reviewer for pointing out these key references in the field of assisted*
*forest restoration. Many other studies explicitly compared assisted versus natural*
*regeneration as two strategies of restoration, e.g. Crouzeilles et al., (2016) and*
*Chazdon et al., (2025). The relevance of our study lies in providing evidence that*
*natural regeneration can be effective in achieving the restoration of complex*
*biodiversity measures – particularly species composition for multiple organism*
*groups – and also to provide a quantitative time frame. In the past, the uncertainty*
*of ecological outcomes and time-scales required for natural regeneration*
*approaches to yield results have hampered their application (see e.g. Chazdon et*
*al. 2016, Chazdon et al. 2025, Williams et al. 2024). We hope the results of our*
*paper may advance the usage of this cost-effective approach to achieve goals for*
*forest restoration and ecosystem conservation. The conclusions were now*
*significantly rephrased to better include this applied aspect by writing (p.15, l.387-*
*389):*

**Overall, our results underline that cost-effective natural regeneration via**
**abandonment of agricultural land is a powerful restoration strategy for large**
**areas of tropical landscapes with small-holder agriculture to meet the UN**
**Decade on Ecosystem Restoration goals.**

8.2 Also, there are some statements that are made without sufficient consideration. In
particular, I disagree with the sentence “only large-scale restoration of secondary forests
can help to achieve global biodiversity conservation and climate change mitigation
goals” (l115). As a matter of fact, secondary forests should not need restoration, unless
they are in a arrested state. What about abandoned lands? Moreover, and maybe more
importantly, this sentence can give a misleading message: if restoration is the only way
to achieve global conservation goals, that means that conserving old-growth forests is
worthless.

Please be careful with these kind of statements in the future.

*We of course regard conservation of old-growth forests as important. We*
*rephrased the sentence to (p.3, l.109-112):*

**Old-growth forests are irreplaceable¹¹ and need to be conserved. Yet, more**
**than half of the world’s tropical forests have already been lost or degraded².**
**Because 70% of tropical forests are secondary¹², their conservation can**
**contribute substantially to achieve global biodiversity conservation and**
**climate change mitigation goals^{13,14,15}.**

*We hope this avoids any misunderstanding regarding the high value we ascribe to*
*pristine old-growth forests.*

----

I hope these comments will be helpful during the revision process. I believe that the
datasets used and the analyses performed can be the basis for a complete and thorough
study evaluating recovery after land use from a multi-taxa perspective. However, I do not
think the scope of this study and the limited generalizability of its results are novel
enough for a high profile journal like Nature.

*We thank the reviewer for very constructive comments and believe that the*
*revision of our manuscript in line with the comments of the reviewer substantially*
*improved the manuscript.*

**Additional changes:**

**The track-changes manuscript highlights all changes to the text. As there were**
**remarks of the reviewers to the introduction, results and conclusion, we improved**
**the wording in the text to incorporate all requested changes but keep the text flow.**
**We also changed the figure captions to include all requests by the reviewers. We**
**also updated affiliations.**

**We removed Extended Data Tables 2 and 3 to make room for Figures requested by**
**the reviewers. The information is now in Supplementary Tables 7 and 8. We also**
**removed Extended Data Fig. 2 and 3, because the information in those figures is now**
**more concisely included as data in Supplementary Tables 3 and the newly created**
**Supplementary Table 9. We now include, motivated by the reviewers, Extended Data**
**Figs. 1-5.**

**The datasets for bats and trees were updated, as new information on species**
**identities became available in some instances in the meantime.**

**All data and code has been updated and is provided as a figshare database as part**
**of the submission.**

**References:**

1. Poorter, L. *et al.* Multidimensional tropical forest recovery. *Science* 374, 1370–
1376 (2021).

2. Rozendaal, D. M. A. *et al.* Biodiversity recovery of Neotropical secondary forests.
*Sci. Adv.* 5, eaau3114 (2019).

3. Hoenle, P. O. *et al.* Rapid ant community reassembly in a Neotropical forest:
Recovery dynamics and land-use legacy. *Ecological Applications*, **32**, e2559.
(2022).

4. Guariguata, M. R., & Ostertag, R. Neotropical secondary forest succession:
changes in structural and functional characteristics. *Forest ecology and*

- *management*, 148, 185-206. (2001).
- 5. Jakovac, C. C. *et al.* The role of land-use history in driving successional pathways
and its implications for the restoration of tropical forests. *Biological Reviews*, 96,
1114-1134. (2021).
- 6. Escobar, S. *et al.* Reassembly of a tropical rainforest: A new chronosequence in
the Chocó tested with the recovery of tree attributes. *Ecosphere*, **16**, e70157.
(2025).
- 7. Pimm, S. L., Donohue, I., Montoya, J. M., & Loreau, M. Measuring resilience is
essential to understand it. *Nature Sustainability*, **2**, 895-897. (2019).
- 8. Van Nes, E. H., & Scheffer, M. Slow recovery from perturbations as a generic
indicator of a nearby catastrophic shift. *The American Naturalist*, **169**, 738-747.
(2007).
- 9. Van Meerbeek, K., Jucker, T. & Svenning, J.-C. Unifying the concepts of stability
and resilience in ecology. *J. Ecol.* 109, 3114–3132 (2021).
- 10. Holling, C. S. *Engineering Within Ecological Constraints*. (National Academies
Press, 1996).
- 11. Pimm, S. L. The complexity and stability of ecosystems. *Nature*, **307**, 321-326.
(1984).
- 12. Justus, J. (2008). Complexity, diversity, and stability.
- 13. Hillebrand, H., Langenheder, S., Lebet, K., Lindström, E., Östman, Ö., & Striebel,
1707 M. Decomposing multiple dimensions of stability in global change experiments.
*Ecology letters*, **21**, 21-30. (2018).
- 14. White, L., O'Connor, N. E., Yang, Q., Emmerson, M. C., & Donohue, I.. Individual
species provide multifaceted contributions to the stability of ecosystems. *Nature*
*Ecology & Evolution*, **4**, 1594-1601. (2020)
- 15. Dixon, P. M. The bootstrap and the jackknife: describing the precision of
ecological indices. In *Design and analysis of ecological experiments* (pp. 290-
318). Chapman and Hall/CRC. (2020).
- 16. Tibshirani, R. J., & Efron, B. An introduction to the bootstrap. *Monographs on*
*statistics and applied probability*, **57**, 1-436. (1993).
- 17. Manly, B. F. A method for the estimation of parameters for natural stage-
structured populations. *Population Ecology*, **39**, 101-111. (1997).

- 18. Meyer, J. S., Ingersoll, C. G., McDonald, L. L., & Boyce, M. S. Estimating
uncertainty in population growth rates: jackknife vs. bootstrap techniques.
*Ecology*, **67**, 1156-1166. (1986).
- 19. Donoso, D. A., Johnston, M. K., & Kaspari, M. Trees as templates for tropical litter
arthropod diversity. *Oecologia*, **164**, 201-211. (2010).
- 20. Chao, A. *et al.* Rarefaction and extrapolation with Hill numbers: a framework for
sampling and estimation in species diversity studies. *Ecol. Monogr.* **84**, 45–67
(2014).
- 21. Chao, A. *et al.* Rarefaction and extrapolation with beta diversity under a
framework of Hill numbers: The iNEXT.beta3D standardization. *Ecol. Monogr.* **93**,
e1588 (2023).
- 22. Whittaker, R. H. Evolution and measurement of species diversity. *Taxon*, **21**, 213-
251. (1972).
- 23. Jost, L. Partitioning diversity into independent alpha and beta components.
*Ecology*, **88**, 2427-2439. (2007).
- 24. Jost, L. Entropy and diversity. *Oikos*, **113**, 363-375. (2006).
- 25. Ingrisch, J., & Bahn, M. Towards a comparable quantification of resilience. *Trends*
*in Ecology & Evolution*, **33**, 251-259. (2018).
- 26. Rüger, N. *et al.* Demographic trade-offs predict tropical forest dynamics. *Science*,
**368**, 165-168. (2020).
- 27. Capdevila, P., *et al.* (2022). Life history mediates the trade-offs among different
components of demographic resilience. *Ecology Letters*, **25**, 1566-1579.
- 28. Li, Z., Zhang, H., Xu, Y., & Wang, S. (2021). Composition of ‘fast–slow’ traits drives
avian community stability over North America. *Functional Ecology*, **35**, 2831-
2840.
- 29. Dent, D. H., & Wright, S. J. The future of tropical species in secondary forests: a
quantitative review. *Biological conservation*, **142**, 2833-2843. (2009).
- 30. Curran, M., Hellweg, S. & Beck, J. Is there any empirical support for biodiversity
offset policy? *Ecol. Appl.* **24**, 617–632 (2014).
- 31. Crouzeilles, R., Ferreira, M. S. & Curran, M. Forest restoration: A global dataset for
biodiversity and vegetation structure. *Ecology*, **97**, 2167 (2016).
- 32. Crouzeilles, R., Curran, M., Ferreira, M. S., Lindenmayer, D. B., Grelle, C. E., & Rey

- Benayas, J. M. (2016). A global meta-analysis on the ecological drivers of forest
restoration success. *Nature communications*, **7**, 11666.
- 33. Chazdon, R. L., Blüthgen, N., Brancalion, P. H., Heinrich, V., & Bongers, F. Drivers
and benefits of natural regeneration in tropical forests. *Nature Reviews*
*Biodiversity*, **1**, 1-17 (2025).
- 34. Chazdon, R. L., & Guariguata, M. R. Natural regeneration as a tool for large-scale
forest restoration in the tropics: prospects and challenges. *Biotropica*, **48**, 716-
730. (2016).
- 35. Williams, B. A. *et al.* Global potential for natural regeneration in deforested
tropical regions. *Nature*, **636**, 131–137 (2024).

Authors' response to review

Nature manuscript 2024-12-26946A

Resistance and resilience of biodiversity in a tropical rainforest
(new title: **Biodiversity resilience in a tropical rainforest**)

We thank all four reviewers for their helpful comments and suggestions. We respond to each reviewer point-by-point below (indented and *italicised*, **boldface** for text passages in the new manuscript). The comments helped to significantly improve the manuscript with respect to clarity and robustness of the findings.

The most important changes are:

1. An improved definition of the concept of resilience, including a change of the title
2. A refinement of the framework of reproductive strategies,
3. A refined description and justification of some methodology
4. An update of Fig. 2 and Extended Data Fig. 4

We want to point out that, to support the answer to major comment #3 of reviewer #4, we included a supplementary table (Table_Rebuttal_only_1.csv) containing all data that was created with the alternative model to this rebuttal. This table is, however, not intended to be included as a supplementary table in the manuscript, but only as part of this rebuttal for the reviewers' kind consideration.

Additional minor changes in addition to the reviewers' suggestions are given at the end. All references mentioned are summarized at the end. Please note that the number of references in the main manuscript now exceeds the limit of 50 references in order to accommodate the adding of references that were suggested by the reviewers.

On behalf of all authors,
Nico Blüthgen and Timo Metz

—

Referees' comments:

Referee #1 (Remarks to the Author):

This is a review of the manuscript revision. The revised manuscript is an improvement on the original version. I have found the revision thorough and addressing the concerns of all three reviewers well. I have no further questions/concerns with this manuscript version, after my already positive review of the original manuscript.

We thank the reviewer for this very positive evaluation of our revised manuscript.

Referee #2 (Remarks to the Author):

The manuscript by Metz et al evaluates for a landscape in the tropical Colombian Chocoo the resistance of 16 taxonomic group to forest conversion to agriculture land use, and their subsequent recovery ('resilience') towards old-growth forest values, and how much time this takes.

The strengths of the manuscript are 1) the comprehensive description of recovery across many taxonomic groups belonging to different trophic levels, using a standardized plot network and analytical framework with a massive dataset which allow for strong generalizations of the results, 2) the interesting analytical framework by decomposing recovery into its two underlying components, resistance and recovery rate which are associated with different ecological processes (tolerance versus colonization and regrowing capacity), 3) the surprising result that some short-lived taxonomic groups show arrested succession, and that simple ecological attributes can not explain resistance or recovery, 4) its important implications that natural regrowth can be used as a low-cost nature based solution, to scale up forest restoration, and provide tangible outcomes for the conservation of multiple taxa.

This is the second time I review the manuscript. I thank the authors for their openness and willingness to revise their manuscript based on the comments and their thorough revision of the manuscript and replies. I think the manuscript has substantially improved because 1) it now also present data on recovery after 30 years, which is more solid and more policy relevant, 2) the legacy effect of the two previous land uses are now explained, and included in all graphs, 3) the terminology on resistance, return rate and predicted recovery time has been clearly

explained and linked to how it is used in the literature, 4) the methods are better explained, justified, the manuscript is now more hypothesis driven, and the main patterns are well explained.

I agree with all replies to my comments and the changes made, and have only two minor remarks. I also have read the comments and replies to reviewer 3, and I think the authors satisfactory addressed the reviewers comments. I disagree with reviewer 3 that the manuscript is not novel, for the four reasons I mentioned above. It is true that the study is carried out at a single site, but the authors made convincingly the point that it is a representative site for large parts in the tropics, and I think the manuscript significantly advances the field because of its broadness and depth by studying succession in a thorough way for so many taxa.

Overall, the manuscript was fluently written, interesting, and a pleasure to read, and it provides in my opinion a novel and important contribution to science and, potentially, to the application of restoration. It is the kind of manuscript I hope and expect to see published in a high impact, high quality journal like Nature.

I congratulate the authors with the result. Please find my last very minor comments below.

We thank the reviewer for these positive comments on our manuscript and we are happy that our revision was satisfactory. We reply to the remaining minor comments point-by-point below.

MINOR COMMENTS

1. RECOVERY TIME. I am fine with your reply to my comments, but please mention explicitly in the methods or the results the caveat that for 21 out of the 56 comparisons the predicted recovery time is longer than the oldest secondary forest plot in the dataset (38 years). This is mitigated to some extent by including a large number of old-growth forest plots as an asymptotic reference, but this leads to a larger uncertainty in the estimates of predicted recovery time larger than 40 years.

We agree with the reviewer that this important information needs to be included. We now write (p.25-26, 1.756-772) in the methods section (including some related improvements to the text in response to major comment 4 of reviewer #4):

Several cases (21/24 for cacao/pasture respectively) out of the 98 calculated recovery times presented in the main text for cacao and pasture had a recovery time longer than the 38 years covered by the chronosequence. The recovery times of taxa not recovering within 38 years may be considered less reliable than those of the taxa that recover within the timespan covered by the chronosequence. However, even such taxa are important to include in the overall comparison despite their low return rate, high uncertainty of predictions, and low R^2 (see Supplementary Table 1) for our negative exponential model. This represents an unbiased characterization of different attributes of forest ecosystems and avoids a bias towards quickly recovering taxa. Despite possible alternative models to describe recovery, we focused on a single model to facilitate a direct comparison of recovery time and return rate. To obtain a comparable measure of recovery that is similarly robust for all taxa, we calculated the percentage of recovery after 30 years. The percentage of recovery after 30 years was calculated using Eq. 4 but instead of ψ_0 we inserted the modelled value ψ_{30} of the respective index at 30 years after beginning of recovery, which was obtained by evaluating Eq. 5 after fitting it to the data at time $t=30$ years. This time was chosen because it represents a time scale that is relevant for many restoration projects but was still within a timespan well covered by our chronosequence to provide robust values.

2. PREVIOUS LAND USE. The description of the two land use systems (pasture, cocoa plantation) in the extended methods is now better. Add to the main methods two lines how these land use systems look like (only cocoa, 5 m tall, open pastures, with occasional palms or remnant trees), as it makes it clear to the reader what kind of system it is and how intensive it is. Add to the extended methods for pasture what kind of grass it is? Native/Exotic? What species. And it would be great if you could add 2-3 lines about the size of these patches (Maybe you did it already) and for how long it was ca. used.

This is indeed a relevant detail to be included in the Methods. We now added the following paragraph to the Methods (p.21, 1.601-614):
Pastures were grazed extensively by low densities of cattle and occasionally by horses and were

dominated by aggressive pasture grasses such as *Brachiaria* or *Axonopus scoparius*. Cacao plantations were monocultures of sun-exposed *Theobroma cacao* trees which were spaced 2–4 m apart and grew to heights of 5–10 m. These plantations were regularly treated with herbicides. Whereas cacao plantations generally lacked shade trees, pastures had some remnant trees or palms, or shrubs along creeks. These characteristics were also typical of the pastures and cacao plantations that were used several decades ago and which now represent regenerating forests in our study area. Old-growth forests contained large, slow-growing trees with potential for timber use and showed no signs of harvesting. Before they regenerated as secondary forests, pastures and cacao plantations had a similar duration of land use (mean 11.4 years, range 1 – 30 years). Regenerating pastures were larger (11.3 ha, range 1.2 – 46.7 ha) than cacao plantations (2.0 ha, 0.3 – 5.7 ha). All plots were located in a relatively intact landscape with a mean forest cover of $74\% \pm 2.8\%$ (24 – 100%) within a 1-km radius. For further details on the study site, see ref. 47.

3. Main text L198. You now emphasize that when you know recovery of one component (slow recovery in spp composition) does not inform of recovery of another (fast recovery in abundance and richness). This is fine, but I think it is more informative that some attributes recover in short time, but if you want recovery of typical old-growth forest species, that it takes a long time

We agree with the reviewer and now write (p.6, l.199-202):

The fast recovery of abundance and species diversity are important to obtain high levels of productivity and other ecosystem functions. Focusing on them, however, may underestimate the much longer time necessary for a complete community composition and thus the typical old-growth forest species to return.

4. Your rebuttal, line 947-965. Include a bit more of this in the extended methods, as with what you now say in the methods (quoted in line 966-970) I could not fully understand and repeat what you have done.

We agree. We now write (p.21-22, l.626-638):

Beta-diversity Hill-numbers were calculated as the fraction of gamma- and alpha-diversity Hill-numbers of orders $q=0, 1$ and 2 according to ref. 54:

$${}^qD_{\beta} = {}^qD_{\alpha}/{}^qD_{\gamma} \quad (1)$$

where gamma-diversity was defined for each pair of plots by pooling data from both communities. The beta-diversity Hill-numbers were subsequently transformed to measures of species composition similarity according to ref. 55. The community overlap of order q of 2 communities is there given by:

$$\text{overlap of order } q = \frac{(1/{}^qD_{\beta})^{(q-1)} - (\frac{1}{2})^{q-1}}{1 - (\frac{1}{2})^{q-1}} \quad (2)$$

This formula does not converge for $q=1$. However, as q approaches 1 the overlap is given by:

$$\text{overlap of order } 1 = (\ln 2 - H_{\beta,Shan})/\ln 2 \quad (3)$$

With $H_{\beta,Shan}$ the Shannon-index. As derived in ref. 56 the above formulas yield for $q=0$ the Sorensen-index, for $q=1$ the Horn-index and for $q=2$ the Morisita-Horn index.

We furthermore added (p.22, l.658-660):

We then extrapolated the beta-diversity Hill-numbers of each pooled pair of plots to twice the observed number of individuals by extrapolating the alpha- and gamma-diversity component separately.

5. Main text L97. In your abstract, it is unclear if the 100 years of trees and seedlings refer to all 3 components, or just to species composition

We agree. We now omitted the mentioning of exact numbers, as they differ between cacao and pasture and also across attributes. We thus now only mention the fact that mobile animal communities recover faster than trees and seedlings, as this is a finding that is true for cacao and pasture and also across attributes. We now write (p.3, l.96-97):

Mobile animal communities acting as seed dispersers or pollinators had high resistance levels and recovered faster than trees or tree seedlings.

6. Main text L125. This is still not totally clear. Is it decreased uncertainty about time scales that fasters adoption of natural regeneration, or a better knowledge how quickly different taxonomic groups return that fosters the adoption?

Yes, the knowledge, not the uncertainty is crucial. This sentence has now been clarified as follows (p.4, l.125-127):

Improving our understanding of how quickly different taxonomic groups recover could enable more informed decisions about when to use natural regeneration as a cost-effective restoration tool, or where assisted restoration measures may be required^{8,15,19,24,25}.

7. L383. Replace 'Timber management plans' by 'forest management plans', or something along those lines

Thanks, corrected.

Referee #4 (Remarks to the Author):

The study reported in this manuscript makes a relevant contribution to tropical forest taxonomic diversity conservation in an era of accelerated transformation of old-growth forests into different agroecosystems. The study is particularly valuable due to its comprehensiveness, as it provides estimations of recovery times upon disturbance cessation for organisms representing multiple taxa across three kingdoms. Similar holistic efforts to assess the recovery of biological diversity are scanty and urgently needed if we are to soundly assess the possibility of recovering fully functional ecosystems in the future. The study shows that the recovery of three diversity metrics, namely, abundance, diversity (mostly species richness), and composition, exhibits enormous variation among taxa and diversity metrics. Also, it shows the differential effects of different land uses on diversity resilience, although it falls short of finding a relationship between this variation and the broad range of life-histories and ecological strategies represented among the groups examined.

Despite the originality and significance of this study, there are a few issues that still require attention from the authors, especially considering that this paper is meant for such a high-profile journal as Nature, with the expected consequences that this potential publication has for advancing the theoretical and conceptual framework of Biological Conservation Science and Ecology in general. These issues are mainly of a conceptual or theoretical nature, but there is also one comment related to the modeling of diversity return rates, which is an important component of the analysis.

We thank the reviewer for this positive evaluation of our manuscript and respond to each of the comments point-by-point below.

1. Ambiguous standing about the concept of resilience.

This issue was raised by Reviewer 3, but in my view, it was not satisfactorily addressed. The first indication of a potential drawback with this issue appears right in the title of the manuscript. When I first read it, I was intrigued by the two main terms it contains (resistance and resilience), presented in a way that implies that these are different but equally ranking properties of a system. Undoubtedly, the importance of resilience as a key ecological concept has increased tremendously in the face of ecosystem degradation and biodiversity losses occurring on the planet. However, it is important to understand that this conceptual framework has undergone a rapid and interesting development since Holling's (1973) seminal paper, particularly in a direction that makes this property easier to measure, rather than an abstract characteristic based on unquantifiable properties like the "potential to change". We must recall that in his 1973 paper, Holling discussed ecosystem resilience as a "measure of its capacity to absorb changes and continue existing". Along with this concept, he defined stability as the ability of a system to return to the equilibrium state after being temporarily perturbed. Formally, he defined resilience as the size of the

stability domain (i.e., the stability basin), or the 'amount' of disturbance a system can tolerate before shifting into an alternative configuration. More than two decades later, Holling (1986) attempted to rid these concepts from their evident subjectivity by conceptually distinguishing between what he called "engineering resilience" (a system's capacity -not necessarily its speed- to return to the stationary state after a disturbance) and "ecological resilience" (the magnitude of the disturbance a system can absorb before shifting from one state to another). The integrated understanding of these two properties led Capdevilla et al. (2021) to think of resilience as a system's ability to face change and cope with it.

Despite these efforts, the measurement of resilience remained loaded with subjectivity for almost two decades, as it required assessing properties as difficult to grasp (and measure) as the system's "potential energy". In the opinion of many scholars working on this topic and interested in the proper assessment of resilience, a major step forward was the conceptualization of resilience as a complex property that thoroughly describes the entire trajectory of one or more state variables of a system since the moment it is affected by a disturbance until its partial or full recovery after the disturbance ceases.

Along this line of thought, Hogdson and colleagues (2015) proposed the measurement of rates of change in ecosystem properties as a proxy of its "potential energy" and put forward a novel conceptualization of resilience as a bivariate property that can be decomposed into two components, namely 'resistance' and 'recovery'. This conceptualization was taken on by Ingrisch & Bahn (2018), who defined resilience as the capacity of a system to maintain its state and recover from disturbances and highlighted its definition as a complex concept encompassing two components (resistance and recovery). Therefore, I am intrigued by the fact that Fig. 1a in this manuscript virtually reproduces the illustration of Ingrisch and Bahn's idea and, at the same time, it seems to disregard it (the figure legend reads "the return rate towards the pre-disturbance reference state is used to quantify resilience". Ingrisch and Bahn underscore the importance of normalizing both resistance and recovery time to achieve more objective and sounder comparisons among ecosystems. In addition to Muñoz et al.'s (2021) paper mentioned by Reviewer 3, I would like to suggest van der Sande et al.'s (2023) paper on soil resistance and recovery in tropical ecosystems as an example of this currently mainstream approach.

Given the current development of the resilience concept, I would like to invite the authors to adopt this mainstream framework, which is virtually the same one they used; the biggest change required in this regard would be in the title of the paper, for which the following phrase would be appropriate: "Diversity resilience in a tropical rain forest".

References:

- Capdevila, P et al. (2021). Reconciling resilience across ecological systems, species and subdisciplines. *J Ecol.*, 109(9), 3102-3113.
- Holling CS (1973). Resilience and stability of ecological systems. *Annu. Rev. Ecol. Syst.*, 4, pp. 1-23.
- Holling CS (1996). Engineering resilience versus ecological resilience. In: National Academy of Engineering. *Engineering Within Ecological Constraints*. Washington, DC: The National Academies Press.
- Hogdson D et al. (2015). What do you mean, 'resilient'? *Trends Ecol. Evol.* 30: 503-506.
- Ingrisch, J & Bahn, M (2018). Towards a comparable quantification of resilience. *Trends Ecol. Evol.*, 33(4), 251-259.
- Muñoz, R et al. (2021). Autogenic regulation and resilience in tropical dry forest. *J. Ecol.*, 109(9), 3295-3307.
- van der Sande, MT et al. (2023). Soil resistance and recovery during neotropical forest succession. *Philosophical Transactions of the Royal Society B, Biological Sciences*, 378(1867): 20210074.

We thank the reviewer for this very detailed, constructive and helpful elaboration on the concept of resilience. In our previous version of the manuscript and in other publications from our research unit, we indeed opted for disentangling resistance and resilience. Thereby, resilience simply describes the recovery component i.e. the process of "rebounding" after the disturbance, whereas resistance describes the impact of disturbance. Resilience then may be measured by the return rate as provided in our initial manuscript. This idea follows the classical definition of resilience as "speed of return" normalized with the strength of impact sensu Pimm, 1984, which has been used quite frequently in the past (e.g. White et al., 2020, Pimm et al., 2019, Justus 2007, Hillebrand et al., 2018) and was taken up also in a recent work by Van Meerbeek et al. 2021 that tried to unify concepts of equilibrium and non-equilibrium stability theory and disentangles resistance and resilience as two separately measurable components of stability and suggested the resilience to be simply measured as the rate of return.

However, we thank the reviewer for pointing out that there has been a strong development in creating unified frameworks for measurable resilience concepts (as nicely summarized in Capdevila et al., 2021), one very common and mainstream one being the one of Ingrisich & Bahn, 2018, which follows the resilience concept of Hodgson et al., 2015, and essentially suggests to view resilience as being composed of resistance and recovery, where the recovery component may be measured through the return rate. As the reviewer points out, we do in fact report all the required components to quantify resilience as defined by Ingrisich & Bahn, 2018.

Therefore, instead of focusing on our previously chosen definition, we do agree that the definition of resilience as a more complex concept encompassing two different dimensions, resistance and recovery (measured via the return rate), has its merit as multiple recent works follow this framework. To avoid misunderstandings we further clearly state in the methods that all presented variables in the main text (resistance, return rate, recovery time and relative recovery after 30 years) are clearly defined and unambiguous and therefore there is the possibility of robust comparisons of these variables across datasets. We therefore modified the manuscript in four ways:

(1) We simplified the title in accordance with the reviewer's suggestion to: **"Biodiversity resilience in a tropical rainforest"**.

(2) We also changed the main text (also in accordance with the reviewer's minor comment 1) and now write (p.4, l.129-134):

The recovery trajectories and recovery times of ecosystems following a perturbation depend on two components: resistance, defined as the ability to withstand disturbance and recovery, which is the process of returning to the reference state as measured by the return rate (Fig. 1)^{26,27}. A common definition conceptualizes the combination of resistance and recovery as the system's resilience^{4,27-29} while other works define resilience more narrowly as the speed of return alone³⁰⁻³⁴.

(3) We additionally changed the text in the Methods in which we now include a short elaboration of different concepts of resilience, consistent with the definitions found in Capdevila et al., 2021 and elsewhere (p.25, l.728-738):

Resilience is often either defined as the ability to maintain a qualitatively similar state in the face of disturbance and not switch to an alternative state (ecological resilience^{27,28,62}) or the process of recovery of the system to reference conditions after a disturbance (engineering resilience^{26,27,28}). The concept of engineering resilience is related to a single stability regime (which is the view we follow in this work), whereas ecological resilience assumes multiple alternative states. Recently it was proposed to consider resistance and recovery (measurable as the rate of return to reference conditions) as two components of resilience^{27,28,29}. To avoid confusion with various definitions of resilience in the literature²⁶⁻³⁴, throughout the manuscript we use the term "return rate" for the specific measure represented by λ , and "recovery time" for describing the time until 90% of the reference value is reached.

(4) We now write in the legend to Fig. 1 (p.5, l. 147-150):
For any recovering system attribute (e.g. the diversity of a certain animal group), the *resistance*, defined as the amount remaining during perturbation (clear cut and agriculture), and the *return rate* towards the pre-disturbance reference state (old-growth forests) determine the *recovery time*.

2. Confusion of the concepts of density-dependent r- and K-selection vs. r- and K-strategies.

In searching for evidence of the effect of life history strategies and trophic levels on the observed recovery times for the different groups examined, the authors classified their study organisms as "r- vs. K-strategists" and cite Mac Arthur and Wilson's (1967) seminal book 'The Theory of Island Biogeography' as the source of this dichotomy, but this is not entirely correct. Mac Arthur and Wilson developed the notion of density-dependent natural selection with two extremes occurring under highly contrasting demographic conditions (very low or very high population densities). Within this framework, frogs are not necessarily "r-strategists"; for any population of any species, r-selection takes place when population densities are low, so that highly reproductive genotypes have higher fitness than genotypes investing more in parental care but reproducing less. The opposite is true when the same population attains a high density, under which circumstance those genotypes that invest more in parental care

have higher fitness than those investing in larger offspring, many of which will not achieve adulthood. If the authors prefer to classify species by strategy not by selection type, they should cite Eric Pianka's (1970) work, as he is responsible for the unfortunate but widely used misinterpretation of MacArthur and Wilson's density-dependent selection model as ecological strategies. In this regard, I invite the authors to review these key references, particularly the first one (Boyce, 1984):

Boyce, MS (1984). Restitution of r-and K-selection as a model of density-dependent natural selection. *Annu. Rev Ecol Syst*, 15, 427-447.

Reznick, D et al. (2002). r-and K-selection revisited: the role of population regulation in life-history evolution. *Ecology*, 83(6), 1509-1520.

Engen, S & Sæther, BE (2017). r-and K-selection in fluctuating populations is determined by the evolutionary trade-off between two fitness measures: Growth rate and lifetime reproductive success. *Evolution*, 71(1), 167-173.

MacArthur, RH & Wilson EO (2001). *The Theory of Island Biogeography* (Vol. 1). Princeton University Press, Princeton.

Pianka, E. R. (1970). On r-and K-selection. *Am. Nat.*, 104(940), 592-597.

This is a very thorough, convincing and helpful comment about a controversial conceptual issue that has been overlooked by us in the previous versions; thanks for also pointing out these very valuable references on r- and K-selection which we were not aware about. Indeed, like several other studies, e.g. on the 'economic spectrum' across taxa, we have misinterpreted the original concept of "density-dependent natural selection" by MacArthur and Wilson (1967) and falsely used it to define a (fixed and relatively coarse) 'strategy' for entire taxa that clearly doesn't represent a clearly defined density-dependent response at a taxon level. We have now completely abandoned the unsuitable r/K terminology, and rephrased the paragraphs concerned with the life-history strategies as follows (reducing the references to those that used such fast/slow 'strategies' at species level):

(1) Main text (p.14, l.352-361):

Differences in environmental responses and recovery dynamics across taxa may correspond to a fundamental divergence in their complex life-history strategies and underlying trade-offs^{41,42}. For example, species with longer generation time often show weaker responses to land-use intensity in grasslands⁴³, as well as a slower demographic post-disturbance recovery⁴⁴. We thus tested whether age at first reproduction explained the order of recovery across taxa (for classification of taxa see Supplementary Table 4). Although taxa differed fundamentally in generation times, this variation could not explain their variation in recovery time, resistance and return rate of community composition (Supplementary Table 2). Even within five taxonomic groups (frogs, trees, birds, bees, saproxylic beetles) these life-history strategies had no consistent effect across species (Supplementary Table 2).

(2) Methods (p.27, l.815-820):

(a) Life-history strategies: Corresponding to a 'slow-fast' continuum according to the life-history of these taxa^{43,44}, taxa were ordered by an increasing age at first reproduction into four levels: (1) bacteria, (2) invertebrates, (3) vertebrates and (4) trees. This coarse rank order is also obviously consistent with their life span and body mass. A more fine-grained continuum⁴¹⁻⁴⁴ is prevented by large variability within each taxon and thus would require data at species level for all taxa.

(3) Methods (p.28, l.837-843):

Five of the taxa (Frogs, trees, birds, bees and saproxylic beetles) have been split into two groups of species each, representing fast versus slow life-history strategies within each taxon. Frogs were split according to fast versus slow reproduction strategies, trees into fast and slow growing ones, bees and saproxylic beetles were split into social/non-social groups. This categorization was based on expert knowledge (frogs: M.O.R., trees: J.E.G.A., bees: U.M.D and S.D.L., saproxylic beetles: J.M.). Birds were split into short and long-lived species according to their generation time.

(4) Methods (p.23, l.667-668):
The same was true for frogs ('fast' strategists) for standardized beta-diversity (q=0 and q=1).

3. Statistical analyses and interpretation

The statistical tests used in this manuscript to analyze the huge data set are appropriate. 95% confidence intervals of recovery times were correctly calculated through bootstrapping.

Recovery time and return rate are two metrics to measure recovery estimated by the authors by fitting a negative exponential function (eq. 2) to empirical data. Figure 1a shows this function fitted to a particular set of data (species composition of bees). In this figure, the model presents a good fit to the data; however, the results show that this is not a general trend. The R^2 column in Supplementary_Table_1.csv and Supplementary_Table_3.csv show a huge variation in this goodness-of-fit metric, ranging from -Infinite (i.e., nearly 0 or no variance explained at all) to 0.826, with a mean of 0.105 (excluding the -Infinities) and an extremely low median of 0.066. Among all R^2 values, only 10% are > 0.5 , while 25% of them are > 0.25 . These results imply that, for most cases, a negative exponential function was not a good model to describe the empirical data. Since recovery time and return rate depend on this fit, two possibilities should be considered: either (1) the results on the topic of recovery should be limited to those cases where the negative exponential function was a good representation of the pattern followed by data (i.e., those cases where R^2 was relatively high, preferably > 0.5), or (2) the authors could explore other models describing recovery to consider alternative recovery patterns and then perform model selection, this hopefully leading to higher R^2 values.

We thank the reviewer for the positive evaluation of our statistical methodology. We also thank the reviewer for suggesting alternatives to our approach of fitting the negative exponential model given by Eq. 2 (now Eq. 5 in the revised version) to obtain higher R^2 values.

One of the main messages of our paper is that species composition provides a more meaningful characterization of species community recovery than species richness, diversity or abundance. The diversity of some taxa remains unchanged with recovery time despite strong compositional turnover from agriculture to old-growth. We therefore want to point out that, in fact, the R^2 values for compositional recovery (which is the main focus of the results presented in the main text) for the 17 taxa shown in the main text are overall much higher than for diversity and abundance. The R^2 values across taxa have a median of 0.26 and 0.38 for cacao and pasture respectively in the 17 taxa shown in the main text. For abundance and diversity, R^2 values are often low (median cacao/pasture: 0.027/0.112 (abundance) and -0.017/0.116 (diversity) for the 17 taxa shown in the main text) due to the frequent absence of a trend leading to an overall impression of low R^2 values when calculating a median from the full dataset including all taxa and all indices.

However, we have also used an alternative model to our negative exponential model already in a previous paper (Hoenle et al. 2022), where we used a linear model with square-root transformed time-axis to predict return rate and recovery time for a single dataset. We had justified that model for three reasons: square-root transformation improves the model by a more uniform distribution of plot age, provides a realistic curve with a decreased recovery with time (dampening), and it has been used in the literature for other chronosequences. When applying this alternative model to our current data it occasionally fits better to certain datasets, as shown below, because it contains a freely chosen intercept and the curve does not necessarily end at the old-growth forest plots. However, this feature has also been often criticized as a general problem of linear (or linearized) functions, as it also allows the curve to 'overshoot' above the old-growth reference, rather than being constrained by a pre-defined start and end.

Compared to the alternative model (linearized function with flexible start and end), the negative exponential function given by Eq. 2 (now Eq. 5) has the advantage of being more motivated by theory of growth and recovery (start to end). It was used in a comparable previous study on trees (Poorter et al., 2021) and is particularly useful in a dataset as ours in which we have the start (agriculture plots) and end (old-growth forest reference) of the recovery process well constrained through the availability of multiple plots. Especially the old growth reference level has no age and thus cannot be included in the linear models, unless we would assign an arbitrary age value. The estimation of recovery times significantly benefits from the inclusion of the large amount of old-growth forest plots as a reference for the asymptote in the negative exponential model that we use.

Thanks to the reviewer's comment, we now consistently re-run the alternative, unconstrained model (square-root transformed linear model, with freely chosen intercept for secondary forests) across the

whole dataset, and compared the outcome of these datasets with the negative exponential function given by Eq. 2 (now Eq. 5) that has fixed start and end values. Overall, the results are highly consistent, and below we show a correlation of the recovery times between these two alternatives (Fig. R1). We show this correlation for all attributes (abundance, diversity and composition), and separately for composition alone which is the main focus of our paper. The results particularly confirm that the rank of taxa in predicted recovery times is highly consistent across methods (important for our main message, see e.g. Fig. 2).

Fig. R1: Recovery times predicted by the negative exponential model used in the paper versus the alternative unconstrained linearized model. The first (upper) figure describes all attributes, the second (lower) focuses on the most important community composition similarity to old-growth forests. Here a few cases of very long recovery time are not displayed for better readability (7 out of $N = 34$ communities for composition and 4 out of $N=64$ for abundance and diversity). We also removed implausible (negative) recovery times that were obtained with the linear model in cases where the slope was negative but the prediction for the intercept was below the value of old-growth forest plots (see Fig. R3, Bray-Curtis similarity of bacteria in cacao legacy plots, for an example).

Apart from a good correspondence of the order of taxa in model predictions for recovery, Fig. R2 shows that R^2 values of both fits are corresponding well, too, confirming that the ecologically more plausible negative exponential model does not create distorted impressions of recovery time or return rate on similarity of communities. When the square-root transformed linear model has higher R^2 (left side of the diagonal in Fig.R2), visual inspection of the cases showed that in most cases this is because the fitted regression is actually decreasing (see e.g. the trajectory for bacteria composition in Fig. R3) despite observed higher values in the old-growth forests. In these cases, recovery times cannot be reasonably estimated using the linear model, which is a disadvantage of that model. In other cases we find overshooting of old-growth forest values by the regression (see e.g. trajectory for the abundance of frugivorous birds in Fig. R3).

Fig. R2: R^2 values of the negative exponential model used in the paper versus the alternative unconstrained linearized model. The diagonal line describes where both R^2 values would be the same. The first (upper) figure describes all attributes, the second (lower) focuses on the most important community composition similarity to old-growth forests. Note that through the forcing of start and end in the negative exponential model, R^2 values can be technically negative (rather than constrained between 0 and 1) if the trend has a different direction, e.g. for bacteria (see Fig. R3).

Fig. R3: Recovery trajectories of species composition similarity to old-growth forest values (Bray-Curtis similarity) of bacteria in 10 cm depth (upper figure) and abundance of frugivorous birds (lower figure) using a linear model with square-root transformed time axis and freely chosen intercept.

This means, while our negative exponential model failed to capture specific dynamics in the secondary forests for some cases (as indicated by a lower R^2 than for the linear model), these dynamics are also not in any way related to the community composition of the old-growth forest corresponding to high or even extreme predictions of recovery times and slow or virtually zero return rates. The linear regression model does not provide a benefit here, despite a higher R^2 in some cases, as it does not describe the recovery trajectory when also taking the old-growth forest reference into account. Most likely, some communities (e.g. bacteria) never return to old-growth forest species composition and rather develop into an alternative state. If so, that lack of return is equally captured by the recovery time estimate of hundreds of years that

was given by our negative exponential model that includes the old-growth forest reference, which we also specifically discuss in the manuscript.

In Fig. R4, we show the relationship between R^2 and return rate for species composition for the main datasets shown in the main text. It is visible that low return rates often correspond to low R^2 , high return rates to higher R^2 . This comparison shows that low R^2 values for some datasets largely emerge due to the fact that the trajectories show no consistent or very shallow recovery trends (low return rates) over the chronosequence.

We believe, however, that it is crucial to retain all taxa in an (unbiased) overall comparison, despite low R^2 in some cases as it is then possible to see which taxa or ecosystem parameters recover much slower than others. Recovery times of slowly recovering taxa are based on extrapolation rather than being based on exact calculation, as the age of the oldest secondary forest in the chronosequence is 38 years. Slowly recovering taxa are nevertheless important to be included in this comparison across taxa (despite the fact that long recovery times are more uncertain to predict), to avoid an unrepresentative bias towards quickly recovering taxa.

Fig. R4: R^2 value of the negative exponential model in relation to the return rate for species composition recovery. Low R^2 values correspond to small return rates.

Moreover, the uncertainty of predictions for slow recovery (discussed in minor comment #1 of reviewer #2) motivated us to show calculations of the percentage of recovery after 30 years. These values are well within the time span covered by the chronosequence and thus represented by plot data rather than being extrapolated. Apart from higher accuracy, these 30 year values are important for communication about conservation goals and applications. Details for values at 30 years and confidence intervals are reported for all taxa.

We further think the calculation of confidence intervals based on a resampling-based technique (rather than based on the fit itself e.g. through usage of the covariance matrix) for recovery times and return rates makes the estimation of all recovery times robust within the certainty of the confidence intervals, despite low R^2 in some cases.

In summary, we decided to keep the focus on the negative exponential model in the results in the main text, even for taxa with low R^2 values, for three reasons: First, our model (now Eq. 5) is internally consistent, ecologically plausible, as well as comparable to another recent study on recovery and tropical

forest succession that was based on trees (Poorter et al., 2021). Second, a mix of models across datasets makes comparisons of return rates and recovery times challenging, because the fitted recovery trajectory is different. However, the comparison of return rates and recovery times across taxa is a major part of our manuscript and is therefore of high importance. Third, the analysis using a linear model with square-root-transformed plot age yields qualitatively the same result, indicating that the results of our non-linear model are highly robust.

To better justify why we include taxa with low R^2 values that often correspond to cases where there is no trend across the chronosequence, and why the calculation of the percentage of recovery after 30 years was added to enhance comparability across taxa with strongly varying recovery times, we have phrased the following sentence (p.25-26, l.756-772) in the Methods (see also response to minor comment 1 of reviewer #2):

Several cases (21/24 for cacao/pasture respectively) out of the 98 calculated recovery times presented in the main text for cacao and pasture had a recovery time longer than the 38 years covered by the chronosequence. The recovery times of taxa not recovering within 38 years may be considered less reliable than those of the taxa that recover within the timespan covered by the chronosequence. However, even such taxa are important to include in the overall comparison despite their low return rate, high uncertainty of predictions, and low R^2 (see Supplementary Table 1) for our negative exponential model. This represents an unbiased characterization of different attributes of forest ecosystems and avoids a bias towards quickly recovering taxa. Despite possible alternative models to describe recovery, we focused on a single model to facilitate a direct comparison of recovery time and return rate. To obtain a comparable measure of recovery that is similarly robust for all taxa, we calculated the percentage of recovery after 30 years. The percentage of recovery after 30 years was calculated using Eq. 4 but instead of ψ_0 we inserted the modelled value ψ_{30} of the respective index at 30 years after beginning of recovery, which was obtained by evaluating Eq. 5 after fitting it to the data at time $t=30$ years. This time was chosen because it represents a time scale that is relevant for many restoration projects but was still within a timespan well covered by our chronosequence to provide robust values.

4. Visualization

Figure 2 provides an attractive summary of this study's main results. It succeeds in visually conveying the complexity of the analyses, given the number of biological groups involved and the differences among the three diversity metrics examined. Although the horizontal axes of these graphs have logarithmic scales, it is unfortunate that the scale of grays depicting recovery times for the different groups has an arithmetic scale. This restricts the possibility of visually discriminating more finely among taxa with contrasting recovery times during the first three decades of recovery, which are critical given current patterns of land use and known maximum recovery times for tropical rainforests, as convincingly argued by the authors. Therefore, it would be very useful to also use a logarithmic color scale to make differences in recovery times more visible during the first three decades of recovery, while putting less emphasis on these differences for the remaining time.

We thank the reviewer for this positive evaluation of our figure and for providing comments for further improving it. We agree and now changed the color map to a logarithmic scale.

5. Minor issues.

1. L 109. There is a small typo here (irreplaceable).

Thanks, corrected.

2. L 134. This line exemplifies the inconsistent or dated use of the resilience concept and the disregard for mainstream current conceptualization.

Thanks for pointing this out. In line with our answer to major comment 1 we now write (p.4, l.129-134):
The recovery trajectories and recovery times of ecosystems following a perturbation depend on two components: resistance, defined as the ability to withstand disturbance and recovery, which

is the process of returning to the reference state as measured by the return rate (Fig. 1)^{26,27}. A common definition conceptualizes the combination of resistance and recovery as the system's resilience^{4,27-29} while other works define resilience more narrowly as the speed of return alone³⁰⁻³⁴.

3. L 161-162. To match the sequence on the system's trajectory from disturbance to recovery, resistance should go first on this list of resilience components.

We agree. We now write (p.5, l.161-165):

Here, we calculate resistance, return rates, recovery trajectories and recovery times to 90% similarity of old-growth forest conditions of species composition, as well as the underlying species diversity and abundance, of 16 taxonomic groups with 10856 species or morphospecies plus 23590 bacteria sequences (amplicon sequence variants), all measured in a well-resolved chronosequence in the Chocó lowland rainforest in Ecuador⁴⁷.

We also changed the order in the caption of Fig. 1 to (p.5, l.147):

Fig. 1. Quantifying resistance, return rates and recovery times of biodiversity in a tropical rainforest.

4. L 220-222. I would have liked to see a broader discussion about the potential consequences of the differences in recovery time of species composition between long-lived trees and many animal groups, considering that biological conservation should not only focus on species assemblages but also on the multiple cross-trophic level interactions that should be preserved.

We agree that this is a very important point to be mentioned in the discussion, which we are going to further analyse in future syntheses of species interactions in our research unit. We now added a discussion (p.7, l.225-229):

It remains to be studied whether the large variation in recovery times across taxa corresponds to an asynchrony, delay or decoupling of ecosystem functions and specific interaction partners with temporal succession. Our result showing that the species composition fully recovers for most taxa, but at different timescales, suggests that eventually all interactions may potentially rewire.

5. L 262. For the same reason mentioned above regarding the system's trajectory since the disturbance acts upon it (the perturbation period) to the potential end of the recovery, the order of the columns in this table seems odd. However, this could be easily improved by moving to the right end of the table the column with the predicted recovery time to 90% of old-growth forest. Also, on this table, does the asterisk in the title of the return rate column denote a multiplication?

Thanks. We now shifted the mentioned columns to the right end of the table. Yes, the asterisk indicates a multiplication in order to be able to read the numbers more easily and omit the zeros at the beginning of each number. We now clarified this by writing (p.10, l.270):

Return rates given in the table need to be multiplied by 10^{-3} to obtain the actual values.

6. L312-313. In many basic Statistics texts (and also in the basic R package), the Greek letter rho (ρ) is used to indicate the estimated Spearman non-parametric correlation coefficient. Unfortunately, this is not strictly correct. In Statistics, Greek letters are used to denote population parameters, whereas Latin characters are used to denote estimates. Thus, rho (ρ) should not be used to denote the estimated correlation coefficient, but another symbol should be used instead. To respect this statistical usage, many authors use the appropriate symbol r_s (r subscript s), and I invite the authors to do the same.

Thanks, corrected.

7. L 602. The authors know well the difference between Shannon diversity and Shannon index. However, I wonder if calling the Hill number of order $q = 1$ Shannon diversity and the one of order $q = 2$ Simpson diversity is warranted. My fear of the possibility of a reader of this manuscript getting confused between Shannon Index and Shannon diversity is fed by your insistence to clarify in parentheses that you are referring to Hill number of order $q = 1$ every time you mention the term 'Shannon diversity'. This is just a reflection on nomenclature, and I kindly ask the authors to think about it.

We thank you for pointing out this source of confusion. Indeed, we put some thought into our way of describing that measure and decided we would like to stick to the nomenclature as it is. The term “Shannon diversity” for the Hill-number of order $q=1$ was suggested by our co-author Anne Chao and is commonly used in papers using the framework of Hill-numbers and the term is used to distinguish the Hill-number from the Shannon-index which does not quantify effective number of species. The choice of adding the clarification in parentheses was only added in the revised manuscript in accordance with a question by reviewer #2 (comment number 39 of reviewer #2 in the rebuttal letter of the first revision) to clarify that we indeed refer to the Hill-number of order $q=1$ to make readers aware that we did not use the Shannon-index which is also a commonly used measure in ecology.

Referee #4 (Remarks on code availability):

The authors provide all the necessary code to make the results of the paper reproducible. However, not all the code is clearly described. For example, `Calculate_recoverytimes.py` and `Explain_recoverytimes.R`, which are essential to understanding how the recovery metrics were calculated, are not fully documented. It is important that the authors complete the documentation of all the code.

We thank the reviewer for this comment. We improved the documentation of the mentioned parts of the code in the README file associated with the code. We also provided a more detailed documentation in the essential parts of the scripts as comments. We would also like to point out that we already voluntarily published an initial reproducible version of our code on the CodeOcean platform, which significantly enhances the accessibility of all parts of the code to interested readers. The initial version will be updated upon acceptance of the work with the final code version.

Referee #5 (Remarks to the Author):

I co-reviewed this manuscript with one of the reviewers who provided the listed reports.

We thank the reviewer for co-reviewing our manuscript and provide a point-by-point response to each comment of reviewer 4 above.

Additional changes:

- 1. We added the word ‘agriculture’ in l. 154 to increase clarity**
- 2. We changed the title of Extended Data Fig. 7 to ‘Recovery trajectory of the abundance of frugivorous birds (a) the Shannon diversity of bats (b) and the Bray-Curtis similarity of saproxylic beetles (c).’ on p. 41 in l. 1103-1104.**
- 3. We changed the legend of Extended Data Fig. 7 on p. 41 l. 1104-1109 to ‘Blue and orange dots indicate values of active and recovering plots with cacao and pasture legacy, respectively, green dots indicate old-growth forest plots. The blue and orange lines indicate the recovery trajectory according to Eq. 5 for plots with cacao and pasture legacy, respectively. The grey curves indicate 95% confidence intervals estimated using a jackknife procedure (see methods).’**

References:

- 1. Ingrisch, J., & Bahn, M. Towards a comparable quantification of resilience. *Trends in Ecology & Evolution* 33, 251-259. (2018).**
- 2. Pimm, S. L. The complexity and stability of ecosystems. *Nature* 307, 321–326 (1984).**
- 3. White, L., O’Connor, N. E., Yang, Q., Emmerson, M. C., & Donohue, I.. Individual species provide multifaceted contributions to the stability of ecosystems. *Nature Ecology & Evolution* 4, 1594-1601. (2020).**
- 4. Pimm, S. L., Donohue, I., Montoya, J. M., & Loreau, M. Measuring resilience is essential to understand it. *Nature Sustainability*, 2, 895-897. (2019).**
- 5. Justus, J. (2008). Complexity, diversity, and stability.**

6. Hillebrand, H., Langenheder, S., Lebet, K., Lindström, E., Östman, Ö., & Striebel, M. Decomposing multiple dimensions of stability in global change experiments. *Ecology letters* 21, 21-30. (2018).
7. Van Meerbeek, K., Jucker, T. & Svenning, J.-C. Unifying the concepts of stability and resilience in ecology. *J. Ecol.* 109, 3114–3132 (2021).
8. Capdevila, P. *et al.* Reconciling resilience across ecological systems, species and subdisciplines. *Journal of Ecology* 109, 3102-3113 (2021).
9. Hodgson, D., McDonald, J. L., & Hosken, D. J. What do you mean, 'resilient'? *Trends Ecol. Evol.* 30, 503-506 (2015).
10. Poorter, L. *et al.* Multidimensional tropical forest recovery. *Science* 374, 1370–1376 (2021).
11. MacArthur, R. H. & Wilson, E. O. *The Theory of Island Biogeography*. (Princeton University Press, 2001).
12. Díaz, S. *et al.* The global spectrum of plant form and function. *Nature* 529, 167–171 (2016).
13. Junker, R. R. *et al.* Towards an animal economics spectrum for ecosystem research. *Funct. Ecol.* 37, 57–72 (2023).
14. Neyret, M. *et al.* A slow-fast trait continuum at the whole community level in relation to land-use intensification. *Nat. Commun.* 15, 1251 (2024).
15. Capdevila, P. *et al.* Life history mediates the trade-offs among different components of demographic resilience. *Ecology Letters* 25, 1566-1579 (2022).
16. Hoenle, P. O. *et al.* Rapid ant community reassembly in a Neotropical forest: Recovery dynamics and land-use legacy. *Ecological Applications*, 32, e2559. (2022).

MAJOR COMMENT:

L620. ‘We defined species composition similarity to the old-growth forest by the mean of all pairwise comparisons between each plot and each of the 17 old-growth forest plots. We did this to account for variation amongst old-growth forest’. Maybe I have commented on this before, or if I overlooked it my apologies. But if I understand it well what you have done, then in this approach you assume that compositional similarity is only driven by variation in time, but not by variation in space. Any two old-growth forest (OGF) plots will differ from each other because of spatial variation in topography, soils, past disturbance regimes, founder effects, etc. So I think you should calculate an average similarity between all pairwise OGF plots, and that is the maximum attainable similarity you can expect. Then you have to divide the average similarity between the specific SF plot and all 17 OGF plots by the average similarity amongst the 17 OGF plots to standardize the similarity to OGF. If you do not do that, then you may overestimate the time to recover towards OGF composition. If you do not want to change the analysis, then at least mention the potential pitfall and explain why you do not want to correct for it.

We thank the reviewer for this comment, as our method description may not have been sufficiently detailed. We agree that the ‘natural’ (spatial) variation among old-growth forests (OGF) needs to be explicitly considered. Given our spatial design where secondary forests (SF) and OGF are well mixed in space, the same spatial variation can be assumed for SF – OGF differences as for differences between OGF. Hence, we did compute the species composition similarity of each of the 17 OGF plots to each of the other 16 OGF plots explicitly and used the median difference as an asymptotic for standardization of recovery rate, consistent with the reviewers’ suggestion. To better acknowledge this property of the methodology choices, we now extended the description in the methods section as follows (now p.19, l. 593-603):

We defined the similarity of species composition to old-growth forests as the mean of all pairwise comparisons between an agricultural or secondary forest plot to each of the 17 old-growth forest plots. Note that old-growth forests were distributed across the entire study area and represent natural spatial variation in biodiversity and composition. To define the old-growth forest reference value for species composition similarity, we thus compared each old-growth forest plot against all other 16 old-growth forest plots, and calculated the mean similarity per plot. The median value of these 17 mean old-growth forest similarities was then used as the asymptotic reference for the recovery (

Ψ_{Ref}

in Fig. 1 and Eq. 5). Therefore, full recovery of species composition refers to the point at which the compositional similarity of an agricultural or secondary forest

plot to an old-growth forest cannot be differentiated from the similarity among old-growth forests

MINOR COMMENTS:

L87. Remove 'ongoing' and say directly 'The UN decade of ecosystem restoration' ... as when most of your future readers will read this paper the decade of restoration will have been finished.

Thanks, corrected.

L111. Define for the general reader what a secondary forest is.

Thanks, we now write (p.3, l.110-112):

Because 70% of tropical forests are secondary¹² (i.e. regrowing after deforestation¹³), their conservation can contribute substantially to achieving global biodiversity conservation goals^{14,15,16}.

L140. Maybe say FLOWER pollination, as this is aimed for a general audience, and some of my students even mix up pollination and dispersal.

Thanks, corrected.

L152 is calculated as THE TIME DIFFERENCE BETWEEN THE INTERCEPT OF THIS FUNCTION and.... ?

Thanks, corrected.

L252. Be explicit if this is the 95% confidence interval, and if it is of the mean or the population?

Thanks. We now write in the legend:

Thereby, confidence intervals indicate the range in which 95% of all jackknife curves are located.

All Figures: when you draw a regression line, and it is not significant, please show the regression line as broken, so that it is crisp and clear for the reader that it is non-significant (the reader could infer this from the confidence intervals, but that is not always easy to see)

Thanks. We now made the non-significant lines dashed and added to the legends of Fig. 3 and Extended Data Figs. 1, 2 and 7 the following sentence:

Dashed lines indicate curves with

not significantly different from 0.

L280. Define what are the whiskers (for example 1.5 x the length of the box??)

Thanks, we now write in all appropriate legends:

whiskers indicate 1.5x the interquartile range

L631. Explain if alpha diversity is then the average alpha diversity of the two plots that are compared?

Thanks. We now write (p.20, l.608 - 611):

where the alpha-diversity is calculated from the joint assemblage of both plots (i.e. by summing over the abundance of each species in each of the plots after joining both plots, see ref. 54) and the gamma-diversity was defined for each pair of plots by pooling data from both communities.